# Rhythmic sampling of multiple decision alternatives in the human brain

Marcus Siems [1] ✉, Yinan Cao [1], Maryam Tohidi-Moghaddam [1], Tobias H. Donner [1] & Konstantinos Tsetsos [1,2] ✉

Humans and other animals navigate decisions by sequentially attending to (sampling) subsets of the available information. The internal dynamics of the selective sampling of decision-relevant information remain unknown. Here we use magnetoencephalography recordings and neural decoding to track the spontaneous dynamics of the locus and strength of covert attention as human participants performed a three-alternative perceptual choice task. The strength of covert attention fluctuated rhythmically around 11 Hz. A shift of attention from one alternative to another tends to occur at the trough of this oscillation, presumably enabling comparisons. These shifts further reset the attentional oscillation. By contrast, at the peak of the oscillation, attention tends to increase the focus on the currently sampled alternative, presumably deepening processing of that alternative. We propose intrinsic attentional oscillations as a core mechanism governing the flexible sampling of decision alternatives.

Every day of our lives we make thousands of decisions based on varying amounts of external information. For example, deciding when to cross the street involves considering multiple pieces of information - i.e., number of cars, state of the traffic lights, movement of other pedestrians, etc. - and translating these into actions guided by incorporating internal goals (e.g., the need to catch the bus). Unlike the simple two-alternative perceptual decision tasks typically used in decision neuroscience – where incoming sensory information can be directly integrated into motor plans[1–5] – in many real-life decisions, the brain may not be able to parse and process all decision-relevant information in parallel[6,7]. Thus, subsets of the available information need to be sampled serially as the decision unfolds[8,9], i.e., via attentional mechanisms.

With attentional resources being limited[10–12] a trade-off during challenging decision computations emerges: The brain needs to process and evaluate single alternatives while also reallocate attention between alternatives to compare them[9,10,13]. Recent work on the neurophysiological basis of attention in simple detection tasks has provided a candidate mechanism that might help decision-makers navigate this trade-off: During periods of sustained focus on one stimulus, attention is not static but oscillates[14–20]. This oscillation is evident in behavioral reports[15,17,18,21–23], and neuronal activity in the theta[15,17,21,24], alpha[25,26] and gamma[19,21,22,26] frequency ranges. As a result, when attention is at an oscillatory peak, stimulus processing is enhanced[14,15,17,19,22,27–29]. By contrast, when at an oscillatory trough, attention is relatively disengaged and might shift from one stimulus to another[30]. Thus, one cycle of this oscillation can confine and support distinct attentional functions[14]. Investigations in pupillometry and neuromodulatory systems further highlight the distinction between a focused versus a disengaged and more exploratory attentional mode on longer timescales[31].

Here, we hypothesized that the rhythmic alternation between focused and disengaged attention may underlie the narrow and exploratory processing needed for valuation and comparison during multi-alternative decisions, respectively. Previous neurophysiological studies have established the role of rhythmic attention during simple detection tasks[32], tasks that could be solved through bottom-up strategies based on perceptual salience. However, it remains unknown whether equivalent rhythmic mechanisms are implicated during demanding multi-alternative decisions that recruit top-down mechanisms and require protracted deliberation about alternatives.

[1]Department of Neurophysiology and Pathophysiology, University Medical Center Hamburg-Eppendorf, Hamburg, Germany. [2]School of Psychological Science, University of Bristol, Bristol, UK. ✉e-mail: m.siems@uke.de; k.tsetsos@bristol.ac.uk

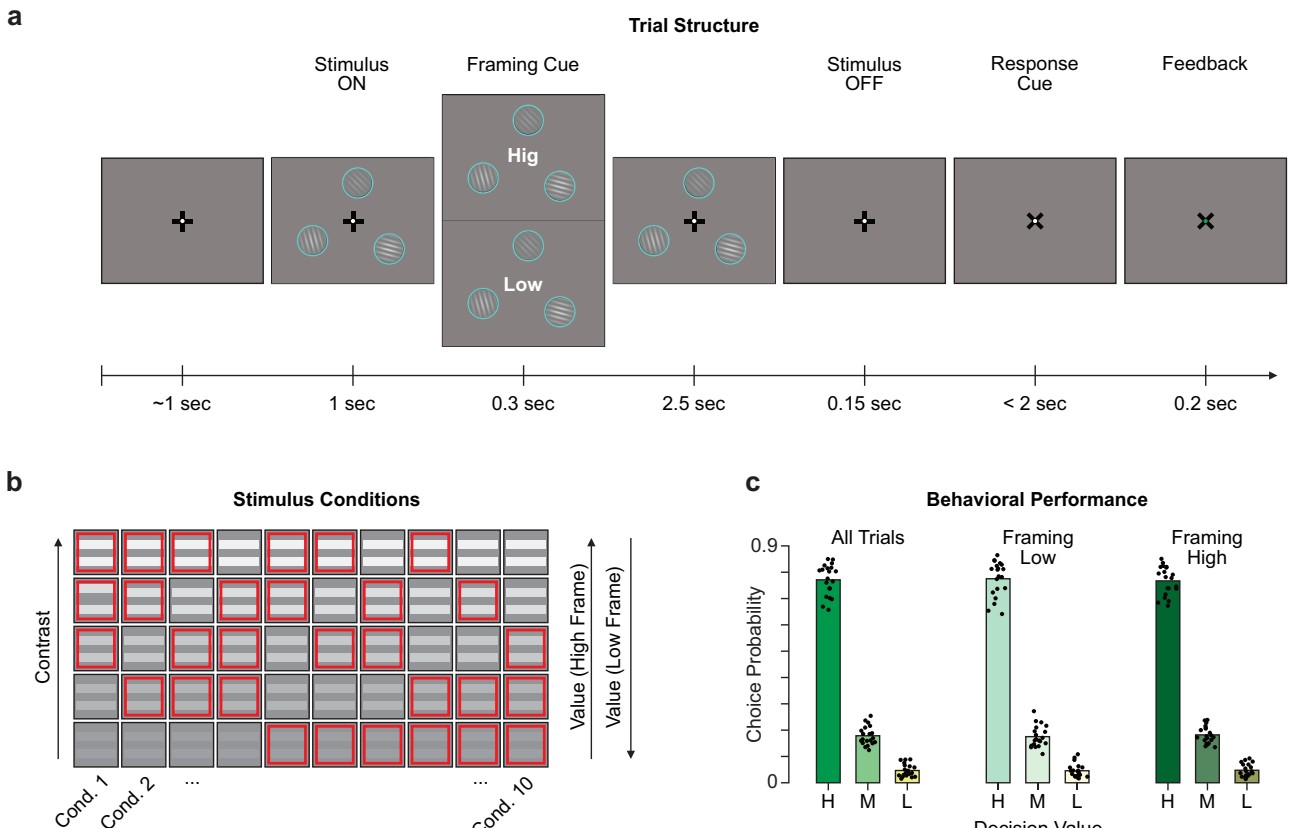

**Fig. 1 | Experimental task & behavioral performance. a** Trial structure of the three-alternative decision-making task with variable task framing. Stimulus size, contrast and eccentricity are indicative and have been changed for visualization purposes. Participants were asked to select the stimulus that had the highest (framing high: 'Hig') or the lowest (framing low: 'Low') contrast. **b** Schematic of the ten unique Gabor grating contrast conditions. For each trial, the three eccentric stimuli (red squares per row) were randomly drawn without replacement from a set of five distinct contrast levels (columns, from low to high contrast). Depending on the trial framing, the lowest contrast grating constitutes a low decision value (framing high) or a high decision value (framing low). **c** Choice probability across the three alternatives (H, M, L indicates high, medium, low decision value) for all trials and within each framing condition. The dots indicate individual participant's ($n_{participant} = 20$) choice probability.

Here, we harnessed the high temporal resolution of MEG recordings to continuously track the locus and strength of covert spatial attention during protracted multi-alternative decisions. Focusing on covert attention enabled us to gauge rich sampling dynamics that cannot be captured by oculomotor activity[24,27,33–35], making them difficult to assess when only overt attention is tracked[8,9,13,36]. Using a multi-alternative choice task that requires valuing and comparing decision alternatives on intrinsically set timescales enabled us to identify attention-driven serial processing of information while the decision unfolded. Importantly, we decoupled low-level stimulus features (contrast) from the decision value through randomized task-framing. We expected that the intrinsic dynamics of attentional allocation are rhythmic.

Indeed, we show that the strength of covert attention intrinsically oscillates around 11 Hz. Refocusing attention to the previously sampled alternative tends to occur at the peak of this oscillation, while switching to a different alternative occurs at the trough and resets the ongoing rhythm. Thus, the conflict between gathering additional information about an alternative and exploring other alternatives is resolved through rhythmic sampling. We propose that rhythmic attentional sampling supports the flexible information processing needed to perform cognitive tasks with increased processing demands.

## Results

We recorded non-invasive magnetoencephalography (MEG) from a total of 20 participants (9 female, $age_{mean} = 28.05$ years $\pm$ $age_{STD}$ 4.6 years). The participants performed two tasks in an interleaved fashion. The main task was a three-alternative visual perceptual choice task (Fig. 1a). We further used a retinotopic stimulus localizer task to be able to track the spontaneous allocation of attention to the visual field locations occupied by the decision alternatives in the main task (see below). Trials of the main task started with a central fixation, followed by three static Gabor-patches of different contrast levels, which were simultaneously presented at different locations on an imaginary radius of 2.2° eccentricity (Fig. 1b). The duration of the sensory evidence presentation was fixed at 2.5 s (so-called "interrogation protocol"[37]), to eschew speed-accuracy tradeoffs and provide ample time for sequential sampling of the different choice alternatives[38]. In two framing conditions (randomly assigned on each trial), participants were instructed to either choose the Gabor-patch with the highest or the lowest contrast, with the framing instruction being revealed one second after stimulus onset. By decoupling the bottom-up stimulus contrast from the task-relevance of the choice alternatives, this task enabled us to probe endogenous attention allocation, independent of exogenous attention capture[39]. Participants performed the task with around 77% accuracy ($range_{min,max} = [66\%, 86\%]$; Fig. 1c) and there was no accuracy difference between task framings ($|t_{19}| = 1.679$; $p = 0.11$).

### Tracking locus and strength of covert attention using MEG
We developed an approach to continuously track the locus and strength of covert spatial attention during decision processing. Specifically, we used the retinotopic stimulus localizer to train an inverted encoding model[40] on the angular positions of the three stimulus

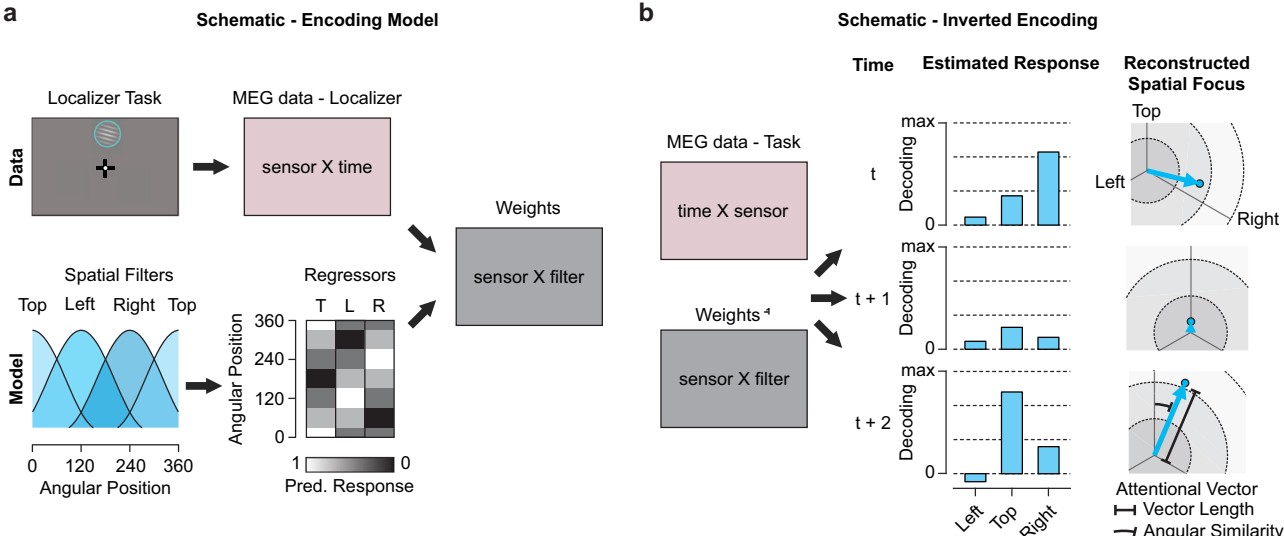

**Fig. 2 | Reconstructing the locus and strength of spatial attention with inverted encoding models. a** Training the encoding model. The encoding model consists of a set of linear spatial filters defining the response given a certain stimulus. It comprises each information channel's tuning curve, which when discretized at presentation angles (T top, L left, R right) forms the Design Matrix with the normalized predicted response (from 0 to 1). In combination with MEG data from the retinotopic stimulus localizer (see "Methods"), the weighting of the information channels within each MEG sensor can be estimated by solving the general linear model (GLM). **b** Inverting the weights to estimate the channel responses during the decision-making task. We multiplied the inverted weighting matrix with the MEG data to estimate the response of the information channels at each time point (t). From the estimated responses, we reconstructed the focus of spatial attention as the vector sum of the three spatial channel responses (left, top, right). The resulting Attentional Vector featured first, the length corresponding to the overall amount of attention and second, the angular similarity to the three stimulus targets.

positions occupied by the choice alternatives in the main task (Fig. 2a and Supplementary Fig. S1; see Methods). Presenting a single stimulus on one of these positions on each trial of the localizer task enabled us to "tag" the strength of the representation of any stimulus at each position within the retinotopic maps of the visual cortical system[41,42]. Top-down attention modulates visual stimulus representations in these visual field maps in a spatially-selective manner[10,11,43,44]. Thus, our inverted encoding model enabled us to track to spontaneous allocation of attention to these different stimulus positions in the main decision-making task, where all three locations were simultaneously occupied by stimuli that competed for processing[45]: The expression of the allocation of attention during that task should manifest as a (stimulus-independent) increase or decrease, respectively, of the strength of the stimulus representations at different positions in the visual field maps. Specifically, the inverted encoding model yielded a time-resolved estimate of the response to each of the three angular positions during the decision-making task (Fig. 2b). We reconstructed spatial attention as the vector sum of the three channel responses (left, top, right) and focused on the resulting vector (in the following 'Attentional Vector') to track internal attentional fluctuations. The vector length corresponded to the strength of attention. Thus, high attention strength might signal increased sensory processing, task engagement or arousal. The angular similarity to the three stimulus positions corresponded to the locus of attention. We further tracked the eye-gaze position to ensure central fixation and disentangle covert attention from eye movements.

### Attention strength oscillates between 7–13 Hz

Building on recent work on the neurophysiological basis of attention in simple perceptual tasks[32], we hypothesized that attention strength (i.e., vector length) would vary throughout the trial as a result of distinct processing demands in the sensory- and decision-phases of the task. First, we quantified the trial-averaged attention strength (Fig. 3a). We z-scored attention strength over the full stimulus-presentation window to compare visual representation dynamics between participants and identify the periods of strongest attention. Overall, during

the full stimulus presentation attention strength is stronger than during the pre-stimulus period (Supplementary Fig. S2; $t_{19, \text{vs.pre-stimulus}} = 5.79 - 8.83$, $p < 1.5 \times 10^{-5}$). After the stimulus onset, attention strength steeply ramps up and displays three peaks at around −0.8 s, 0.2 s and between 0.7-1.2 s (from framing cue) overlapping with key moments of the trial (stimulus onset, framing and early decision formation) before it slightly decreases in later periods. For external validation of this metric, we compared attention strength to the trial-averaged pupil diameter, a peripheral proxy for arousal and task engagement[46]. The pupil diameter (Fig. 3b) resembled the attentional vector distribution (Fig. 3a), albeit with more sluggish dynamics ($r_{\text{pupil,veclen}} = 0.54$; $p = 0.007$).

Importantly, there were substantial fluctuations of attention strength between trials (Fig. 3c), which we quantified using spectral analyses (Fig. 3d–g). Descriptively, attention strength fluctuated most strongly in the frequency range between 8–22 Hz (Fig. 3 d), and significantly increased relative to the pre-stimulus period (−1.5 to −1 s prior to framing) between 4–16 Hz, and decreased between 25–50 Hz (Fig. 3e,f; $|t_{19}| > 2.38$, $p_{\text{FDR}} < 0.05$). The prolonged increase of the fluctuations between 7 − 13 Hz was most pronounced between 0.5 seconds prior to 1.5 s after the framing cue (Fig. 3 g).

To examine the link between attentional and neural dynamics, we pairwise correlated the time-frequency resolved attention strength fluctuations with the frequency-equivalent neuronal signal amplitude envelope (Supplementary Fig. S3) over the stimulus presentation period. Following the spectral properties of attention strength (Fig. 3g), we focused our analysis on the alpha (7–13 Hz) and gamma (25–50 Hz) frequencies as well as on broadband activity (0.2–200 Hz). We found strong positive correlations of attention strength with visual cortex activity in alpha, gamma, and broadband frequencies ($r > 0.25$; $p_{\text{FDR}} < 0.05$; Fig. 3h). Attention-neuronal correlations in the alpha-band highlighted lateral prefrontal cortical areas ($p_{\text{FDR}} < 0.05$).

The observed attentional fluctuations could be a consequence of stimulus-evoked cortical (and in particular visual) activity[47]. However, conversely to the 7–13 Hz increase in attention strength, we observed stimulus-evoked suppression of cortical activity in the alpha and

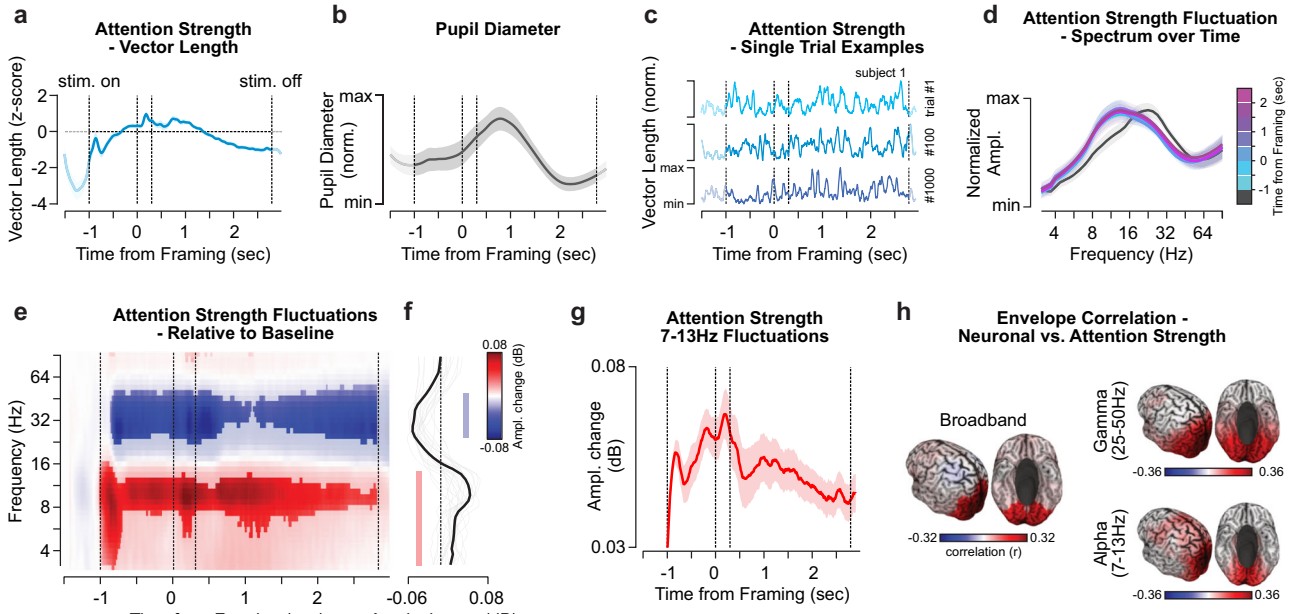

**Fig. 3 | Attention strength fluctuates at around 11 Hz. a** The distribution of the z-scored attention strength, i.e., vector length. The blue line indicates the average over trials and participants, the shaded area the standard error of the mean (SEM). Vertical dashed lines indicate the stimulus onset, framing cue onset and offset, as well as the stimulus offset (from left to right). **b** The average normalized pupil diameter during task execution. The thick line and shaded area indicate the average and the SEM over participants. **c** Single trial example traces of the normalized attention strength. The depicted trials are from participant 1, session 1. **d** Windowed normalized amplitude (Ampl.) spectra of attention strength fluctuations over the trial. Lines indicate the participant average, shaded areas display the SEM. The lines are color-coded according to the window onset (0.5 sec. window length) with regard to the framing cue, from early (blue) to late (pink) windows. **e**–**g** Frequency-specific change of attention strength amplitude relative to baseline. The frequency-wise amplitude strength values were normalized by dividing the average amplitude during the 0.5 s prior to stimulus onset and displayed in dB. **e** Time-frequency

resolved (TFR) distribution of the amplitude of attention strength fluctuations. Red and blue colors indicate an increase or decrease relative to baseline, respectively. The values were statistically masked at $p_{FDR} < 0.01$ within each frequency. Vertical dashed lines as in (**a**). **f** Average frequency-wise amplitude change (as in **e**) throughout the stimulus presentation period. Thick and thin lines display the average and single participant amplitude changes ($n_{participant} = 20$), respectively. Colored bars denote change from baseline (two-sided $t$ test(19), $p_{FDR} < 0.05$, FDR-corrected over frequencies). **g** Attention strength fluctuation amplitude averaged for frequencies between 7–13 Hz, relative to baseline in dB. The shaded area indicates the SEM. **h** Correlation of trial-averaged amplitude envelopes of the neuronal signal (compare Supplementary Fig. S2) and attention strength fluctuation displayed on the cortical surface[77]. The correlation was computed for the broadband as well as frequency resolved signals. The values were statistically masked at $p_{FDR} < 0.05$ within each panel.

neighboring frequency ranges throughout the entire trial duration (Supplementary Fig. S3). Similarly, we assessed the dynamics of the estimated responses (Fig. 2b), the precursor to the attentional vector, and remapped each response to its decision value class (High, Medium, Low). The estimated responses for all alternatives displayed a broadband decrease between 8–45 Hz (Supplementary Fig. S4) and could not explain the increase of attention strength between 7–13 Hz. Further, the difference in physical contrast between framing conditions appeared to not affect the estimated response within its value class apart from a ~ 0.25 second stimulus onset response (Supplementary Fig. S5). Thus, the increased attentional fluctuation around 11 Hz cannot be readily explained by stimulus-induced cortical activity responses or low-level feature decodability dynamics (Supplementary Figs. S3, S4).

**Attention fluctuations correlate with behavioral performance**
As shown above, attention strength varies throughout the trial and displays prominent band-limited fluctuations, but it remains to be shown whether these attention dynamics are relevant for decision-making performance. To evaluate the behavioral relevance of these dynamics, we compared attentional fluctuations between correct and error trials. In a first analysis, broadband attention strength (Fig. 3a) was slightly larger (although not significantly) late in the decision phase for correct choices relative to incorrect ones (Fig. 4a; all $|t_{19}| < 1.5$, $p > 0.05$).

In a second analysis, we repeated the comparison between correct and error trials, focusing on the frequency resolved attention strength fluctuations (compare Fig. 3e). We identified significantly increased fluctuations associated with correct choices from 1.5 s post-framing until the end of the trial, peaking in a frequency range from 8–16 Hz ($p_{FDR} < 0.05$). This indicates that band-limited attention strength fluctuations are predictive of choosing correctly (Fig. 4b). Importantly, we matched correct and error trials in difficulty and framing conditions (see Methods).

**Attention allocation is nested within an 11 Hz rhythm**
The above analyses highlight prominent modulations of attention strength around 7–13 Hz. However, these findings are unrelated to the locus of attention. Is there a relationship between attention strength and the dynamics of spatial attention allocation? According to recent literature and the "rhythmic sampling" framework[14,19], attention strength is expected to index the stability of the attention locus. In particular, when attention strength peaks, the tendency to maintain the representation of one alternative is strong (locus stability). When attention is weak, the tendency to switch to different alternatives increases (locus instability)[14,48], with reallocations occurring swiftly, analogous to 'attentional saccades'[18,27,49].

To test this hypothesis directly, we extracted brief attentional events of maximal engagement with one decision alternative and assessed their temporal dynamics. This approach allows us to quantify

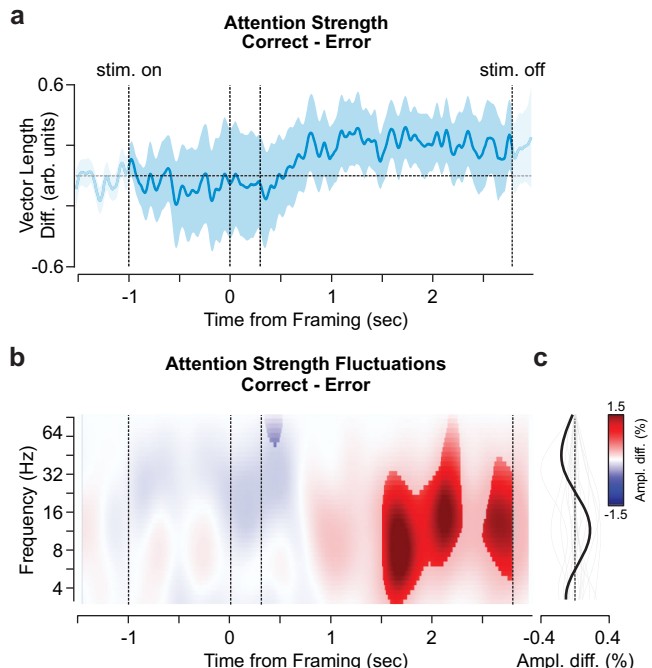

**a** Attention Strength Correct - Error

**b** Attention Strength Fluctuations Correct - Error

**Fig. 4 | Attention strength fluctuations correlate with performance. a** The distribution of the overall attention strength difference, i.e., vector length, between correct and error trials (condition matched, see Methods). The blue line and shaded area indicate the average over participants ($n_{participant} = 20$) and the standard error of the mean (SEM). Vertical dashed lines denote the stimulus onset (stim. on), framing cue onset and offset as well as the stimulus offset (stim. off, from left to right). **b, c** Time-frequency resolved (TFR) attention strength fluctuation differences between correct and error trials as a percent change to correct. **b** The distribution of TFR amplitude differences (Ampl. diff., correct-error) over the full spectrum and trial duration. The values were statistically masked at $p_{FDR} < 0.05$ over the spectrum. **c** Frequency-wise amplitude change (as in **b**) averaged over the stimulus presentation period (−1 to 2.8 s). Thick and thin lines display the average and single participant amplitude changes, respectively.

several aspects of information sampling, such as identifying periods when participants are engaged with a given alternative, or quantifying how long it takes to switch attention between alternatives. Crucially, we can test if these temporal regularities occur within the prominent 7–13 Hz attention strength rhythm.

Attentional events are points in time when the Attentional Vector aligns with one of the three decision alternatives. Quantitatively, these are timepoints of maximum angular similarity (MAS) between the Attentional Vector and one of the decision alternatives (Fig. 5a). Hereby, MAS displays no systematic variations over the course of a trial. Next, we binarized MAS by applying a threshold (MAS 90% percentile, see Methods) to define brief events associated with selective processing of a single alternative (akin to an 'attentional saccade'). Importantly, the MAS metric is defined in a circular space and thus under completely random conditions (periods without visual stimulation) or when decodability of the alternatives is weak (inattentiveness) MAS will exceed the angular threshold at random times. In these cases, we would falsely infer that attention is allocated to one alternative. We thus included a second threshold on the vector length to exclude noise-driven events as much as possible (Fig. 5b and Supplementary S6). An attentional event is thus recorded when both the attention locus is selective for one alternative, and the attention strength is sufficiently high. Importantly, these events appear to carry information about the decision behavior (Fig. 5c): During correct choices the high-value alternatives are sampled more often (logistic regression predicting choice from event counts within participants, weights $t$ test against zero: $t_{19,chooseH,stayH} = 2.35$, $p = 0.03$;

$t_{19,chooseH,switchH} = 2.42$, $p = 0.02$). When choosing the second-highest value alternative the highest value target is sampled less ($t_{19,chooseM,stayH} = -2.52$, $p = 0.02$; $t_{19,chooseM,switchH} = -2.71$, $p = 0.01$). In addition, the mid-value alternative was sampled less frequently when the low-value alternative was chosen ($t_{19,chooseL,stayM} = -3.83$, $p = 0.001$; $t_{19,chooseL,switchM} = -3.91$, $p = 0.001$). Temporally, increased neuronal representation of the highest valued alternative was positively related to correct choices (Supplementary Fig. S7) during an early (around 0.4-0.6 s post framing, $|t|_{19,IEMhigh-IEMmid} > 0$, $p_{cluster} < 0.05$ cluster-mass permutation corrected) and a late trial interval (around 1.8-2.8 s, $|t|_{19,IEMhigh-IEMmid} > 0$, $p_{cluster} < 0.05$). We quantified the relationship with logistic regression analyses from the difference in estimated IEM channel responses (IEM$_{high}$ - IEM$_{mid}$ predicting correct choices) for each participant. Further, we found pupillometric measures of neuromodulation to display a general link between arousal and information sampling (Supplementary Information & Supplementary Fig. S8).

Next, we characterized the dynamics of attentional events in a data-driven fashion. To assess temporal structures, we timed the interval in-between successive attentional events (inter-event-intervals, IEI; Fig. 5d). Assuming that attentional allocation is governed by a random Poisson process (i.e., no temporal structure), the IEI histogram would display an exponential decay of probability density with interval duration (Fig. 5d, e, dashed lines). However, we identified clear deviations from this prediction, as the IEI histogram exceeded random conditions at 0.13 s and at around 0.18 s ($p_{FDR} < 0.05$). These short latencies indicate that events tend to cluster together in a burst-like pattern over the course of the trial and confirm that the temporal structure of attention allocation is not random.

We next examined whether the dynamics of attentional allocation change as a function of the identity of the previously attended alternative. Attention can either re-focus on the previously attended alternative (staying) or shift to a different alternative (switching). As evident in the corresponding IEI histograms, staying and switching events exhibited distinct temporal profiles: staying predominantly occurred after 0.09 s and 0.17 s from the preceding event; and switching mostly after 0.05 s and 0.14 s (Fig. 5e, f and Supplementary S9), independent of the framing condition (Supplementary Fig. S10). Generally, we observed more stay than switch events (mean $p_{Stay} = 0.63$, $p_{Stay,25\%-75\%} = 0.60 - 0.64$) and that switching was relatively increased right after stimulus onset and later in the decision phase (Supplementary Fig. S11).

The IEI histograms associated with stay and switch events can help assess the (potential) rhythmicity of these events: Assuming that events occur at distinct phases of an underlying rhythm, we can expect maxima (modes) of the IEI probability density reappearing in the first, second, third, etc. cycle of a putative oscillation. Thus, the latency between reappearing modes in the IEI histogram (effectively the cycle length) would enable us to quantify rhythmicity. Both staying and switching events displayed approximately 0.09 s first-to-second mode latencies (median$_{stay}$ = 0.09 s, range$_{stay,25\%-75\%}$ = 0.088 s – 0.098 s; median$_{switch}$ = 0.088 s, range$_{switch,25\%-75\%}$ = 0.08 s – 0.095 s) – corresponding to rhythmicity at ~11 Hz (Fig. 5g). Thus, the probability distribution of the events is consistent with the frequency of the dominant attention rhythm (Fig. 3) and the events appear to follow the attentional rhythm (Fig. 5i).

We further observed a prominent offset between switch and stay latencies (Fig. 5e, f). While stay events displayed the first mode at around 0.085 s (median = 0.085 s, range$_{25\%-75\%}$ = 0.076 s – 0.094 s), and switch events earlier at around 0.045 s (median = 0.046 s, range$_{25\%-75\%}$ = 0.041 s – 0.054 s; paired $t_{18,stay-switch}$ = 12.35, $p < 10^{-9}$). With respect to each participant's event cycle length (Fig. 5g) we identified the first 'switching' at around 0.5 (median = 0.51 cycle length, range$_{25\%-75\%}$ = 0.46 - 0.64) and the first 'staying' mode after a full cycle (median = 0.95 cycle length, range$_{25\%-75\%}$ = 0.79 - 1.01) relative to the previous event. The switch-to-stay phase-shift was around half a cycle

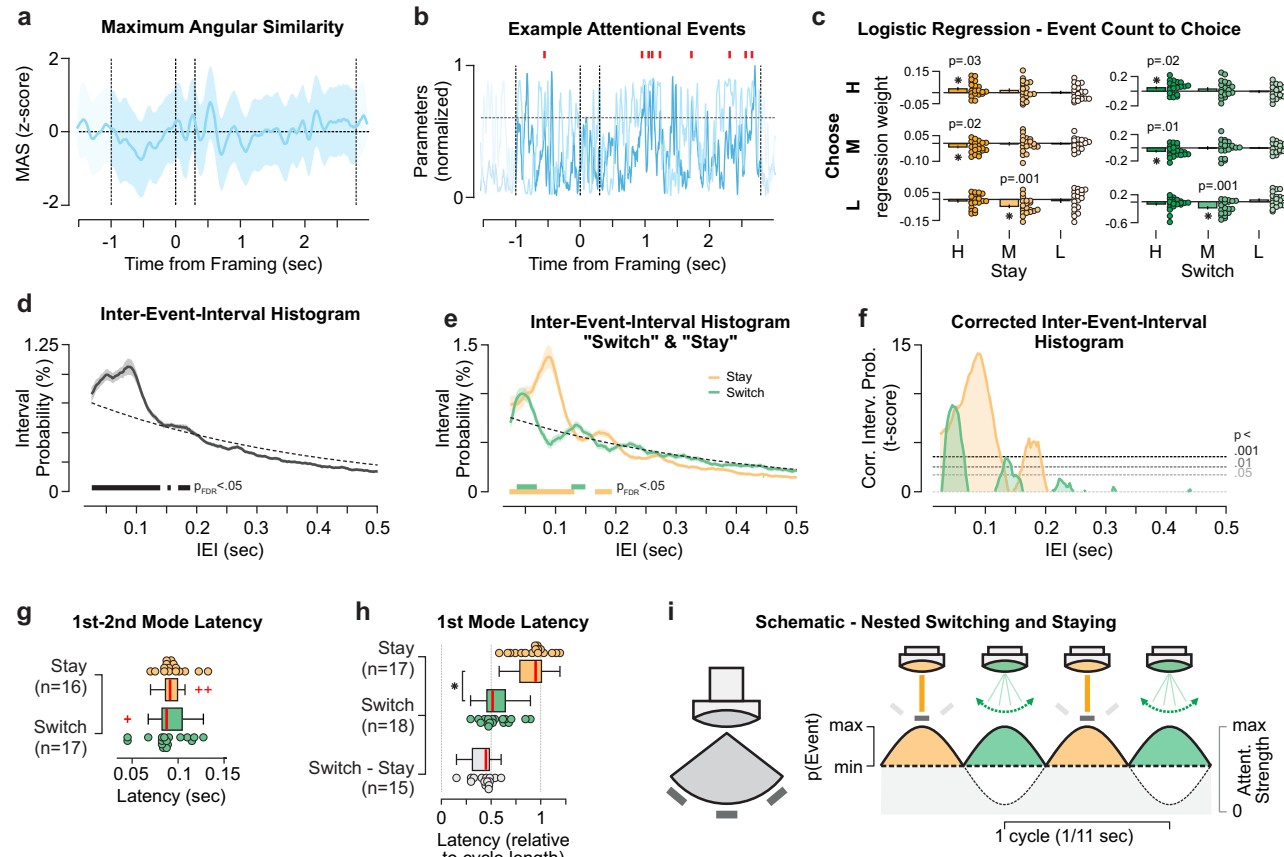

**Fig. 5 | Attention allocation to distinct stimuli is nested within an 11 Hz rhythm.**
**a** Maximum angular similarity (MAS) averaged over trials. The blue line indicates the z-scored average over participants ($n_{participant}$ = 20). The shaded area displays the SEM. Vertical dashed lines indicate the stimulus onset, framing cue onset and offset, as well as the stimulus offset (from left to right). **b** Schematic definition of attentional events in an example trial. An attentional saccade (red mark) is recorded when vector orientation (90% MAS percentile) and attention strength are sufficiently high (90% vector length percentile). **c** Logistic regression predicting choice (H, M, L) from the event count per trial. Bars and vertical lines denote the mean and SEM over participants ($n_{participant}$ = 20). Asterisks denote significant differences from zero (two-sided $t$ test(19), uncorrected $p < 0.05$). **d** Inter-event-interval (IEI) probability histograms over all trials and attentional events. The black line indicates the average over participants, the shaded area the SEM. The black bars denote significant (two-sided $t$ test(19) per timepoint; $p_{FDR} < 0.05$, FDR-corrected over time) interval probabilities exceeding random conditions (dashed line). **e** Inter-event-interval histograms separately for stay (orange line, ± SEM) and switch (green line,

± SEM) attentional events. The bars denote significant (two-sided $t$ test(19) per timepoint; $p_{FDR} < 0.05$, FDR-corrected over time) interval probabilities exceeding random conditions. **f** Same data as in (**e**) plotted as t-score against random, i.e., corrected inter-event-interval probabilities (Corr. Interv. Prob.). Dashed lines indicate the uncorrected significance levels and zero. **g** Box-plots of first-to-second mode latencies. Colored dots display single participants, color-coded as above. N denotes the number of participants with at least two significant peaks within the IEI histogram. **h** First mode latency relative to the cycle length (see **f**). Asterisk denotes a significant difference between event types (two-sided $t$ test(19), uncorrected $p = 1.03*10^{-8}$). N denotes the number of participants with significant peaks within the IEI histogram. **i** Schematic - Covert attention "spotlight" allocations nested within the underlying 11 Hz rhythm (gray) with "staying" on the same stimulus and "switching" the focus of attention offset by half a cycle. All box plots (**g**, **h**) display the 25%–75% percentile range (box), median (red line) and whiskers maximally up to 1.5 times the inter-quartile range. The schematic was created using Affinity Designer (Version 1.10.8, Serif Europe Ltd).

(median difference = 0.45 cycle length, range$_{25\%-75\%}$ = 0.31 - 0.48; Fig. 5h, gray box).

In sum, the temporal regularities characterizing attentional events provided a window into the dynamics of covert attentional sampling during decision formation. These results show that both event types rhythmically reappear over time, offset by half a cycle (Fig. 5h, i)[20], representing two distinct attentional modes: focus and reallocation.

**Attentional saccades impact the ongoing attentional rhythm**
The offset between switch and stay events can help us address another central hypothesis: Are attentional saccades impacting, for example, by resetting[14,17,19], the attentional rhythm? The related alternative hypothesis would be that attentional events passively follow the attentional oscillation ("follow" hypothesis), with stay at the peak and switch events occurring at the trough. Under the follow hypothesis, event probabilities should depend on the previous event identity (Fig. 6a). For example, switch events should occur i) half a cycle after a

previous stay event but ii) at full cycle latency following switches. This would induce a history effect on the IEI histograms: Depending on the preceding event, IEI modes should be temporally offset. Further, when ignoring the history, this should result in IEI mode-latencies at double the frequency and no offset between switch- and stay events (Fig. 6a, lower panel).

To directly test the "follow" hypothesis, we assessed second-order dependencies in longer event sequences. We defined IEI histograms for four sequence types: Switch after stay, switch after switch, stay after stay, stay after switch. Under the following hypothesis, we would expect repetition (switch-after-switch, stay-after-stay) histograms to be offset by half a cycle relative to alternations (Fig. 6a). However, we found that the IEI histograms within each event type were invariant to the preceding event (Fig. 6b, c), and the temporal dynamics patterns expected under the "follow" hypothesis do not hold.

Taken together, this finding challenges the concept that the attentional rhythm is unaltered over time. Attentional events do not

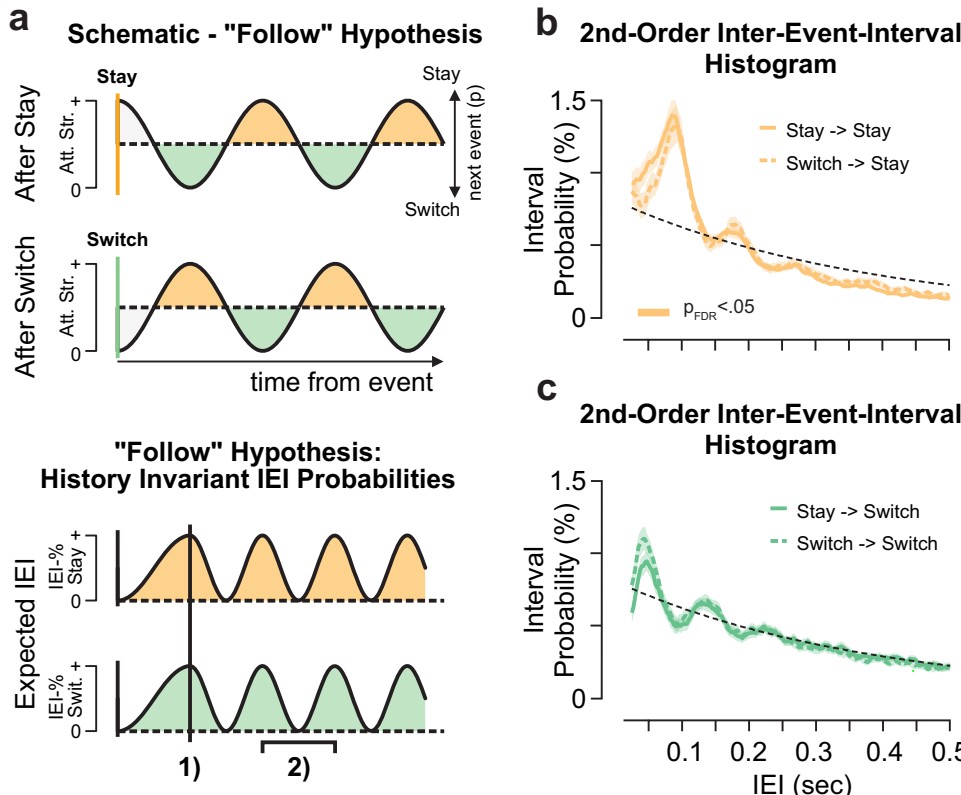

**Fig. 6 | Schematic and empirical second-order sequence inter-event-interval probabilities. a** Schematic of IEI probability depending on previous event identity. If the attention oscillation persists irrespective of event identity ("follow" hypothesis), we would expect **1)** no offset between stay and switch modes and **2)** mode-latencies at double the attention modulation frequency (Fig. 3e vs. 5 g) when event history is ignored. Attention strength is abbreviated as Att. Str. The schematics were created using Affinity Designer (Version 1.10.8, Serif Europe Ltd). **b, c** Second-order sequence effects on inter-event-interval probabilities separately for (**b**) stay (orange line, mean ± SEM, $n_{participant} = 20$) and (**c**) switch (green line, mean ± SEM) attentional events. Solid and dashed colored lines indicate the mean interval probability following a stay or a switching event, respectively. The bars denote significant (two-sided $t$ test(19) per timepoint; $p_{FDR} < 0.05$, FDR-corrected over time) interval probabilities differences between respective 2nd-order sequence types, color-coded as above. The empirical data discards the "follow" hypothesis (compare **a**).

passively follow the attentional strength oscillation. In the next section, we present evidence for an "interaction" hypothesis, in which attentional saccades can impact the rhythm, starting it anew.

**Attentional switching resets rhythmic attention**

To better understand how attentional events might alter the attentional oscillation, we focused on attention strength around event onset. We averaged attention strength in a time window from −0.2 s prior to 0.2 s after each event (Fig. 7a, b; MAS Supplementary Fig. S12). Prior to events, the traces of attention strength were relatively consistent over participants but largely unaligned between switch and stay (mean $r_{pre} = 0.13$, $r_{pre,25\%-75\%} = -0.05 \cdot 0.34$; $t_{19} = 0.49$, $p = 0.31$; correlations on detrended signals). By contrast, right after the event, the attention strength traces strongly overlap between both event types (mean $r_{post} = 0.60$, $r_{post,25\%-75\%} = 0.54 - 0.76$; $t_{19} = 2.59$, $p = 0.009$; correlations on detrended signals). Moreover, attention strength was weaker around switching- compared to stay events (Fig. 7c) both pre- (mean $ratio_{Switch/Stay} = 0.90$, $range_{25\%-75\%} = 0.85 - 0.94$; paired $t_{19} = -8.09$, $p < 10^{-6}$) and post-event (mean $ratio_{Switch/Stay} = 0.92$, $range_{25\%-75\%} = 0.88 - 0.95$; paired $t_{19} = -7.40$, $p < 10^{-6}$) but increased post-event by about 2% (mean $\Delta ratio_{Switch/Stay} = 0.019$, $range_{25\%-75\%} = 0.011 - 0.029$; paired $t_{19} = 7.21$, $p < 10^{-6}$).

Focusing on pre-event attention strength, we observed a faster fluctuation for switch events (Fig. 7a, b). We thus quantified the dominant fluctuation frequencies pre- and post-event, and for both event types. Attention strength fluctuated at around 9–12 Hz before

and after staying events, as well as after switching events (Fig. 7d). However, prior to switching, there was an increase (paired $t_{16} = 5.82$, $p_{FDR} = 10^{-4}$) of the fluctuation frequency to around 20–26 Hz, dissociating switch and stay events. Correlations between event distributions over the trial and band-limited neuronal activity further underline this qualitative dissociation between switch and stay events in an extended visual and lateral prefrontal cortical network (Supplementary Fig. S13). Overall, our findings suggest that switching events in particular are preceded by a higher frequency profile (Fig. 7d) and they 'reset' the ongoing attention oscillation to the dominant frequency around 11 Hz. Following both event types, the 11 Hz attentional rhythm starts from the local maximum (Fig. 7a, b) and attention strength peaks at around 0.09 s and 0.18 s post-event, i.e., approximately after one and two full cycles. This reset can also explain the lack of second-order sequence effects (Fig. 6).

We further directly compared the temporal contingencies between post-event attention strength and the switch/stay IEI probabilities (Supplementary Fig. S14). We find that stay events cluster around post-event attention strength peaks (time from first attention peak: $median_{1st\ stay} = -0.08$ cycles, $range_{25-75\%,1st\ stay} = -0.13 - -0.01$ cycles; $median_{2nd\ stay} = 0.97$ cycles, $range_{25-75\%,2nd\ stay} = 0.79 - 1.04$), while switch events predominantly occurred half a cycle earlier during attention strength troughs (time from first attention peak: $median_{1st\ switch} = -0.51$ cycles, $range_{25-75\%,1st\ switch} = -0.54 - -0.47$ cycles; $median_{2nd\ switch} = 0.48$ cycles, $range_{25-75\%,2nd\ switch} = 0.40 - 0.57$). For each participant, the cycle length was defined from the post-event fluctuation frequency (Fig. 7d).

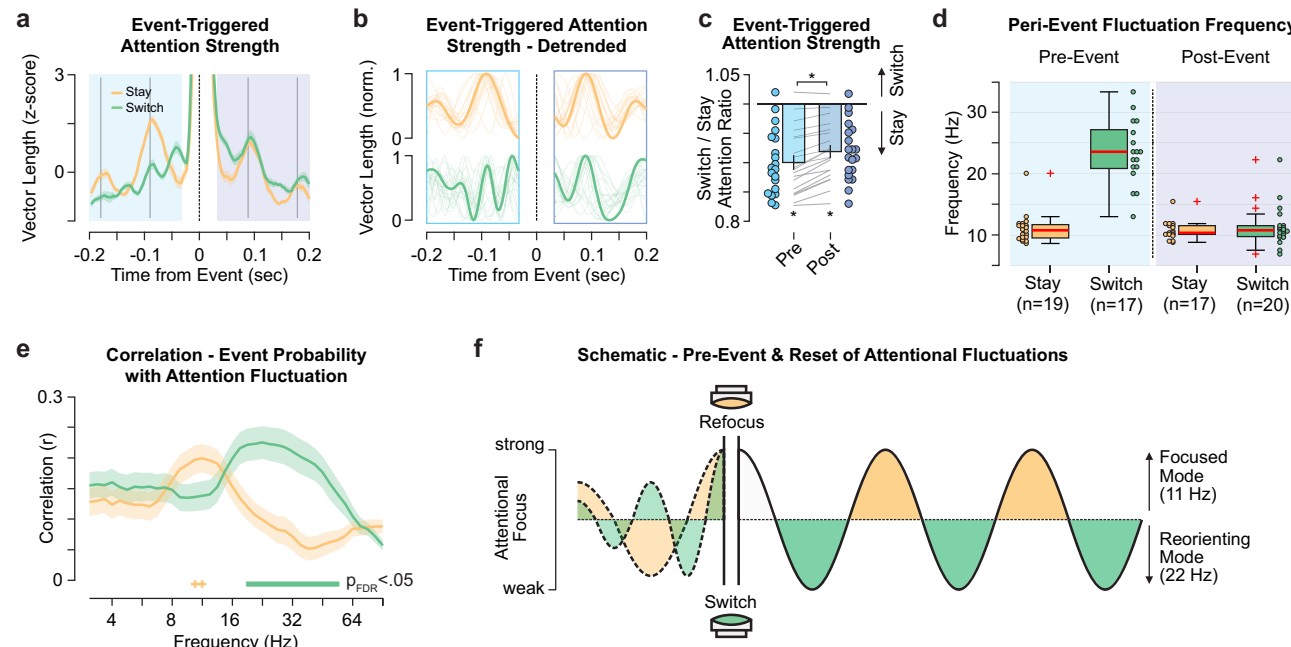

**Fig. 7 | Peri-event fluctuations of attention strength. a** Attentional event-triggered average of the z-scored attention strength, i.e., vector length. The colored lines and shaded area indicate the average and SEM over events and participants ($n_{participant}$ = 20) separately for stay (orange) and switch events (green). Pre- and post-event epochs are highlighted in blue and purple. Gray lines display full 11 Hz cycles from the event. **b** The same data as in (**a**) detrended individually pre- (− 0.2 s to − 0.05 s) and post-event (0.05 s to 0.2 s). **c** Peri-event attention strength difference between switch and stay, displayed as the ratio of average activity over a time window from − 0.15 to − 0.05 (Pre, blue) and 0.05 to 0.15 (Post, purple). Bars and vertical lines denote the mean and SEM over participants. Colored dots and gray lines indicate the single-participant ratio ($n$ = 20) and the difference from pre- to post-event, respectively. Asterisks indicate significant differences (two-sided $t$ test(19), uncorrected $p$-values: $p_{pre}$ = 1.43*10$^{-7}$, $p_{post}$ = 5.22*10$^{-7}$, $p_{post-pre}$ = 8.31*10$^{-7}$).

**d** Dominant peri-event fluctuation frequencies based on (**b**). *N* denotes the number of participants with two peaks per interval (see Methods). Box plots display the 25%–75% percentile range (box), median (red line), whiskers maximally up to 1.5 times the inter-quartile range and outliers (red plus). **e** Correlation between the attentional event probability (Supplementary Fig. S11) and the amplitude of the attentional fluctuations (Fig. 3e) over the stimulus presentation period. Lines depict the average, shaded areas the SEM over participants. Thick lines indicate significant differences (two-sided $t$ test(19), $p_{FDR}$ < 0.05; range FDR-corrected $p_{17.4-59Hz}$ = [0.0003; 0.0447]) and crosses a statistical trend for significance (two-sided $t$ test(19), uncorrected $p_{10.3Hz}$ = 0.0692, $p_{11.3Hz}$ = 0.0686). **f** Schematic - pre-event activity dissociation and the reset of attentional fluctuations at each event. The schematic was created using Affinity Designer (Version 1.10.8, Serif Europe Ltd).

Given the presence of a higher frequency attentional rhythm prior to switching, we next asked whether faster or slower attentional fluctuations predict the occurrence of event types[12,14]. We thus correlated the attention strength fluctuation amplitude in each frequency range (Fig. 3e) with the trial-averaged event probability (Supplementary Fig. S11). Stronger 9–11 Hz attention modulation trended towards more stay events (Fig. 7e, paired $t_{19}$ > 1.92, $p_{FDR}$ < 0.08) and stronger 19–40 Hz modulations significantly predicted more switch events (Fig. 7e, paired $t_{19}$ < − 3.42, $p_{FDR}$ < 0.05). Hence, faster attention rhythms corresponded to an increased probability to re-allocate attention (switching).

Taken together, our results suggest that switching the attention locus exhibited a unique relation to attentional fluctuations (Fig. 7f), resetting the frequency, as well as distinct spectral fingerprints, particularly highlighting higher frequencies (>11 Hz). Switch events appear to represent a neurophysiological mechanism of reallocating attention between options dissociable from the (re-)focus on single alternatives (stay). These two distinct modes of attention appear to be organized through a common oscillation: With attentional switches embedded in down-phases of the attention oscillation, the conflict between prolonged focusing on one decision alternative and the need to shift attention between alternatives to compare them can be resolved temporally.

## Eye movements dynamics are dissociable from covert attention

Previous research has shown that overt, i.e., eye-movements, and covert attention are linked. Particularly, it has been proposed that

(micro)saccades are overt manifestations of covert attentional processing[50–52]. Similarly, the recent literature on the interplay between attention and decision-making has predominantly focused on overt attention via saccadic eye movements[9,13]. However, new findings indicate that overt and covert attention allocation is dissociable and may offer distinct insights into information sampling and processing[24,27,33,34]. We thus assessed if and how the rhythmic fluctuations of covert attention relate to, or potentially originate from, the activity of the oculomotor system. In other words, do eye movements show compatible dynamics as covert shifts of attention?

We applied eye-tracking and extracted (micro)saccadic eye-movements (MS), to evaluate the relationship of overt gaze to covert attention. The trial-averaged MS rate displayed pronounced variation with the trial structure (Fig. 8a), increasing after stimulus- and framing cue onset and following a - 0.2 s suppression period. To characterize the temporal structure of (micro)saccades, we computed the inter-saccade-intervals (ISI) over the stimulus period. ISI histograms displayed a single peak at around 0.25 s (Fig. 8b). The oculomotor first-peak latencies were thus considerably longer in comparison to those of covert attention (Fig. 5e; $p < 10^{-10}$). Moreover, due to the lack of additional local maxima, there is no evidence of persistent rhythmicity in the (micro)saccade distribution beyond isolated serial saccades (i.e., absence of first-to-second mode latencies).

To directly assess the relationship between oculomotor activity and covert attentional events, we computed the (attentional) event-triggered MS rate. We identified a prominent increase around -0.15 s, exclusively before switching events (Fig. 8c; $p_{FDR}$ < 0.05). Conversely,

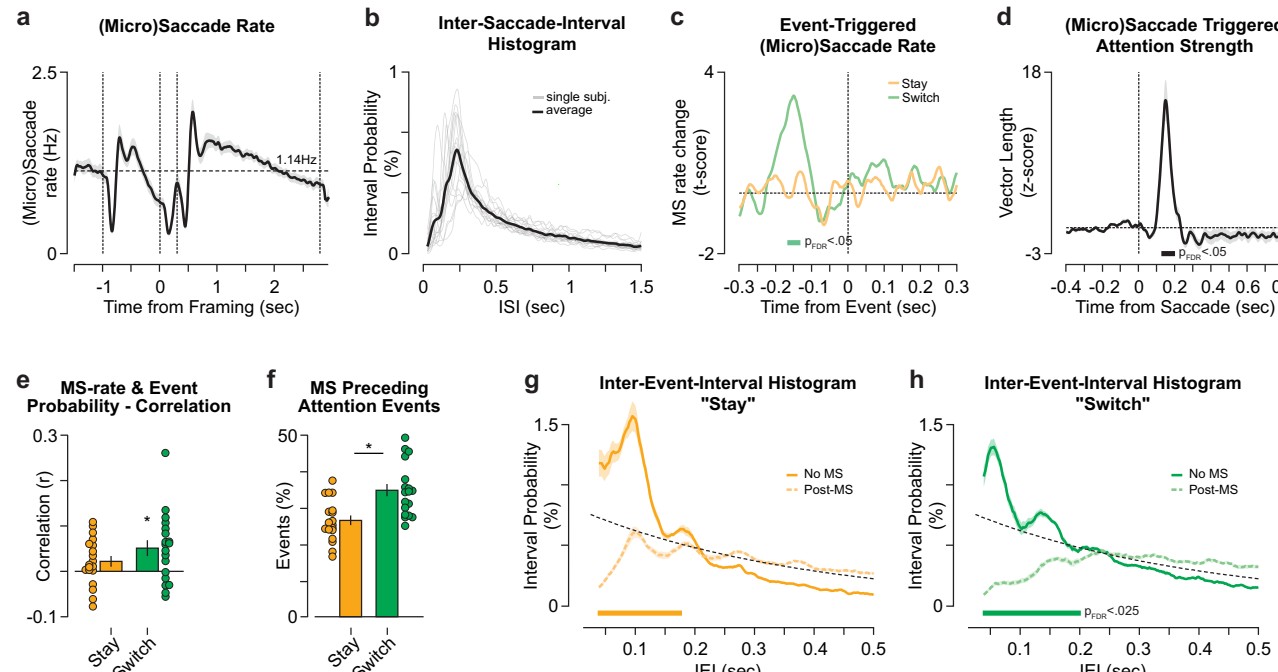

**Fig. 8 | Relationship between covert attentional fluctuations and oculomotor activity. a** Average (micro)saccade (MS) rate during task execution. The thick line indicates the average over participants, and the shaded area the SEM. Vertical dashed lines indicate the stimulus onset, framing cue onset and offset, as well as the stimulus offset. The horizontal line indicates the average MS rate over the stimulus on period (1.14 Hz). **b** Inter-saccade-interval histograms for micro(saccades). Thick and thin lines display the group-average and single-participant ISI histograms ($n_{participant} = 19$), respectively. **c** T-scores of oculomotor event rates, i.e., (micro) saccade, pre- and post-attentional saccades (switch in green, stay orange) tested against shuffled events (two-sided $t$ test per timepoint). The thick colored bar denotes statistical significance (df = 18, $p_{FDR} < 0.05$; FDR-corrected over time). **d** Attention strength around (micro)saccade onset. The data is z-scored based on the 0.4 s prior to MS. The black line indicates the average over participants, and the shaded area the SEM. The thick bar denotes statistical significance from zero (two-sided $t$ test(18) per timepoint, $p_{FDR} < 0.05$, FDR-corrected over time). **e** The correlation of the attentional event probability (Supplementary Fig. S11) with the trial-averaged (micro)saccade rate (MS-rate) **a** for each participant ($n_{participant} = 19$). The bars and vertical lines display the average and SEM over participants, respectively. The colored dots represent the single participant Pearson correlation coefficients (color-coded as in **c**). Asterisk denotes significant correlations (two-sided $t$ test(18) for $|r| > 0$, uncorrected p = 0.0047). **f** Proportion of attentional events preceded by (micro)saccades (MS). Color-coded as above. Bars, vertical black lines and colored dots display the mean and SEM over participants as well as single participant data ($n_{participant} = 19$). Asterisk denotes significant difference between switch and stay (two-sided $t$ test(18), uncorrected $p = 2.31 \times 10^{-10}$). **g**, **h** Mean inter-event-interval histograms for stay (**g**) and switch (**h**) attentional events split into events with (dashed) and without a (micro)saccade (solid line) between successive events. Colored lines and shaded areas denote mean and SEM over participants ($n_{participant} = 19$). Color-coded as above. The thick colored bar denotes statistical significance (one-sided $t$ test(18) for No MS > post-MS; $p_{FDR} < 0.025$, FDR-corrected over time).

attention strength was strongly increased following a (micro)saccade at around 0.17 s post-saccade (Fig. 8d). Correlating the average MS rate with the corresponding attentional event probability within each participant (Fig. 8e), we found that attention switches are positively correlated with the MS rate ($r_{mean} = 0.06$; $t_{18} = 3.23$; $p = 0.005$). Further, we found that (micro)saccades preceded switching events (mean = 35%, range = 25 – 50%) more often than staying events (mean = 26%, range = 17 – 38%; $t_{18} = -13.2$, $p < 10^{-9}$). Taken together, oculomotor activity appears to play a prominent role in switching attention between decision alternatives but less when staying focused on the same alternative.

We next evaluated the effect of oculomotor activity on the ongoing attention oscillation. We examined if a (micro)saccade occurring between two attentional events affects the respective interval lengths. We found that on average, (micro)saccades decrease the probability for short-latency intervals up to 0.2 s both for staying and switching events (Fig. 8g, h). Although, overt gaze displacements appear to delay the upcoming attentional event, the rhythmicity persisted (Fig. 8g, h): We identified post-saccadic rhythmicity (first-to-second mode latency) in 15 out of 19 participants for both event types with no detectable change in oscillation frequency relative to events without (micro)saccades (paired $t_{14,stay} < 1.8$, $p_{stay} = 0.09$; paired $t_{14,switch} < 1.1$, $p_{switch} = 0.29$).

In conclusion, oculomotor activity interacted with the way attention switched between decision alternatives. However, overt gaze cannot explain the dynamic aspects of attentional allocation. Thus, the analyses reported in this section indicate that the presented insights on covert attention dynamics cannot readily be inferred from monitoring eye position.

## Discussion

Attention likely plays a pivotal role in complex, naturalistic decision-making, steering information sampling in order to assess and compare alternatives. However, little has been known about the role of internal attentional dynamics in decision formation. By means of high-precision, continuous tracking of the locus and strength of covert attention in the human brain, we here uncovered rich spontaneous attentional dynamics that govern information sampling during multi-alternative decisions. We established that the strength of attention fluctuated around 11 Hz frequency and that peaks of this attentional oscillation were associated with an increased tendency to maintain focus on a single decision alternative indicative of focal processing. By contrast, troughs of the attentional oscillation were associated with redirecting attention between decision alternatives. Redirecting attention was further preceded by faster attentional fluctuations and resetting the ongoing attentional oscillation.

Recent literature has quantified generic temporal fluctuations in behavioral performance[17,19,22,28,48] and/or in cortical population activity[15,17,18,23,26], without explicating the information content of the latter. By contrast, we here constructed content-specific decoders from patterns of cortical population activity (trained on independent localizer data). This enabled us to track the strength and locus of attention as the competition between visual representations during decision formation, i.e., the expression of attention on early visual areas[10,12,14,39,43,44]. Importantly, this approach enabled us to discover attentional dynamics in a data-driven fashion, without relying on prior assumptions about rhythmicity or temporal structure. The framing manipulation in our decision task further decoupled the (bottom-up) stimulus intensity from the (top-down) task-relevance of the decision alternatives. Further, apart from a ~0.25 s stimulus onset response we did not find evidence that the physical contrast was affecting the decoding between framing conditions (Supplementary Fig. S5). We can thus conclude that our findings, and particularly the dominant attention oscillation frequency around 11 Hz, were not solely driven by bottom-up sensory cortical responses (Supplementary Figs. S3, S4, S10). However, our analyses cannot fully disentangle bottom-up and top-down attentional control during attention allocation[53].

Rhythmic attention has been associated with the 4–8 Hz range[14,19] but also found at higher sampling rates (7–12 Hz)[16,32]. The exact frequency channel, here 11 Hz, might depend on the cognitive domain and also the task and stimulus complexity[16,27,54–56]. For example, compatible sampling rhythms have been reported in tasks with extensive processing demands of the attended stimuli[54,55], or when decoding information from peripherally presented stimuli[16,57]. However, further research is needed to explicate the influence of external (e.g., binary vs. ternary decisions) and internal factors (e.g., neuromodulation) on the sampling speed.

Our findings provide a window into the computational and neural basis of decision-making by alluding to the mechanisms underlying key decision computations. In decision-theoretic models, central computations such as valuation and comparison are conceptualized in idealized ways and assumed to occur at distinct stages of the decision process[58]. Within our proposed framework, re-focusing attention on a previously attended alternative corresponds to a processing mode that could implement valuation. Similarly, switching attention between alternatives may underpin comparative processing. We show that although tendencies to re-focus (valuation) and switch (comparison) attention occur at different times (Supplementary Figs. S11, S14)[20], they are temporally coordinated within a single cycle of attentional oscillations. Oscillating attentional strength, a feature potentially established through cortico-thalamic connections[26,59,60] or visual cortex center-surround properties[24], resolves the competition between sampling information of one alternative and comparing between decision alternatives[12,14,19]. Thus, rhythmicity (identified at the level of single participants) is the mechanism that allows key computations to be performed without competing for the same cognitive resources or substrates[12,14,20].

One central finding is that stronger attentional rhythmicity right before the response is associated with more accurate decisions (Fig. 4), potentially due to enhanced attentional gain[19]. At times of strong attentiveness participants were more engaged[31,46] with the task (Supplementary Fig. S8) and increased processing of the attended decision alternative (Fig. 5 and Supplementary Figs. S7, S13). Consequently, the attention-modulated input driving evidence accumulation is likely rhythmic as well[61]. Yet, while attentional rhythmicity appeared to be stable over time (Fig. 3e and Supplementary Fig. S15) its effect on overt behavior might vary with time (Fig. 4b and Supplementary Fig. S7). In other words, the effect attended information has on the upcoming decision does not only depend on whether[8,9] a decision alternative is sampled but also on downstream decision mechanisms, such as e.g., accumulation leak[62] or adaptive gain[63,64], that modulate incoming

information. This points to temporally dissociable, but interlinked, processes of attention and decision-making. Future research is needed to explicitly elucidate the temporal relationship between the two processes. Overall, integrating covert attention as a time-varying variable, both in locus as well as strength, into existing models[8,9] of evidence accumulation can improve our understanding of dynamic decision making as well as the suboptimalities characterizing behavior in complex, multiattribute contexts[9,63,65].

Another prominent aspect of our findings is that, unlike re-focusing, switching events are accompanied by changes in neuronal processing. The frequency of the attentional oscillation is higher prior to switching (Fig. 7), indicating dissociable spectral features between attention functions. 'In-depth evaluation' (stay, ~11 Hz) and the 'comparison of alternatives' (switch, ~22 Hz) might for example, be coordinated through cross-frequency interactions[12,14,19]. With a switch, the frequency of the attention oscillation is reset to ~11 Hz, signaling a shift from a disengaged to a focused mode of processing. This reset of the attentional rhythm speaks for transient interactions between two functions of attention, i.e., switching and orienting vs. staying and alertness[10], and is further supported by their corresponding neural substrates (Supplementary Figs. S8, S13). The reset might serve two purposes. First, it could correspond to an internal go cue, indicating readiness of the system to change[11] and briefly enhancing processing power[66]. Second, it might temporally (re)align processing at various stages of the functional hierarchy[17,19,67] or multiplexed information channels[68]. In the context of decision theory, an attentional reset and its accompanying neural dynamics could be the cornerstone neuro-computational mechanism underlying binary comparisons[1,66,69]. However, further research is needed to better understand the functional role that switching resets have during decision-making.

Finally, our findings suggest that the fast dynamics of sampling information from various decision alternatives are distinct from oculomotor activity. In principle, oculomotor responses, i.e., (micro)saccades, can guide subsequent neuronal processing[10,51,66] by triggering shifts of attention locus. Indeed, we showed that after fixational eye movements and microsaccades, the probability to switch attention between alternatives increased (Fig. 8). Overt gaze displacements yielded a brief post-saccadic suppression of attentional event probability followed by strengthened visual representations of the focused alternative. However, not every covert shift of attention is accompanied by compatible eye-movements[24,27,33,34,49]. Particularly, oculomotor activity itself did not display rhythmicity, inter-saccadic interval maxima occurred later than attentional modes, and (micro)saccades displayed a relationship to switching but not to re-focusing attentional events. Lastly, the rhythmicity within the attentional cycle was not affected by overt gaze. Although recent experimental and modeling studies have emphasized the causal role of overt fixations on choice[13,36] and intrinsic value representations[9], our results paint a more nuanced picture. We show that the relationship between covert information sampling and eye movements might be one-way: microsaccades can elicit covert reorienting, but covert reorienting is not necessarily associated with microsaccades. Thus, where one is fixating and what is intrinsically processed are not interchangeable units of attention and neuronal processing. Our study offers a methodological framework that dissociates the role that latent and overt attentional components play on decision formation.

In conclusion, our findings highlight rhythmic sampling as a general neurophysiological mechanism that resolves conflicts between neuronal processing stages in visual search, target detection, perception, working memory[70] and, here, multi-alternative decision-making. Focused and comparative sampling of information display distinct spectral profiles but are coordinated within a common oscillatory cycle of about 0.1 seconds, and operate distinctly from overt oculomotor markers. Similarly, decision-relevant quantities such as the value representations of decision alternatives likely fluctuate over

time. These dynamics challenge extant sequential sampling models of choice, which assume stable and parallel throughput of value information to downstream accumulator representations. These insights open new theoretical avenues, highlighting the role of rhythmic information processing in shaping key decision computations.

# Methods

## Participants and experimental procedure

Twenty right-handed participants (9 female, $age_{mean} = 28.05$ years $\pm age_{STD}$ 4.6 years) participated in a three-alternative decision-making task with variable task framing. The experiment was approved by the ethics committee of the Hamburg Medical Association and conducted in accordance with the Declaration of Helsinki. All participants gave their written informed consent and received monetary compensation for their participation.

**Decision-making task.** During central fixation, three Gabor patches of varying contrast levels and orientations appeared simultaneously on screen. The stimuli were presented peripherally (2.2° eccentricity, 1.6° stimulus diameter) at the top and 120° left and right of it. The three stimuli were drawn without replacement from a set of five linearly spaced contrast levels (10 possible combinations). We derived the distance between neighboring contrast levels via a staircase procedure individually within each participant. The patch orientations were drawn from a set of three equally distant orientations with either all three being identical or different and pooled our analyses over orientation conditions. After 1 second into the stimulus presentation a central framing cue (0.3 sec.) indicated the target contrast, either the highest ("Hig") or lowest ("Low"). The high contrast stimulus has the highest value in high framing trials, while it has the lowest value in low framing trials. Framing the contrast is analogous to for example asking a participant to choose from a set of food items the one they would prefer[71] (high framing) or the food they want to avoid (low framing). We randomized the task framing between trials. After the framing cue offset the central fixation cross re-appeared and initiated a fixed decision phase (2.5 sec.). The three Gabor stimuli were visible and unaltered all throughout the 3.8 s stimulus period. After the stimuli turned off the fixation cross rotated 45° and cued the response. Participants had 2 s to indicate their decision with a left thumb, right thumb or foot button press for the left, right or top alternative, respectively. We randomized the foot (left or right) between participants. To visually guide the participants the entire stimulus array was displayed with a 6° clockwise or counter-clockwise tilt when participants were using their right or left foot, respectively. The foot pedal was mounted on Styrofoam. We adjusted the height and location of the pedal for each participant such that the position was comfortable and a response could be triggered with a small flexion of the porcellus fori. With the individual adjustments to the foot pedal placement, we tried to make the motions between effectors as compatible as possible. Median response times (RT) were slightly but not significantly slower for foot responses ($RT_{foot} = 0.46$ sec $\pm 0.18$ sec; $RT_{left} = 0.40$ sec $\pm 0.18$ sec; $RT_{right} = 0.39$ sec $\pm 0.16$ sec, each mean $\pm$ standard deviation; paired sign test $p_{foot,left} = 0.06$; $p_{foot,right} = 0.08$).

Task framing, contrast levels, Gabor orientation and stimulus locations were fully randomized, yielding a total of 1080 unique trials. The trials were blocked into random sets of 120 trials, and each participant concluded two full randomization runs (18 blocks).

**Retinotopic stimulus localizer.** Prior to each block of the decision-making task, the participants concluded a localizer block. Each block consisted of 90 consecutive 0.35-second presentations of single Gabor stimuli at full contrast with 0.5 s inter-trial intervals. Each stimulus was presented with the same size and eccentricity as during the decision-making task. We instructed the participants to ignore the peripheral

stimuli and fixate at the center. The participants' task was to detect a flickering of the central fixation cross that could occur randomly at each stimulus, in total 9 times per block, and indicate the detection with a finger button press. We excluded trials with flicker presentation from further analyses. In each block, we fully randomized the orientation and the position over the three locations. Each unique stimulus combination was repeated ten times, paired once with a central fixation flickering.

## Functional data acquisition and preprocessing

**Data acquisition.** During the experiment, the participants were sitting upright in a whole-head magnetoencephalography scanner (MEG; 275 axial gradiometer sensors, CTF Systems Inc.) in a magnetically shielded room. The data was acquired at a sampling rate of 1200 Hz. Each participant's head position was tracked online using three head localization coils (nasion, left/right preauricular points). *n addition*, we recorded a 2-channel bipolar electro-oculogram (horizontal and vertical EOG) and a single-channel bipolar electro-cardiogram (ECG) to control for cardiovascular and ocular signal artifacts. Eye movements and pupil diameter were tracked with an MEG-compatible Eyelink 1000 system (SR Research). We further collected T1-weighted structural magnetic resonance images (MRI; sagittal MP-RAGE) to reconstruct individualized high-resolution head models.

**Preprocessing.** We band-pass filtered the continuous MEG data segments between 0.2 and 200 Hz (4th order Butterworth filter) and downsampled it to 400 Hz. Next, we filtered the line noise with a notch filter at 50 Hz and its first 6 harmonics (1 Hz stop-band filter width). We implemented a two-stage procedure for artifact rejection[72,73]: First, we inspected all data segments for technical- (i.e., SQUID-jumps, passing cars), muscle- and eye-blink artifacts and excluded corresponding time intervals from further analyses. Second, we filtered the data into two distinct frequency ranges, a low- from 0.2 to 30 Hz and a high-range from 30 to 200 Hz, with a 4th order Butterworth filter. We computed independent component analyses (ICA) on both frequency ranges separately. This approach takes advantage of the intrinsic properties of different artifact types. On the one hand, eye-blinks, eye-movements, passing cars and cardiovascular artifacts display predominantly low-frequency features. On the other hand, muscle activity shows the strongest effects in higher frequencies. We visually inspected the independent components, rejected artifact components based on their topology, power spectra and time-courses[72] and subsequently recombined both frequency bands.

## Spectral analysis and source reconstruction

**Spectral analysis.** We estimated time-frequency representations of the time-domain MEG signals using Morlet's wavelets[73]. We set the wavelet bandwidth at 0.5 octaves ($f/\sigma_f = 5.83$, kernel width covered $\pm \sigma_t$; $\sigma_f$ and $\sigma_t$ corresponds to standard deviation in frequency and time domain) and derived complex spectral estimates for frequencies between 2 to 90.5 Hz in quarter octave steps ($2^1$ to $2^{6.5}$ Hz).

**Source projection.** Based on each participant's individual structural MRI we generated single-shell boundary-element method models[74] (BEM) and computed the physical forward model (leadfields) for 457 equally spaced cortical sources[75] (~1.2 cm distance, at 0.7 cm depth below pial surface). Sensor locations, source and leadfield positions were co-registered to the common MNI-space using the position of the three head-localization coils. We applied dynamic imaging of coherent sources[76] (DICS) linear beamforming to estimate source-level neuronal activity. DICS is a spatial filtering approach that reconstructs activity at the source of interest with unit gain while at the same time maximally suppressing signal contributions from other sources. For the visualization of neuronal activity topographies, we used the FieldTrip toolbox[74] (version; sensor level) and custom code[77] (source level).

## Encoding the focus of covert attention

To reconstruct the focus of covert attention during free information sampling of multiple stimuli, we implemented an inverted encoding model[40]. The modeled activity hereby displays momentary changes of the visual representations of the presented stimuli (Fig. 2). The model assumes that the sensor-level MEG activity can be described as a weighted sum of distinct abstract neuronal populations, i.e., information channels, with individual tuning properties to the angular positions plus noise $N$.

$$\mathbf{MEG}_{\text{Loc}} = \mathbf{W}\mathbf{C}_{\text{Loc}} + \mathbf{N} \qquad (1)$$

We trained the encoding model on the retinotopic stimulus localizer task $\mathbf{MEG}_{\text{Loc}}$. $\mathbf{C}_{\text{Loc}}$ was a design matrix describing a set of basis functions, i.e., the tuning curves, for each information channel (Fig. 2a) and W described the weight matrix specifying the relationship between the information channels and the neuronal data. Each information channel was modeled as a half-wave rectified sinusoidal filter $f$ with a fifth power exponent ($\alpha = 5$). We derived the basis functions $f$ for the design matrix as

$$f = |\cos^{\alpha}[0.5(\theta - \mu)\pi/180]| \qquad (2)$$

Here, $\theta$ is a continuous variable indicating the angular position relative to central fixation, and $\mu$ the preferred angle of each filter ($\theta$ and $\mu$ are in degrees, with 0° pointing upward along the vertical midline and angular value increasing counter-clockwise e.g., 90° pointing leftward). $\alpha$ was the power exponent set to 5. The target stimuli were set to $\mu = 0°$, 120° and 240° for top, left and right, respectively. During the localizer task only one stimulus out of the full set has been visible on screen. Hence, we trained the model during the localizer, as both $\mathbf{MEG}_{\text{Loc}}$ and $\mathbf{C}_{\text{Loc}}$ were known variables. We estimated the weight matrix W (sensors x information channels) as follows

$$\mathbf{w}_k = \mathbf{MEG}_{\text{Loc}}\mathbf{c}_{\text{Loc},k}^{\text{T}}(\mathbf{c}_{\text{Loc},k}\mathbf{c}_{\text{Loc},k}^{\text{T}})^{-1} \qquad (3)$$

Where $\mathbf{w}_k$ denotes the weights of the $k$th information channel $\mathbf{c}_{\text{Loc},k}$, i.e., a single tuning curve. As neighboring MEG channels are not independent estimates of neuronal activity due to volume conduction, we further included multivariate noise normalization to improve the pattern reliability. Thus, we transformed each $\mathbf{w}_k$ into the pre-whitened weighting vector $\mathbf{g}_k$

$$\mathbf{g}_k = \mathbf{E}_k^{-1}\mathbf{w}_k(\mathbf{w}_k^{\text{T}}\mathbf{E}_k^{-1}\mathbf{w}_k)^{-1} \qquad (4)$$

Where $\mathbf{E}$ is the noise covariance matrix for channel $k$ using the channels noise term as

$$\boldsymbol{\varepsilon}_k = \mathbf{MEG}_{\text{Loc}} - \mathbf{w}_k\mathbf{c}_{\text{Loc},k} \qquad (5)$$

$\mathbf{E}$ was regularized by shrinking it towards the diagonal matrix, using the optimal shrinkage factor, i.e., minimizing the expected squared loss of the resultant covariance estimator. With the optimized weight-matrix G we can estimate the latent visual representation of the stimuli during the multialternative decision-making task (Fig. 2b). Therefore, we invert the encoding model

$$\mathbf{C}_{\text{MDM}} = \mathbf{G}^{\text{T}}\mathbf{MEG}_{\text{MDM}} \qquad (6)$$

$\mathbf{C}_{\text{MDM}}$ comprises the estimated response as the representational activity distribution over angular positions for each time point during the task. The strength of the representation indexes the covert visual attention towards each target on screen.

Lastly, we reconstructed the momentary spatial focus as the vector sum over the three estimated stimulus activities. The vector orientations were set to 0°, 120° and 240° for the top, left and right stimulus, respectively and the corresponding length at each orientation from the channel response $\mathbf{C}_{\text{MDM}}$. This reconstructed attentional vector visualizes two main characteristics of spatial attention, namely the overall attention paid at each time point and from which target information is sampled. We considered the attentional vector length as a proxy of overall attention and the maximum of the angular similarity to the displayed alternatives as the most likely target of covert spatial attention.

We trained the model for different time points during the retinotopic stimulus localizer task with respect to stimulus onset. To achieve the best decoding performance, we selected the optimal training epoch using cross-validation (leave-one-block-out, 18 blocks per participant) within the localizer task. For the training model we excluded localizer trials that displayed (micro-)saccadic activity (see below) within the first 200 msec after stimulus presentation. On average we excluded 3% of all localizer trials per participant (mean = 3.03%, median = 1.82% range = 0.50% - 10.68%). We found decoding strength to peak between 140 to 170 msec post stimulus onset (Supplementary Fig. S1) and averaged the MEG signal over this period to construct the decoding model for the application on the $\mathbf{MEG}_{\text{MDM}}$ data. Important for the interpretation of the IEM output is whether or not a stimulus is on screen. In the pre-stimulus phase the Attentional Vector might vary due to for example random fluctuations of ongoing neuronal activity. On the other hand, directly after the stimulus offset in the post-stimulus period participants might have still been remembering the options or display extended stimulus processing from memory. We thus centered our analyses on the stimulus presentation period.

## Spectral analysis of attention strength

We computed time-frequency resolved estimates of the attentional vector length to assess the spectral dynamics of attention strength. We band-pass filtered (4th order Butterworth filter) the trialed vector length ($f_{\text{sample}} = 400\,\text{Hz}$, from $-1.5\,\text{s}$ relative to stimulus onset to 2 s after stimulus offset) to center frequencies between 2.8 Hz to 90.5 Hz in quarter octave steps ($2^{1.5}$ to $2^{6.5}$). The pass-band covered the center frequency plus/minus half an octave, e.g., for $2^{1.5}$ Hz center frequency we filtered the data between $2^1$ and $2^2$ Hz. We estimated the frequency-dependent attention strength as the absolute value of the complex Hilbert transform from the time-domain vector length signal. To exclude edge artifacts, we cut two cycles of the center frequency from the beginning and end of each trial. We visualized the data between $-1.5$ to $+3$ seconds relative to the framing cue onset. We applied the same procedure for the reconstructed stimulus activities at each location (Supplementary Figs. S3, S4).

For the analysis of attention strength fluctuations as a function of correct versus error trials we stratified the included trials. Hereby, we matched the number of trials included from each of the 10 possible contrast combinations per framing condition. For all participants we identified more correct choices than error choices in all stimulus combinations. We thus randomly subsampled one hundred times from all correct trials according to the stratification assumptions and averaged the spectral amplitude and the vector length over repetitions before comparing against error trials (Fig. 4).

## Defining attentional events

We estimated the momentary focus of spatial attention via the angular similarity of the attentional vector to the three stimuli on screen. At each point in time the attentional vector can display the focus to be close to one or in between targets. The maximum of the angular similarity (MAS) between the Attentional Vector and the three alternatives is an estimate of how focused a participant is. If MAS equals 1, i.e., the cosine between two angles, then the participant is focused solely on one alternative at a given time point. The minimum MAS is

0.5 for the case when the Attentional Vector points exactly in between two alternatives. To investigate the dynamics of information sampling, i.e., when one alternative is selectively focused, we thresholded the maximum angular similarity at the participant-specific 90th percentile over the entire dataset and binarized the time-courses accordingly. However, as the vector orients in circular space and even under completely random conditions will point towards one of the three stimuli at times, we added a second threshold on the vector length (90th percentile within a trial). Thus, we recorded attentional events when both thresholds were crossed at the same time. In other words, we traced points in time when participants were both attentive and locked on one stimulus. We recorded the first time point of passing both thresholds as an event onset and excluded events occurring within 0.025 sec after the previous onset. We further tested the behavioral relevance of event occurrences and event targets on the reported choice using logistic regression. Within each participant, we predicted the categorical choice (H, M, L individually) from the number of events at each decision alternative (stay-H, stay-M, stay-L, switch-H, switch-M, switch-L). We computed $t$ tests against zero over participants and report the related t-scores and $p$-values.

**Event dynamics.** To assess the dynamics of these attentional saccades, we investigated the intervals between subsequent events. Here, we recorded the time from event $e$ onset to the onset of event $e + 1$ within a trial and generated inter-event-interval (IEI) histograms over all events and trials within a participant. We excluded intervals reaching over trials. The histograms were computed over a median of 17,696 events per participant (range 12,377 to 28,909; Supplementary Fig. S6). We further generated Null-models for inference statistical testing using randomization. Hereby, we randomized the trial each event was belonging to while keeping its position within the trial. In other words, when we detected for example 10 events at time point $t$ scattered over 2160 trials, we randomly reassigned trial identities to these events (without replacement) and proceeded in the same way with time point $t + 1$. We computed the IEI histogram after every randomization run. With this approach we can randomize event relationships without tempering with the task-dependent dynamics. We repeated the randomization step 1000 times for each participant.

**Event sequences.** We further tracked first-order event sequences. An event can occur either at the same stimulus location as the one before or a different location, i.e., a participant either refocuses (stay) or reorients (switch) their attention. We generated IEI histograms and the randomization for both event types separately and statistically tested interval probabilities against the mean of the randomized Null-models (one-sided paired $t$ test). Hereby, we obtained t-scores for each interval from 0.025 to 0.5 s (at $f_{sample} = 400$ Hz) for both stay and switch.

**Interval-probability fluctuations.** If we assume that attention is rhythmic, then attentional events would occur probabilistically at preferential phases within an underlying oscillation, and we would expect to observe distinct peaks in the IEI histograms: The first peak would display the preferential phase within the first oscillatory cycle, the second peak the second cycle, etc. Thus, we can estimate the prevalent attentional fluctuation frequency as the inverse of the latency between peaks. Therefore, we subtracted the participant-specific expected IEI histogram (Null-model) from the observed histogram. All non-significant values were set to zero according to random-effect statistics based on randomization (one-sided paired $t$ test against zero, $p < 0.01$; see also above). We smoothed the resulting truncated histograms twice with a 0.01 sec boxcar kernel, padding the edges to the first and last histogram value. Finally, we extracted the local maxima of the truncated and smoothed histograms. Overall, we identified one significant peak in 17 and 18 participants and at least two

peaks in 16 and 17 participants for the stay- and switch-IEI histograms, respectively. For further analyses, we defined the cycle length of the attention rhythm as the latency from the first to the second peak.

**Threshold assessment.** We further controlled for the effects of the applied threshold through a systematic assessment for a wide range of threshold choices. In theory, if the thresholds are too liberal, the analysis will include several random events converging the IEI histogram to an exponential decay. If the threshold is too high, there won't be enough events to have sufficient statistical power for the full IEI distribution. We varied the applied thresholds between the 50% and 95% percentile in 5% steps, as well as for the 97.5% and 99% percentile separately for the maximum angular similarity and the vector length (Supplementary Fig. S16). This approach yielded a total of 144 ($12 \times 12$) threshold combinations. We replicated our main findings (individual peaks in the IEI histogram) for a variety of threshold choices between the 80% and 95% percentile (Supplementary Fig. S16a, b). Importantly, the timing of histogram peaks appeared to be stable over thresholds. Further, we quantified the deviation of the histogram from an exponential decay, i.e., randomly distributed events, through the variability of interval probabilities further referred to as modulation intensity. Hereby, we first log-transformed the interval probabilities and linearly detrended the resulting values. Next, we smoothed these distributions with a 0.025 s (10 samples) box-car kernel twice and defined local maxima. Finally, we computed the modulation intensity as the standard deviation of the detrended histogram between the first and second maximum (Supplementary Fig. S16c, d). We found the strongest median modulation intensity for the highest vector length thresholds at over 95% with variably applicable thresholds for the MAS. Further increasing both thresholds beyond the 95% percentile yielded inconclusive modulation in several participants due to a lack of identifiable events (Supplementary Fig. S16e, f). We thus conclude that our main findings are threshold invariant within a sensible range between the 80% to 97.5% percentile with an optimal range approximately between the 90% and 95% percentile for both parameters. At the 90% percentile threshold for both measures, i.e., the applied threshold during the main text, thus granted both reasonably reliable modulation intensity and further a sufficient number of attentional events to increase statistical power for event-triggered analyses (Supplementary Fig. S16g, h). Finally, the optimal threshold parameters depend on the research question, the number and the length of the trials. It is thus pivotal to control these parameters for each new analysis. However, we must stress that optimizing the parameters does not evoke rhythmicity or temporal structure but can only reveal existing structure.

### Peri-event fluctuations of attention

To compare the dynamics leading up to attentional events we investigated the event-triggered fluctuations of attention from 0.2 s before to 0.2 s after each event. Here, we averaged the attention strength, i.e., attention vector length, and the attention focus, i.e., maximum angular similarity, over all events within each participant and z-scored the traces according to the distribution between $-0.2$ to $-0.03$ sec (pre-event period) and 0.03 to 0.2 sec (post-event period) before averaging over participants. We excluded the peri-event period ($\pm 0.03$ sec.) from normalization as the event itself, by definition the highest values, would skew the distribution. For further visualization, we de-trended and smoothed the event-triggered traces with a 0.025 sec boxcar kernel separately for the pre- and post-event period. Further, we used the de-trended and smoothed traces to define prominent peri-event fluctuation frequencies via the inverse of the average peak-to-peak latencies. We used the two peaks closest to the event, both in the pre- and post-event interval to define peri-event fluctuation frequencies. We could only identify a frequency when a participant would display at least two peaks within an interval.

## Pupillometry and oculomotor activity analysis

We tracked the eye-gaze and pupil size of the right eye at 1000 Hz sampling rate with an EyeLink 1000 Long Range Mount (SR Research, Osgoode, Canada). We calibrated and validated the eye-tracker at the start of each block. The system's spatial resolution was < 0.01° and the average position accuracy between 15 – 30 min arc. We down-sampled the sampling to 400 Hz compatible to the MEG data. Due to hardware issues, we had to exclude one participant from the analysis. Eye-tracking data was available for 19 out of 20 participants.

**Defining pupil responses.** We detected blinks according to the MEG preprocessing pipeline. Within each participant, we normalized the average pupil size between the 5% and 95% percentile. We derived the change in pupil diameter as the temporal derivative of the pupil size by subtracting adjacent frames.

**Microsaccade detection.** We converted the SR Research.edf files to Matlab through FieldTrip[74]. We normalized the gaze position through the data from the calibration. We defined (micro)saccades as brief events of high gaze velocity. We quantified gaze velocity as the absolute value of temporal derivate over x-y coordinates and smoothed with a 0.02 s Gaussian kernel. A saccade was recorded when the velocity exceeded 6 times the trial-based median[78]. To avoid counting the same event multiple times, we imposed a minimum delay of 0.025 s between gaze shifts. This saccade definition is agnostic to the length of the gaze displacement and can thus include both saccades and microsaccades (microsaccade$_{threshold}$: gaze displacement <1 deg visual angle[29]). We thus denote all oculomotor events as (micro)saccades. The (micro)saccade rate was generated as the saccade probability over trials at each sample multiplied by the sampling rate (400 Hz).

**Relating oculomotor events and attentional saccades.** We assessed the attentional event-triggered (micro)saccade rate in time-windows between − 0.3 to 0.3 s from attention event onset and averaged over events separately for switch and stay. We compared the average (micro)saccade rate within these windows against trial-shuffled attention-event definitions (see also above), i.e., the change in (micro)saccade rate. We further analyzed the (micro)saccade triggered attention strength, i.e., vector length, in a time-window between −0.4 to 0.8 s from (micro)saccade onset. Finally, we investigated the effect of (micro)saccades on the rhythmicity of attentional events by re-computing the inter-event-interval histograms separately for intervals containing or not containing a (micro)saccade.

## Neural correlates of covert attentional saccades

We computed the neuronal signal strength, i.e., the amplitude envelope, over the entire cortex to assess the relationship to attentional events (see also Spectral analysis and source reconstruction). According to our results (Figs. 3e and 5g), we focused the analysis on the alpha (11 Hz), beta (22 Hz) and gamma (40 Hz) frequency range. With respect to the literature, we further included the theta frequency range (4 Hz)[14,16,19]. Within each participant and source, we correlated the frequency-specific signal amplitude envelope with the event probability throughout the stimulus on period. For each event type, as well as the difference between them, we tested the correlation against zero over participants (two-sided $t$ test). We further computed the correlation between the neural signal envelopes and the corresponding band-limited attentional fluctuations.

**Event-triggered neural signals.** To assess the relationship between event-triggered attention- and neural modulations we averaged over signal amplitudes for switch and stay events separately in each cortical source for a time window of ± 0.3 sec relative to event onset. We z-scored the neuronal activity within each window prior to averaging over events and participants. For visualization purposes, we grouped a subset of cortical sources into four regions-of-interest[14,16,19] approximating: early visual areas including V1, medial parietal cortex, medial prefrontal cortex (mPFC) and the dorsolateral prefrontal cortex (dlPFC), including the frontal eye field (FEF). The average ROI activity was statistically compared between event types using a two-sided paired $t$ test.

## Reporting summary

Further information on research design is available in the Nature Portfolio Reporting Summary linked to this article.

## Data availability

The processed data generated in this study has been deposited in the osf.io database under accession code [https://osf.io/tf6bw/].

## Code availability

The code for reproducing the main results and figures has been deposited in the osf.io database under accession code [https://osf.io/tf6bw/].

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

## Acknowledgements

This research was supported by the EU Horizon 2020 Research and Innovation Program (ERC starting grant no. 802905, K.T.) and the BMBF (01GQ1907, T.H.D.)

## Author contributions

All authors contributed to the conceptualization and experimental protocol. M.S., Y.C., and M.T.M. collected the data. M.S. and Y.C. did the formal data analysis, and M.S. visualized the results. M.S., T.H.D., and K.T. wrote the initial draft, all authors contributed to editing the manuscript.

## Funding

## Competing interests

The authors declare no competing interests.
