## [Transparent Peer Review file · Nature Communications]

Rhythmic sampling of multiple decision alternatives in the human brain

Corresponding Author: Dr Marcus Siems

Version 0:

Reviewer comments:

Reviewer #1

(Remarks to the Author)

Siems et al. examine the temporal dynamics of visual attention and information sampling using MEG and neural decoding. In a 3AFC perceptual choice task, they report a rhythmic sampling of covert attention at 11 Hz, such that attention is focused at a single location or shifts to different alternatives during the peak and trough of the oscillation, respectively. They also report a reset mechanism for the attentional oscillation -- such that the sampling frequencies of staying and switching between alternatives align following a switching event. These findings shed light on possible mechanisms for serial information processing during multi-alternative decision making. They also explore the link between information sampling with other markers such as microsaccades and pupil diameter.

While the authors have performed many interesting analyses to support their claim, some points must be addressed to improve the clarity and robustness of the results:

Major points

Introduction (Pages 3-4):

There is an existing body of work on rhythmic sampling models of attention and decision making (e.g., Busch et al., 2009; vanRullen, 2016; Landau & Fries, 2012). The introduction appears under-motivated. It could benefit from expanding on previous studies and highlighting the existing knowledge and unknowns in the field, and how this study addresses these unknowns. For example, how have the "current neurophysiological approaches to decision-making ignored these internal sampling dynamics"? How exactly do the "non-invasive markers of attention, ..., display an incomplete picture of covert attention and its dynamics"? How does the authors' approach add to the existing studies?

Did the authors have specific hypotheses for this study? For example, the studies listed above report sampling frequencies at 4 Hz (for two alternatives) and 8 Hz (for a single stimulus). Did the authors hypothesize a similar sampling frequency? If not, why not? If yes, why choose a sampling frequency of 11 Hz instead? What might be the significance of 11 Hz sampling frequency in terms of the underlying neural mechanisms or physiology? This could be motivated in the introduction/discussion.

Results (Page 5) and Figure 1B:

How is accuracy being determined using decision value and contrast framing? I could not find it in the methods. The term "decision value" (H,M,L) in the figure legend (1B) is not defined in the text. Is this different from the term "option value" (used in the figure)? Overall, the accuracy results are unclear in Figure 1B.

Task Design (Figure 1A):

The stimulus is presented on screen for a total of 3.8 s (including 2.5 s of decision period). How do we account for fatigue later in this period, or for inhibition of return? Furthermore, selecting the stimulus with high or low contrast is a cognitively low-load task. Does the task demand 2.5 s of processing time (post-framing)? If the subjects make a decision early, it might be reasonable to assume that they would not keep sampling the space for the rest of the decision period. Why was a shorter presentation period not tested? Also, what was the purpose of the mid-contrast stimulus? Would the results be different if a two-alternative (high and low) decision making task were used?

Figure 2:

The encoding model was trained using the Sensory Localizer Task, which is a bottom-up paradigm. However, the model is then tested on the Main Task which is a top-down task. Can the bottom-up channel responses be translated to a top-down attention task? The figure could be improved (more details could be added) to describe the IEM process in a clearer manner. The color map in panel A (design matrix) for predicted response is missing/blank. It should be white-to-black for 0-1. Figure abbreviations and notations are not explained in the legend.

Results in section "Attention strength oscillates at 11 Hz" (Page 7-9):

Why is the attention strength vector length non-zero and negative during fixation and stimulus onset? Would we not expect a vector length of ~ 0 magnitude with gaze at the fixation cross during this period? Eye-movements in this period would be randomly directed and thus, should nullify each other to give a ~ 0 vector length. What does negative vector length (i.e., magnitude) mean, and with respect to what baseline is it negative?

Page-8:

"Evoked pupil responses resembled the trial-averaged attentional vector dynamics, albeit with more sluggish dynamics (Figure 3B; $r_{\text{pupil,veclen}} = 0.54$; $p = 0.007$)." Which time period has been used to calculate the correlation and why?

Page-9:

What is the justification for choosing 11 Hz for the alpha band and 40 Hz for the gamma band? Broadband frequency range is not consistently defined.

The authors estimated the correlation of neuronal activity with attentional fluctuations, at both alpha and gamma frequencies. Now, neuronal activity (Figure S2) shows a decrease in both alpha and gamma frequencies, while attentional strength (Figure 3E) shows an increase in the alpha band but a decrease in the gamma band. As such, the positive correlation reported (Figure 3H) for both alpha and gamma frequencies between these two signals is contradictory -- should the correlation for alpha frequency not be negative?

Some results are described in the text without any supporting evidence -- descriptive summary measures (mean, std) or statistical test results (p values) are not reported consistently or sufficiently for all results.

Figure 3:

Panel A - Legend mentions a shaded area but it is not visible in the figure.

Panel C - The single trial examples do not support the trends in panel A. Data is too noisy to show any distinct or identifiable patterns resembling the three peaks described in text and panel A.

Panel D - The significance/inference of this result is not clear. The amplitude spectra for different time points overlap with each other, and are not visible for comprehension. The figure could be improved for a clearer depiction of the shift in attention fluctuations. Crucially, the x-axis is plotted in log-scale and it is difficult to visually corroborate the peak at 11 Hz (same in Panel E).

Panel E - The authors report a frequency band for decrease (25-45 Hz), but a specific frequency of 11 Hz for increase in attention strength. If the singular report of 11 Hz (and not, say, a band from 8-16 Hz as shown in the figure) is an inference from the results, it is not supported by the reported results or figure. Could the authors report additional analyses to support their claim of fluctuation at 11 Hz? One simple addition would be a comparative report of fluctuations in the adjacent frequencies. If the focus on 11 Hz is hypothesis driven, the motivation should be clearly explained. Additionally, there is no change in the fluctuation in the 8-16 Hz band over time. Should we not expect a difference in attentional strength before and after framing in this frequency band? What can we infer about neural decisional processes from this? However, there is a change in the fluctuation for the 25-45 Hz (increase at around 1 second relative to Framing onset). This is not reported in text. But is there any significance of this result?

Panel G - We see a decrease in attention strength after 1 second. Is it due to (a) fatigue, (b) voluntary decrease in attention strength, or (c) shift from attention to decision making?

Results in the section "Attention strength fluctuations correlate with behavioral performance" and Figure 4:

Same issues as in Figure 3E. There is no evidence or values reported for the statement in Page-10, "However, this was a nonsignificant trend". Additionally, authors report a peak in attention strength fluctuation in a frequency range from 8-16 Hz, then state that "fluctuations around 11 Hz might be predictive..." Once again, the focus on 11 Hz is neither supported nor explained robustly.

Switch and Stay attentional events v/s High and Low contrast framing:

Did the authors analyze the switch and stay latencies for low and high framed trials separately? In high contrast framing trials, the salient stimulus is also the task-relevant stimulus, thus it is expected that the individual's attention will "stay" following Framing cue, and will last longer than switch. On the other hand, in low contrast framing trials, the salient stimulus (high contrast) is different from the framed, task-relevant stimulus (low contrast). This could lead to a faster switch from the bottom-up target (high contrast stimulus) to the top-down target (low contrast stimulus) after framing cue onset. While the study reports latencies between two events, could this framing effect (i.e., latency difference w.r.t. framing onset) contribute to the phase difference between switch and stay event frequencies?

Figure 7:

"N denotes the number of participants for which a sine-fit was possible".

The rejection criteria and sine-fitting is not explained in the Methods. What are the parameters of this sine fit? Were the fit parameters different for each subject, and in turn, was the rejection subjective? What is the justification for rejecting a

participant? Furthermore, it is difficult to track which analyses are for paired data and which are not (Figures 5 and 7). What is the justification for the choice of paired or unpaired tests across analyses?

Results in the section “Brief attentional saccades impact the ongoing attention strength fluctuation” (Page 16)

“We would expect that events, ..., are falling out from the ongoing 11 Hz oscillation.”

It might be recommended to avoid using relatively colloquial terms like “falling out” while describing observations. This could perhaps be replaced with terms like “deviate”?

The “naive model” requires a clearer background and explanation. What is the naive model? What are the expectations and how the diverging findings can be reconciled with the naive model can be explained in more detail.

We see in Figure 5G that there is a consistent phase offset between Switch and Stay events. However, it is unclear which results were used by authors to infer or derive the conclusion of Stay events clustering during the oscillation peaks and Switch events clustering during the oscillation troughs.

Result in Page 20:

“Given the presence of a higher frequency attentional oscillation prior to switching, we next asked whether faster or slower attention strength fluctuations predict the occurrence of switch or stay events. We thus correlated the attention strength fluctuation amplitude in each frequency range (compare with Figure 3E) with the event probability over the course of the trial (Figure S5). A stronger 11 Hz attention modulation predicted more stay events (Figure 7E, orange line, pFDR < 0.08) and a stronger modulation between 19 - 40 Hz more switch events (Figure 7E, green line, pFDR < 0.05).”

There is once again a disproportionately strong conclusion drawn for 11 Hz versus a very broad band of 19-40 Hz. Results should appropriately characterize these claims. Again, the significance of 11 Hz versus any adjacent frequency should be explained in the discussion, given the claims made.

“Hence, faster attention strength dynamics prior to switching the locus of attention generally corresponded to the appearance of high-frequency modulations of attention strength throughout the trial.”

That is, (faster) attention strength dynamics...corresponded to...modulations of attention strength. This statement appears circular and could be rephrased better.

Result in the section “Dynamics of covert attention can be dissociated from (micro)saccadic eye movements” (Pages 23-24)

“...we found that attention switches are positively correlated with the

MS rate ($r = 0.06$; $p = 0.02$).” Is a correlation coefficient of $r = 0.06$ a robust correlation? In fact, the result here indicates there is no significant correlation between MS rate over trial and attentional event probabilities.

Similarly, in “Conversely, attention switches were negatively correlated with the change in pupil size ($r_{\text{mean}} = -0.07$, $p = 0.01$; Figure S9C)...”, the correlation coefficient of $r = -0.07$ indicates that there is no meaningful negative correlation between attention switches and changes in pupil size. The inferences should be re-evaluated and updated.

“...we found that (micro)saccades preceded switching events (mean = 35%, range = 25 – 50%) more often than staying events (mean = 26%, range = 17 – 38%; $t_{19} = -13.2$, $p < 10^{-9}$). Taken together, oculomotor activity appears to play a role in switching attention between decision alternatives but not when staying focused on the same alternative.”

Switching involves planning and preparation for eye movements. Is the finding in the above statement an unexpected or novel result? The direction of the microsaccades could be informative -- for example, if the microsaccades are highly directed toward the next alternative even prior to switching, the finding above is not a surprising correlation, but predictive of the chosen alternative. If non-directional, the microsaccades might indicate a general “dislodge” from the attended location before the actual voluntary “reorienting” and subsequent “anchoring” to the next alternative.

“Recent literature highlighted pupil size as a peripheral measure associated with switching attention.”

Authors could elaborate on the direction of modulation observed in the cited recent literature here (or in the introduction), i.e., were there reports of an increase or decrease in size with switching? Does the result in this analysis support or contradict the existing literature? The motivation for this analysis and significance could be explained in further details.

Discussion:

(Page 27): “The reset might serve two purposes. First, it could correspond to an internal go cue[54], indicating readiness of the system to change[20] and briefly enhancing processing power[27,55].” Could this be elaborated? Why would readiness to change be expected after a switch (instead of before the switch)?

Similarly for (Page 28): “These sampling dynamics challenge extant models of choice, which assume that value information is optimally read-out by downstream areas implementing comparison between alternatives.” This statement could be expanded in further details. What do the extant models suggest about temporal dynamics of sampling and comparison downstream? How are the current results challenging these extant models?

Methods (Page 30):

“Participants had 2 seconds to indicate their decision with a left thumb, right thumb or foot button press for the left, right or top alternative, respectively.”

Can a foot make dexterous motions like pressing a button, matching an individual's thumbs? If not, how did this design affect

the response time or accuracy for the top alternative? Why was foot used instead of another (e.g., index) finger? Furthermore, I could not find the participants' handedness reports. Were the left and right thumb associations with the left and right alternatives counterbalanced? How are these possible motor biases accounted for in the behavior?

Minor points

Page 2: The use of words like "universal" (in "universal cognitive mechanism") should be justified or appropriately rephrased to match findings.

Page 3: "Spontaneous" seems to contradict the notion of serial execution. Should be removed or replaced appropriately.

Page 3: Potentially incorrect reference citation numbers in "Moreover, the deployed non-invasive markers of attention, namely (micro)saccades and lateralization of posterior alpha activity, might display only an incomplete picture of covert attention and its dynamics[38,69-71]." The preceding citations are numbered at [9-11] (first line in the same paragraph).

Page 4: There is a word ("and") omitted. The statement should say, "Our approach showed that during complex decisions, the strength of covert attention and decision formation oscillates at 11 Hz."

Page 10: Please add the missing figure reference for Figure 4B.

(Remarks on code availability)

Reviewer #2

(Remarks to the Author)

This MEG study investigates whether the attentional focus oscillates. First, in a localiser run, human participants see each of three single gabor patches with different contrasts and locations, used to train an inverted encoding model. This model is then tested on the main task where participants see all three gabor patches and are then cued to respond to which has the highest or lowest contrast. The success at decoding the correct stimulus is thresholded to give a modelled measure of when participants are fully attending to any particular stimuli and of overall attentional strength. This decoded attentional measure is then used to generate the following list of results: Attention strength increased most in the range 8-22 Hz and decreased between 25-45 Hz. Attention strength correlated with neuronal single amplitude envelope in the alpha and gamma bands especially over occipital and dorsolateral frontal sensors. A separate analysis of these attentional events shows that participants tended to stay at the same location or switch to another location with alternating phases matching the 11 Hz frequency reported previously, but with stay and switch events half a cycle apart. Some dissociations between switch and stay events are observed: Attentional strength was weaker around switching than stay events. Attentional strength varied in phase after either event but was not aligned before events. Attentional strength fluctuated at 22 Hz before switch events only. 11 Hz signals predicted stay, whereas higher frequencies predicted switch. Eye movements did not show such 11 Hz effects, rather were present at a lower frequency. There were more eye movements after attentional events and before switch events. Attentional events were less likely in the 0.2 s after microsaccades. A pupil analysis does not show that pupils were bigger on switch trials but if anything the opposite.

There is an interesting literature providing evidence in primate and human that attention operates rhythmically. The novel contribution here is that the analysis of oscillatory behaviour is not applied to behaviour or neural activity directly but rather to the parameters of attentional strength and locus which are generated through decoding. This comes across as a noteworthy and original development that breaks new ground while itself being well founded. It is a fertile approach in that it immediately enables the large swathe of analyses included here. Across many analyses a general common pattern does emerge, highlighting the presence of 11 Hz oscillations in attentional shifts. In general this is a striking and interesting set of results. The major concerns I have regard how the results are interpreted.

MAJOR

To prevent the results being purely driven by bottom-up attention, participants do not always simply respond to the highest contrast stimulus but rather are cued afterwards whether to respond to the highest or lowest contrast stimulus. However both these two trial types - high and low contrast targets - are collapsed together. Is there evidence for rhythmic attention when looking at the low contrast targets only? If not then the results may purely be revealing the operation of bottom-up orienting mechanisms on the high contrast trials which survive averaging together with noise on the low contrast target trials.

I do not follow the argument for the key "resetting" claim here e.g. that "attentional switching resets rhythmic attention strength fluctuations." The current argument for that includes at the moment statements like on pg 20: "Overall, our findings suggest that in particular switching events 'reset' the ongoing attention oscillation to the attentional up-state (the focused mode): post-event activity is synced between event types restarting the dominant 11 Hz (Figure 7A and 7B) and attention strength peaks at around 0.09 s and 0.18 s post-event, i.e. approximately after one and two full cycles." Why is this suggesting resetting after switch events? I would imagine that evidence for switch-specific resetting would be showing that the phase of the oscillation is always the same after switch events but not stay events - please clarify where this is present in the results.

Before switch events two things happened compared to stay events. Attentional strength fluctuated at 22 Hz, and there were more microsaccades. How are these related, could one be causing the other?

The "attention strength" is an extremely global measure. What more precisely is meant by this, how does it relate to other attentional terms in the literature? Why hypothesise that "that attention strength (i.e., vector length) would vary throughout the course of the trial as a result of the distinct processing demands in the sensory- and decision-phases of the task." (Pg 7). If attention is defined in a very global way, potentially encompassing many processes, why is it considered to vary during the task? Is the idea that this "attention" is equivalent to arousal and should vary during the task? Or that this "attention" reflects specific processes within the "sensory- and decision-phases of the task". Why would we expect "attention" to peak three times close to stimulus onset, then again early and then later in the decision phase?

It is confusing in the manuscript whether stay and switch are both attentional events with their own (orthogonal) phase - the

impression received from the results section - or whether are they part of one ongoing oscillation - as they are interpreted in the discussion? Are there two cycles for staying and switching or one? Please reconcile the descriptions of these in results and discussion.

Why is it interpreted that “oculomotor activity itself did not display rhythmicity” (pg 27) - the impression I have from the oculomotor data here is that there was clear rhythmicity but was just dissociable from that of the attentional strength. What was the ocular rhythmicity - reflecting three serial saccades to the stimuli?

Why were effects not apparent in the theta range, which is usually highlighted in other studies of attentional rhythmicity such as by Kastner, or Van Rullen.

I do not understand the “caveat” on pg 16: Why is that if stay events are more likely, why would this have an effect on the first peak offset between stay and switch events?

Most of the first page of introduction is embedding this within a decision making context, this seems unnecessary and the space might instead be used to address the above.

MINOR

As I understand it the effect shown in Fig 4 A was not statistically significant, this panel should then either be removed or made clear in the legend that its not significant.

Do the authors think that the same results could be measured with EEG, albeit without the spatial resolution

Pg 15: I don't get the interpretation of the stay data: why would “more sustained “stay rate” early-on in the decision can be mapped to the independent evaluation of the different alternatives” - why is staying for longer related to processing different alternatives independently?

Why more microsaccades after events, or before switch events? Please tone down the causal relationship implied here:

“Indeed, we showed that fixational eye movements and microsaccades had an effect on switching attention between Alternatives” pg 27

More details necessary pg 24 for this analysis: “We identified post-saccadic rhythmicity (first-to-second peak latency) in 15 out of 20 participants for both event types with no detectable change in oscillation frequency”

Pg 30: Why are there 10 possible combinations if three stimuli are drawn without replacement from a set of 5 linearly spaced contrast levels

Pg 30: what was the purpose of the experimental manipulation whereby the three gabor patch orientations either all had the same or different orientations? Were there other analyses here not reported in the manuscript?

I don't understand what is going on in this sentence; “To visually guide the participants we tilted the entire stimulus array 60 clockwise or counter-clockwise when participants were using their right or left foot, respectively.” When was this tilting happening?

Pg 39: give the most important parameters for the blink and saccades definition beyond just saying “default settings” of the software.

Why are saccades here defined as including “microsaccades” (e.g. pg 40): what was the sensitivity of the eyetracker, was it even capable of measuring microsaccades?

Pg 7 “channels” > “channel's”

Pg 32 comma after eye-blinks, formatting of wavelet bandwidth, “participants”>”participant's”

Reference 51: check if this is the right citation, is divisive normalisation the correct reference for rhythmic attentional gain?

Reference “79” is missing on page 34

“Micromechanisms” needs defining/qualifying on pg 25

(Remarks on code availability)

Reviewer #3

(Remarks to the Author)

The study utilized MEG recordings to track attention during an alternative decision task. Presumably, both the strength and locus of covert attention were extracted using data-driven analyses tools to examine how attention fluctuates during decision-making with multiple options. The approach is based on the idea that different phases of brain waves, such as peaks and troughs, are linked to distinct mental processes or states, such as maintaining focus or shifting attention. The findings reveal that the strength of attention oscillates rhythmically at around 11 Hz, influencing where attention is allocated. Attention is more likely to switch location at the trough of the 11Hz oscillation and at the peak of the oscillation, attention is more likely to stay at the previously sampled location. Further, findings show that prior to switching location, attention strength starts to oscillate at a higher frequency, thereby predicting the switch. Notably, switching attention resets the 11Hz rhythmic oscillation. The authors finally show that oculomotor behavior in terms of micro-saccades did not show rhythmic behavior and was not exclusively associated with covert information sampling. The study puts forward very interesting and novel insights into covert information sampling during decision-making. However, we have some major concerns related to

the methodology and interpretation of the results that we believe must be addressed and clarified prior to any publication.

Major concerns:

1. The authors claim that different aspects of covert top-down attention are tracked in time. In other words, the inverted encoding model must then have identified MEG activity patterns related to covert attention orientation. Since this model has been trained on data from the sensory localizer task in which Gabor patches were displayed while subjects maintained their attention on the fixation cross (i.e. did not orient attention), it does not seem possible. It is plausible that the sensory localizer task involves exogenous/bottom-up attentional shifts and perceptual processing of the Gabor patches, and hence this is more likely what the authors are decoding. Indeed, previous reports have shown rhythmic modulation of perception (vanRullen 2016, 2018) which might be in line with what the current study is showing. We believe the results remain novel and relevant, however, the authors need to clarify the context of the sensory localizer task and the impact it has on the interpretation of the decoding. Also, the decoding of covert attention needs to be justified.
2. Much of the novelty of the study relies on the link between attention and decision-making processes. However, what makes this study exceptional in this regard? Is it the use of a multiple-choice task (instead of e.g. a detection task used in many previous studies)? To make a credible statement on the relationship between attention and decision-making processes, the tracking of attention should be correlated with decision behavior during the decision task. The authors show that attention strength is correlated to performance, how about the correlation to the decision (independent of performance)? How is attention strength and locus correlated to the decision?
3. Overall, the manuscript lacks proper acknowledgement to previous literature on attentional rhythms. The introduction should contain a broader description of relevant studies on attentional rhythms and the results should be discussed and put in context to previous findings. There is quite a lot of previous work that has detected attentional sampling at lower frequencies, 8 Hz (e.g. Gaillard et al 2020 which you have cited). Also, previous work suggests that information sampling across hemispheres occurs at a Theta rhythm (Fiebelkorn 2013; vanRullen 2016). Since the Gabor patches were displayed across hemispheres in the current study, how would you interpret your results and the lack of theta rhythm in your data? A more complete picture will strengthen the article and highlight the novelties.

Minor concerns:

1. In figure 1B. It is not clear what the different option values are (or decision values ?)? In particular, in the low framing condition, why are the participants choosing the high option values (high contrast Gabor patch)?
2. Figure 2. The Design matrix is missing color coding, which hampers interpretation.
3. Page 7: The comparison between “Evoked pupil responses” and trial-averaged attentional vector dynamics (Figure 3B) needs clarification. Please specify which figures are being compared.
4. Page 9: The paragraph about decoding the three alternatives lacks clarity regarding the mapping of Low, Mid, and High options to target positions. The relationship between Figures 2B and S3 is unclear, especially since Figure 2 does not mention High, Medium, and Low options, while Figure S3 does. Same in Figure 1B, it is not clear the context of H, M and L.
5. Figure 3D is not clear. What does the color bar correspond to?
6. Figure 4C; it is not clear which time period is used. The figure legend describes “stimulus on” but it doesn’t seem to fit. Do the authors mean “framing cue on”? Using a hyphen in “stimulus/framing cue-ON period” could enhance clarity by specifying that this period refers to when the stimulus is actively “ON.”
7. At the beginning of page 16 there seems to be a mistake in the latency written in the text. They do not correspond to the figures. Should it be 0.085s? I.e. all latencies should be divided by 10 in this section? Or are you referring to latencies relative to cycle length? In that case, it should be clarified, and seconds should be omitted.
8. Page 22: Rephrase for clarity: “ISI histograms showed a single peak around 0.25 seconds (Figure 8B), indicating that the latency for the first peak of covert attention events was significantly longer.”
9. Page 23: Clarify that oculomotor activity may play a role in maintaining focus but is less pronounced compared to its role in switching attention. The statement “oculomotor activity appears to play a role in switching attention between decision alternatives but not when staying focused on the same alternative” could be misinterpreted.
10. Page 38: Correct the reference to Figure S11 in the Methods section to Figure S10, as Figure S11 is not present.
11. There is no mention of figure 7C in the text.
12. In methods, under spectral analysis, there seems to be some text missing “... 0.5 octaves (f/σ σ).
13. While the language is generally clear, consider simplifying it for a broader audience to enhance readability

References

- VanRullen, R. (2018). Attention cycles. *Neuron*, 99, 632–634. <https://doi.org/10.1016/j.neuron.2018.08.006>
- Fiebelkorn, I. C., Saalman, Y. B., & Kastner, S. (2013). Rhythmic sampling within and between objects despite sustained attention at a cued location. *Current Biology*, 23, 2553–2558.
- VanRullen, R. (2016). Perceptual cycles. *Trends in Cognitive Sciences*, 20, 723–735. <https://doi.org/10.1016/j.tics.2016.07.006>

(Remarks on code availability)

Reviewer #4

(Remarks to the Author)

I co-reviewed this manuscript with one of the reviewers who provided the listed reports. This is part of the Nature Communications initiative to facilitate training in peer review and to provide appropriate recognition for Early Career

Researchers who co-review manuscripts.

(Remarks on code availability)

Version 1:

Reviewer comments:

Reviewer #1

(Remarks to the Author)

We thank the authors for addressing and clarifying several important issues raised. However, we still have a few remaining queries. We would appreciate it if the authors could address them to substantiate the claims made in the paper.

1.2 : We continue to remain unclear about the significance of the medium decision value: if there are only two framing conditions, when does the medium value occur? What was the purpose of using this condition?

1.3 : The authors have clarified the issue on fatigue and IOR due to long presentation duration. Unfortunately, it is still not clear to us whether the authors are claiming that there was fatigue (or not) during that period? Furthermore:

(a) Although the late interval was a reliable predictor of decision accuracy, the pupillometry results show that task engagement varied within the trial.

(b) As the authors mention in the next comment (R1.1.4), there was no significant difference in overall accuracy between shorter and longer time durations: does addition of extra time during the trial might still offer any benefits in terms of supporting participant behavior or performance? Moreover, as observed in figure 4 (in main text) and figure R2, the attention strength reaches its peak at 1s. Therefore, could the authors kindly provide further clarification regarding the necessity of long stimulus presentations for studying attention strength?

1.6 : We understand that the authors wanted to train an IEM without the confound of oculomotor dynamics. However, we wonder if it would not be convenient to design a localiser task closer in design to the main task. We would also like to know if the authors controlled for eye-movements in the main task. It would help if the authors could also elaborate on how the IEM is able to capture top-down attentional processes in the later periods of the trial.

1.8 and 1.9 : We found it somewhat unconventional to normalize the attentional vector strengths with the stimulus presentation period, particularly given that there is active processing of the visual stimuli during this time. Would it not be convenient to use the pre-stimulus phase for normalisation where the random fluctuations of the attention vector would nullify each other and act as a more reliable baseline?

Additionally, in figure 3a, we observe that the attentional strength decreases below average after ~1s. This observation raises questions about the necessity of a longer task duration and the reliability of the results during the later stages of the trial.

1.23 : We appreciate the authors' explanation of the naive model and the "follow" interpretation. However, we think it would improve the readability further if the authors could elaborate the interpretation in more detail with its underlying assumptions and hypotheses.

1.24 : While we appreciate the arguments, they are qualitative in nature. It would be beneficial to provide additional quantitative evidence regarding the occurrences of "stay" and "switch" events to support the conclusions. Additionally, it would be helpful to include a more detailed quantitative explanation for the cosine fitting of the event oscillations.

(Remarks on code availability)

N/a

Reviewer #2

(Remarks to the Author)

In this resubmission, the authors have addressed my comments adequately as I briefly outline below for completeness and I have no further comments.

The authors have substantially overhauled the abstract and introduction to focus more on the attentional sampling literature.

The authors have well accommodated a point also raised by the other reviewers about the need to address whether bottom-up processing may be driving these effects. They have principally addressed this in a new analysis (Fig. 3) that shows that the key effects are present later on in the presentation interval suggesting that it is not driven solely by bottom-up effects which would be affected to be only early and transient. Another new analysis shows that the decoder still works when the task was to decode which stimulus was the lowest contrast: earlier in the trial the decoder is affected so the low-level features affect only the early response.

The authors have clarified that the process described as “reset” relates principally to frequency and not phase, and they now better explain their definition of “Attention strength” as how well the decision alternatives are represented.

A new figure illustrates how the two event types (switch and stay) are considered part of (and arising from) an ongoing oscillation of attention strength and the authors argue they are also indicative of two modes with distinct substrates.

(Note to authors: In the pdf I received of the rebuttal letter the Links didn't work, some references were listed as [source] and the logistic regression in Fig 5 was described as “logarithmic” but these problems did not extend to the actual manuscript.)

(Remarks on code availability)

Reviewer #3

(Remarks to the Author)

We thank the authors for their extensive revision of the manuscript. Overall, the manuscript has become much clearer with a better focus. However, we still have some concerns that we would like the authors to address.

R3.1.1 The procedure of IEM is clear but the rationale behind using the sensory localizer (SL) task to reconstruct representations of covert spatial attention still needs some clarification in the main text. The use of a task that does not involve covert shifts of attention to prevent micro-saccadic contamination of the data is well motivated. However, using the SL task, in which participants were given both instructions to ignore the peripheral stimuli and a specific central flickering detection task to prevent covert attentional shifts, seems counter-intuitive when the aim is to extract cortical representations of covert attentional shifts. As mentioned in my first review, if there were shifts in attention during the SL task, they are most likely bottom-up driven as the stimuli were behaviorally irrelevant for the task. However, from figure S1, there is a sustained representation in the data related to the Gabor patch location (i.e. indeed probably an attentional shift). The sensor contribution during the optimal decoding time shows contribution from the occipital cortex which suggests that the SL task initially evoked visual responses to the Gabor patch locations (which is expected). However, for the sustained attentional information, we would expect the sensor contributions to cover more parieto-frontal areas. We would therefore like to see the sensor contribution after the initial transient increase in decoding accuracy (e.g. 0.2-0-35s) as this would clarify the information that is encoded in the IEM.

We agree with the authors that the use of the IEM on data from the three-alternative decision-making task seems to be able to reconstruct the strength and locus of covert attention (and plausibly other intertwined cognitive processes), due to the tight correlation with behavior that you show during the task. However, we would like the authors to elaborate on alternative interpretations related to the origin of the IEM (i.e. built from the SL task). For example, the weight matrix G represents how each MEG sensor weighs different positions of the Gabor patches, which were displayed without behavioral relevance (i.e. MEG activity should dominantly reflect visual processing and bottom-up driven attention). When applying this G matrix on MEG data from the three-alternative decision-making task (i.e. multiplying the inverse G with MEGMDM), it should weigh the MEGMDM data based on the activity evoked by the Gabor patches in the SL task. If this activity dominantly represents visual information processing and bottom-up attention, from a functional perspective, what does this mean? Is there a functional overlap between visual, bottom-up and top-down attention that you are capturing? We believe it is crucial to clarify and discuss this in the main text.

R3.1.2 I am happy with the amendments and have no further comments.

R3.1.3 The revised introduction does a better job incorporating literature on attentional rhythms, and the discussion of oscillations across theta, alpha, and gamma is a welcome addition. However, we noticed that some key points outlined in your response to our comment, particularly the discussion of how task complexity and peripheral stimulus presentation might influence the observed attentional frequency—are not explicitly reflected in the revised text.

In the discussion you added a brief section stating that the 11Hz rhythm might further depend on the cognitive domain and task and stimulus complexity. However, it would be appropriate to elaborate a bit more (like in your response) on how your observed 11 Hz frequency compares to previous findings, which have reported attentional sampling at both lower (4-8 Hz) and higher (7-12 Hz) frequencies. Additionally, prior work suggests that information sampling across hemispheres is often associated with theta rhythms (e.g., Fiebelkorn, 2013; vanRullen, 2016). Given that your task involves sampling across hemispheres (Gabor patches were displayed across hemispheres), could you clarify or attempt to explain why a theta rhythm was not observed in your data? Including these points would provide a clearer connection between your findings and the existing literature.

R3.2.1 Ok

R3.2.2 Ok

R3.2.3 Ok

R3.2.4 Thank you for the clarification. However, we still find this point somewhat unclear in the text. Do you mean that Figure 2B illustrates the estimated responses and attentional vector, while Figure S3 demonstrates that—regardless of the option values (High, Medium, or Low)—these responses consistently show a broadband decrease between 8–45 Hz? If so, it might help to state this more explicitly in the manuscript to ensure clarity.

R3.2.5 Ok

R3.2.6 Ok
R3.2.7 Ok
R3.2.8 Ok
R3.2.9 Ok
R3.2.10 Ok
R3.2.11 Ok
R3.2.12 Ok
R3.2.12 Ok.

(Remarks on code availability)

It seems that the link above is a view-only link. I cannot find any files or data.

Reviewer #4

(Remarks to the Author)

(Remarks on code availability)

Version 2:

Reviewer comments:

Reviewer #1

(Remarks to the Author)

Thank you for your efforts with addressing our comments. We have a few pending concerns:

R1.1.2 - We appreciate the additional explanation and background reference provided regarding the task design. However, examining the referenced study raises new questions. In that study, participants valued the stimuli before the task, whereas in the present work, responses are determined by the experimenter based on contrast differences, which may introduce a substantial bottom-up component of attention. This again raises the question of whether participants maintained attention for an extended duration. Additionally, we understand that the use of three stimuli was meant to increase task difficulty by introducing more distractors, but the authors seem to suggest that the justification for this design is beyond the scope of the current paper.

R1.1.3 - While we agree with the authors' analysis, the specific points regarding pupillometry and behavioral results remain insufficiently addressed. The authors demonstrate that attention allocation is consistent throughout the trial duration. However, our concern persists regarding whether participants are actively sampling the stimuli during this period to make a decision, or whether decisions are made almost immediately after stimulus onset given the strong bottom-up component of the task.

R1.1.6 - We appreciate the acknowledgment that the localizer used is not optimal for disentangling top-down and bottom-up processes between the localizer and the main task. The explanation is helpful; however, the approach misses support from existing literature to establish the protocol's validity. The argument that task framing can dissociate low-level from high-level features remains unconvincing, as participants were always asked to choose one of the extreme stimuli. The eye-tracking controls are clarified sufficiently.

R1.1.8 & R1.1.9 - The analysis to show increased attention in the late-trial period (when stimulus is on the screen) compared with pre-stimulus periods seems to confirm an expected result. Our main concern persists that the reported late-trial attentional dynamics may not be necessary for participants' decision-making.

R1.24 - The details regarding methodology used for cosine fitting is not fully addressed and the references to appropriate literature to support the method are lacking.

(Remarks on code availability)

Reviewer #2

(Remarks to the Author)

I had raised no further comments in the previous review round. The new changes to the manuscript address points relating to for example whether the non-monotonic pattern of results may reflect fatigue; on the dissimilarity between the localizer and the main task; the appropriate time for baselining in decoding; the absence of theta effects; and particularly on further elaboration of the interpretation. Accordingly, much new material has been introduced particularly in the intro and

discussions, none of which comes across as problematic, and without for example over-inference of causal relations (between eye movements and attention or either with brain processes). I have no additional comments (aside from some new typos: Pg 6: Track to -> track; Pg 8: When attention "is" weak; Pg 21: aren't: are not).

(Remarks on code availability)

Reviewer #3

(Remarks to the Author)

The authors have adequately addressed all my comments and I have nothing further to add.

(Remarks on code availability)

I still cannot access the code as permission is required. When I request access an error is returned.

Reviewer #4

(Remarks to the Author)

(Remarks on code availability)

Version 3:

Reviewer comments:

Reviewer #1

(Remarks to the Author)

(Remarks on code availability)

We thank the editor and the reviewers for their positive assessment of our work and for their constructive comments, which were very helpful for improving the manuscript. The reviewers' feedback has led us to carefully revise the entire manuscript and to perform several new analyses, with interesting new results that we included in the revision. We feel that this has substantially strengthened the manuscript, the accessibility of the reported findings and the clarity of our conclusions. Before providing point-by-point replies to all reviewers' comments, we first summarize the main changes and new results below:

- 1) We thoroughly revised the Introduction to emphasize the existing body of literature on rhythmic attention and clarify how our approach advances the field.
- 2) We motivated the choice of the sensory localizer task and identified potential caveats in the interpretation of the decoder results in the main task.
- 3) We performed new analyses to understand the effect of bottom-up processing and low-level salience. We directly compared the influence of low-level features (contrast) on the decoding of frame-dependent decision-relevant features (high vs. low value) and found a contrast-driven effect of the decoder for the initial 0.25 seconds after stimulus onset (that is, prior to the presentation of the framing cue). We consider this contrast-driven effect negligible with respect to our main conclusions since it disappeared for the remaining 3.5 seconds of the stimulus presentation period (Supplementary Fig. S4). Further, the rhythmicity was not affected by the framing condition (Supplementary Fig. S8).
- 4) We analyzed the relationship between the attentional events with the reported choice. We included this analysis in the main text and Figure 5 as a new panel. In brief, more stay and switch events towards the high value option was correlated with more correct responses, i.e., a stronger tendency to choose the high value option (logarithmic regression). This establishes the functional significance of attentional events on behavior, an aspect that was underexplored in our initial submission.

- 5) We clarified important concepts that previously caused confusion, specifically: the nature of the proposed “reset” and the relationship of staying/switching events and the underlying attentional strength oscillation.

Reviewer #1 (Remarks to the Author):

Siems et al. examine the temporal dynamics of visual attention and information sampling using MEG and neural decoding. In a 3AFC perceptual choice task, they report a rhythmic sampling of covert attention at 11 Hz, such that attention is focused at a single location or shifts to different alternatives during the peak and trough of the oscillation, respectively. They also report a reset mechanism for the attentional oscillation -- such that the sampling frequencies of staying and switching between alternatives align following a switching event. These findings shed light on possible mechanisms for serial information processing during multi-alternative decision making. They also explore the link between information sampling with other markers such as microsaccades and pupil diameter. While the authors have performed many interesting analyses to support their claim, some points must be addressed to improve the clarity and robustness of the results:

We thank the reviewer for the overall positive assessment of our manuscript and the constructive and detailed comments. We will address each of the raised points individually in the following.

Major concerns:

R1.1.1

Introduction (Pages 3-4): There is an existing body of work on rhythmic sampling models of attention and decision making (e.g., Busch et al., 2009; vanRullen, 2016; Landau & Fries, 2012). The introduction appears under-motivated. It could benefit from expanding on previous studies and highlighting the existing knowledge and unknowns in the field, and how this study addresses these unknowns. For example, how have the “current neurophysiological approaches to decision-making ignored these internal sampling dynamics”? How exactly do the “non-invasive markers of attention, ..., display an

incomplete picture of covert attention and its dynamics”? How does the authors’ approach add to the existing studies?

Did the authors have specific hypotheses for this study? For example, the studies listed above report sampling frequencies at 4 Hz (for two alternatives) and 8 Hz (for a single stimulus). Did the authors hypothesize a similar sampling frequency? If not, why not? If yes, why choose a sampling frequency of 11 Hz instead? What might be the significance of 11 Hz sampling frequency in terms of the underlying neural mechanisms or physiology? This could be motivated in the introduction/discussion.

We thank the reviewer for these very important insights. Indeed, a growing body of literature has reported attentional rhythmicity at around 4-8 cycles per second and related neural activity in the theta range. However, findings for rhythmic sampling at higher frequencies at around 7-12 Hz are common as well (compare Kienitz, Schmid & Dugue, 2021, EJN, Link; Van Rullen, 2016, TiCS, Link). There might be several reasons for a higher sampling rate in our study potentially resulting from the task setup. Our protocol consisted of a three-alternative protracted decision-making task with peripheral stimuli. First, one speculative interpretation highlighted in the literature is that peripheral stimuli elicit a higher sampling frequency than centrally presented stimuli (van Rullen, 2016, TiCS). Second, attention re-allocation in a protracted decision-making task might rely on different goal-oriented cognitive strategies than in target-detection or simple visual search tasks. Studies with target detection and visual search protocols often display rhythmicity around 4-8Hz (Kienitz, Schmid, & Dugue, 2021, EJN; Fiebelkorn & Kastner, 2019, TiCS, Link; Senoussi, et al., 2019, J Vis, Link), whereas protocols relying on more extensive processing of attended stimuli (compare for example visual recognition: Caplette, Jerbi, & Gosselin, 2023, JNeurosci, Link) as well as graded response reports (compare Michel, Dugue, & Busch, 2020, EJN, Link) can display sampling at higher rates. It appears that further increasing the stimulus and task complexity can impact the sampling frequency (compare for example Merholz et al., 2022, SciRep, Link; Gaillard et al., 2020, NatComm, Link).

We agree with all the reviewers that the details of the rhythmic attention literature and its implications for our task and findings need to be discussed in more detail. We thus extensively revised the Introduction and Discussion sections to address this field of research in more depth and to better clarify our hypotheses. That way we could better highlight the benefits of our research for the attention, decision-making, and broader neuroscience communities.

R1.1.2

Results (Page 5) and Figure 1B: How is accuracy being determined using decision value and contrast framing? I could not find it in the methods. The term “decision value” (H,M,L) in the figure legend (1B) is not defined in the text. Is this different from the term “option value” (used in the figure)? Overall, the accuracy results are unclear in Figure 1B.

We thank the reviewer for raising this point. First, ‘option value’ and ‘decision value’ are the same and we converged in the text and figure legend to ‘decision value’. Second, the decision value is distinct from the stimulus’ contrast level. Thus, the highest displayed contrast will hold a high decision value in high framing trials while the same stimulus will have a low value in low framing trials. We updated the Figure legend 1b and the Method section accordingly.

p. 6: “[...] The highest contrast stimulus has the highest decision value in high framing trials, while it has the lowest value in low framing trials.”

R1.1.3

Task Design (Figure 1A): The stimulus is presented on screen for a total of 3.8 s (including 2.5 s of decision period). How do we account for fatigue later in this period, or for inhibition of return?

Generally, we chose the length of the decision phase to accommodate for a sufficiently long time window to a) observe the decision-making process unfolding, b) register enough covert attentional

saccades to analyze their behavior and c) give enough time to participants to adapt to the potentially demanding rule-switching (framing) between trials. Fatigue is an important factor for these extended time periods of neural processing. However, recent work on “expanded judgment” paradigms has underscored that participants are capable to integrate evidence over periods of several times the length of our decision phase (2.5 s): Waskom and Kiani (CurrBio, 2018, Link) for example could show that contrast information can be successfully integrated over periods of up to 30 seconds (over 10 times our accumulation window; compare also Vickers et al., 1985, ActaPsych, Link).

That said, we acknowledge that contrary to “expanded judgment” tasks where information is presented serially, our experiment involved static stimuli, which can induce fatigue under prolonged viewing. Indeed, pupil diameter (i.e. a peripheral proxy for task engagement), peaked early-on and dissipated towards the end of the viewing period (Fig. R1). However, in the section “Attention strength fluctuations correlate with behavioral performance” we emphasize that the late interval was a relevant predictor of decision accuracy.

Fig. R1 | Pupil diameter (normalized over all trials within each participant) separately for correct and error trials.

We thus conclude that, even though task engagement varied within the trial, participants meaningfully used the entire length of the decision phase. It is possible that the length was more

than sufficient in relatively easy trials but participants seem to have adjusted to the trial length they were presented with (please see also next comment).

We also agree that Inhibition of return (IOR) could modulate attention within the course of a trial. However, we do not find evidence for IOR in our data but rather the opposite. First, switch rates are relatively sustained throughout the trial (Supplementary Fig. S6). Second, we observe an increased tendency to return to the initially attended decision alternative when two switches appear in succession (such as when switching from left to right and back to left) relative to when switches do not return to the original location (left, right and top): return rate = $70.28\% \pm 6.23\%$, mean \pm standard deviation, minimum return rate = 64.62% ; return rate = $1 - \text{IOR rate}$. This finding underlines the potential role of the identified attentional switches as implementing binary comparisons of options.

R1.1.4

Furthermore, selecting the stimulus with high or low contrast is a cognitively low-load task. Does the task demand 2.5 s of processing time (post-framing)? If the subjects make a decision early, it might be reasonable to assume that they would not keep sampling the space for the rest of the decision period. Why was a shorter presentation period not tested?

We acknowledge that the presentation duration was on the longer side for a perceptual task. In fact, before determining the duration of a trial for the MEG experiment, we conducted a similar experiment (without the framing manipulation) outside the MEG scanner, where we used shorter stimulus presentation (0.5 and 1.5 s in different trials). The overall accuracy in these experiments was comparable to that obtained in the MEG study, and we did not observe a strong effect of presentation duration (data not shown here, manuscript in preparation). In the MEG experiment reported here, we opted for an “interrogation” paradigm which eschews speed-accuracy trade-off

considerations. Practically, we wanted to avoid placing participants under time pressure, which could prevent them from sufficiently sampling all available alternatives.

The reviewer is correct in noting that participants could potentially reach an early decision and “switch off” for the remainder of the stimulus presentation. We considered this possibility as a trade-off against the opposite scenario (participants not having sufficient time to sample all information) and opted for a generous presentation duration. In another set of analyses (data not shown, manuscript in preparation), we do find that the early decision phase is pivotal but participants were engaged in confirmatory sampling strategies all throughout the trial. We have chosen to discuss these more elaborate strategic sampling aspects (e.g. confirmatory sampling) in a different manuscript. Within the rhythmic sampling analysis framework presented here, we see that rhythmic patterns are evident in the late presentation interval suggesting that rhythmicity can last for several seconds (see Fig. 3). Overall, we acknowledge that the stimulus presentation was on the longer side; nevertheless, participants adapted their processing to this extended window and continued to sample information until the end of the presentation interval.

We further agree with the reviewer that the framing manipulation seems low demand. However, participants performed long blocks embedded in a very lengthy experiment (5 hours total), while they had to remain still. The challenging experimental context (interleaved low vs. high framing) was an additional motivation for a more generous presentation duration.

Lastly, following the reviewer’s intuition, during the late decision phase we might expect a decrease in attention strength for ‘easy’ trials compared to ‘hard’ trials. However, empirically we did not find any significant differences in attention strength for different ‘difficulty levels’ (difference between high and mid value option):

Fig. R2 | Z-scored Vector Length separately for difficult (high - medium value = 1), medium (high - medium value = 2), and easy trials (high - medium value = 3). The z-scoring was performed based on the mean and standard deviation when combining all trials.

R1.1.5

Also, what was the purpose of the mid-contrast stimulus? Would the results be different if a two-alternative (high and low) decision making task were used?

Most studies and models in decision-making have traditionally focused on two-alternative forced-choice (2AFC) paradigms. A key question arising from this focus is how mechanistic insights derived from 2AFC paradigms scale-up to multialternative decision-making scenarios. As highlighted in the Introduction, multialternative paradigms differ fundamentally from 2AFC tasks, particularly in their demand for serial information processing. Additionally, a substantial body of research on “decoy” effects in experimental psychology demonstrates that decision processing changes radically when a third alternative is introduced to a binary choice set (Tsetsos et al., 2016, PNAS, Link; Bussemeyer et al., 2019, TiCS, Link; Louie, Khaw, & Glimcher, 2013, PNAS, Link).

Our goal in this paper was to investigate information sampling within a paradigm that is “complex enough” to induce interesting sampling dynamics. We believe that rhythmic sampling

would have been present in binary trials, but we also think that binary decisions would settle much faster and with fewer switches. Ideally, we would have interleaved binary and ternary trials to enable direct comparisons across the two. We circle back to this point and the need to directly compare ternary and binary decisions in the Discussion.

With our experimental design, we were also interested in considering the role of information sampling in the presence of decoys. However, these analyses are beyond the scope of the presented manuscript and will be discussed in detail elsewhere (manuscript in preparation).

R1.1.6

Figure 2: The encoding model was trained using the Sensory Localizer Task, which is a bottom-up paradigm. However, the model is then tested on the Main Task which is a top-down task. Can the bottom-up channel responses be translated to a top-down attention task?

We thank the reviewer for the insightful comment and we agree that the nature of both tasks (sensory localizer and main task) is in principle different.

The inverted encoding model (IEM) is a method for reconstructing population-level stimulus representations from aggregate neural activity data. We thus agree that the approach is generally different from mapping out an attention-based topography where the contribution of visual stimulation could be partialled out almost completely (see for example Silver, Ress, & Heeger, 2005, *J Neurophysiol.*, Link). Further, we followed the assumption that the stimulus representation would be stronger for attended than for unattended stimuli (compare for example Buschman & Kastner, 2015, *Neuron*, Link; Posner, 2016, *Quart. J. Exp. Psy.*, Link). However, we do not argue that the estimated responses are pure ‘attention signals’ but could as well be a compound signal of attentionally modulated decision input.

Furthermore, our choice of a sensory localizer was motivated by the goal to eliminate the effects of eye movements as much as possible. Recent literature has shown that cued covert

attention protocols display strong contaminations with miniscule eye movements (see for example Hafed, Lovejoy, & Krauzlis, 2011, J Neurosci, Link; Liu, Nobre, & van Ede, 2022, NatComm, Link; Linde-Domingo & Spitzer, 2023, NatHumBeh, Link), which can potentially affect stimulus processing and their representations (see for example Hafed, Chen & Tian, 2015, FiSN, Link). This relationship has largely been overlooked. Based on these findings a localizer including spatially cued targets would thus be contaminated with eye-movements or movement preparation artifacts. Consequently, decoded attention allocation and its dynamics would be intertwined with oculomotor dynamics.

For the revision we further quantified whether low-level features, i.e. contrast, lead to differences in stimulus decodability, i.e. estimated responses. We ask whether the estimated response is different within each decision value class (high, medium, low) between the two framing conditions. For example, in the “high value” class, we compare the decodability of the highest contrast (high framing) with the lowest contrast (low framing). Thus, the low-level (bottom-up) feature differs but the decision value is the same. If contrast was the driving force behind the estimated response we would expect a significant difference between framing conditions within each decision value:

Fig. R3 | Decodability (estimated response) differences between High- and Low-Framing trials separately for **a** high, **b** medium, and **c** low value option within each respective trial. The circles schematically depict the compared contrast levels. The figure has been added to the Supplement of the revised manuscript.

We find a significant difference ($p < 0.01$) right after the stimulus comes onto screen (-1 seconds) but not in the decision-relevant stages of the trial (i.e., after the framing cue is presented). Thus, low-level features of the stimulus appear to affect only the onset response. We thus conclude that the IEM in combination with a bottom-up driven sensory localizer task is sensitive to top-down evoked signals. We added the analysis to the Results (p. 10) and Figure R3 to the Supplement (S4).

R1.1.7

The figure could be improved (more details could be added) to describe the IEM process in a clearer manner. The color map in panel A (design matrix) for predicted response is missing/blank. It should be white-to-black for 0-1. Figure abbreviations and notations are not explained in the legend.

We thank the reviewer for the attentive reading and revised the figure accordingly.

R1.1.8

Results in section "Attention strength oscillates at 11 Hz" (Page 7-9): Why is the attention strength vector length non-zero and negative during fixation and stimulus onset? Would we not expect a vector length of ~0 magnitude with gaze at the fixation cross during this period? Eye-movements in this period would be randomly directed and thus, should nullify each other to give a ~0 vector length. What does negative vector length (i.e., magnitude) mean, and with respect to what baseline is it negative?

We thank the reviewer for the comment. The values in Figure 3a were z-scored with respect to the stimulus presentation period. A strongly negative value in the pre-stimulus interval indicates that the vector length is below the average of the stimulus presentation period. This detail was

only mentioned in the figure legend and we added it to the main text in the revised manuscript as well (p. 7).

R1.1.9

Page-8: “Evoked pupil responses resembled the trial-averaged attentional vector dynamics, albeit with more sluggish dynamics (Figure 3B; $r_{\{pupil,veclen\}} = 0.54$; $p = 0.007$.” Which time period has been used to calculate the correlation and why?

We used the entire stimulus presentation period to compute the correlation. Generally, we excluded the pre- and post-stimulus intervals as the vector length must be interpreted differently when the stimulus is off. In the pre-stimulus phase the Attentional Vector might vary due to for example random fluctuations of ongoing neuronal activity. On the other hand, directly after the stimulus offset in the post-stimulus period participants might have still been remembering the options or display extended stimulus processing from memory. We further updated the Methods section (pp. 36-37) to address the interpretational caveats of the IEM decoding output during stimulus off periods.

R1.1.10

Page-9: What is the justification for choosing 11 Hz for the alpha band and 40 Hz for the gamma band? Broadband frequency range is not consistently defined.

We thank the reviewer for pointing this out. The broadband signal on the neuronal level was between 0.2 to 200 Hz and on the decoder level the raw vector length signal. We added the broadband frequency range to the revised manuscript (p. 9). Further, we agree with the reviewer that focusing the band-limited analysis in Figure 3 on individual frequencies is not warranted from the presented data alone. We thus adapted Figure 3 and revised the manuscript to display

fluctuation activity averaged over identified frequency ranges (7-13Hz & 25-50Hz; see also below).

R1.1.11

The authors estimated the correlation of neuronal activity with attentional fluctuations, at both alpha and gamma frequencies. Now, neuronal activity (Figure S2) shows a decrease in both alpha and gamma frequencies, while attentional strength (Figure 3E) shows an increase in the alpha band but a decrease in the gamma band. As such, the positive correlation reported (Figure 3H) for both alpha and gamma frequencies between these two signals is contradictory -- should the correlation for alpha frequency not be negative?

We thank the reviewer for the attentive comment and would like to clarify the potential misunderstanding. Indeed, the change of the attentional fluctuation signal with respect to baseline is reciprocal between the two analyzed frequency bands and measures. However, we correlate the time courses of activity, both neuronal and attention fluctuations, over the stimulus presentation period. A positively offset signal (with respect to baseline) can still be positively correlated with a negatively offset signal (with respect to baseline) if their time courses are similar. A positive correlation in the alpha band (11 Hz) can thus be interpreted as follows: When attention strength increases, neuronal alpha power is less negative/decreased; vice versa when attention strength decreases, alpha power is more negative/decreased. We tried to clarify this aspect in the revised manuscript (p. 9).

R1.1.12

Some results are described in the text without any supporting evidence -- descriptive summary measures (mean, std) or statistical test results (p values) are not reported consistently or sufficiently for all results.

We thank the reviewer for the attentive reading. We revised the manuscript accordingly.

R1.1.13

Figure 3: Panel A - Legend mentions a shaded area but it is not visible in the figure.

We apologize for the poor quality of panel 3A. We uploaded the high-definition figures with the revised manuscript.

R1.1.14

Panel C – The single trial examples do not support the trends in panel A. Data is too noisy to show any distinct or identifiable patterns resembling the three peaks described in text and panel A.

We fully agree with the reviewer. This is indeed the intended interpretation of Figure 3c and the rationale for applying spectral analysis of this signal in the following. We clarified this point in the revised manuscript (p. 9).

R1.1.15

Panel D – The significance/inference of this result is not clear. The amplitude spectra for different time points overlap with each other, and are not visible for comprehension. The figure could be improved for a clearer depiction of the shift in attention fluctuations. Crucially, the x-axis is plotted in log-scale and it is difficult to visually corroborate the peak at 11 Hz (same in Panel E).

We thank the reviewer for pointing this out. Indeed, the main conclusion/inference from panel 3d is the descriptive dissociation of the attention strength fluctuations from baseline to stimulus presentation. We revised the manuscript accordingly (p. 9).

R1.1.16

Panel E – The authors report a frequency band for decrease (25-45 Hz), but a specific frequency of 11 Hz for increase in attention strength. If the singular report of 11 Hz (and not, say, a band from 8-16 Hz as shown in the figure) is an inference from the results, it is not supported by the reported results or figure. Could the authors report additional analyses to support their claim of fluctuation at 11 Hz? One simple addition would be a comparative report of fluctuations in the adjacent frequencies. If the focus on 11 Hz is hypothesis driven, the motivation should be clearly explained.

We agree with the reviewer that the focus on 11 Hz was not motivated by the results in this section but later findings. We revised the manuscript accordingly and added test statistics based on the data presented in panel f.

R1.1.17

Additionally, there is no change in the fluctuation in the 8-16 Hz band over time. Should we not expect a difference in attentional strength before and after framing in this frequency band? What can we infer about neural decisional processes from this? However, there is a change in the fluctuation for the 25-45 Hz (increase at around 1 second relative to Framing onset). This is not reported in text. But is there any significance of this result?

Panel G - We see a decrease in attention strength after 1 second. Is it due to (a) fatigue, (b) voluntary decrease in attention strength, or (c) shift from attention to decision making?

We thank the reviewer for the insightful questions and would like to address both points combined. As shown in panel G there is indeed variation in the attention strength fluctuation over time. We

apologize if this may have been missed due to poor Figure quality in the previous submission. However, the lower-frequency dynamics are stronger over the entire stimulus presentation than during baseline (compare the minimum on the y-axis is 0.03). We (descriptively) see an increase right before and right after framing cue onset. The same behavior is mirrored in the high frequencies but negatively offset.

Fig. R4 | High-Frequency attention strength fluctuations relative to baseline.

Here, in Figure R4, we observe a very similar pattern to Figure 3g: A local maximum around framing onset and a local minimum at about 0.5 seconds. This translates to a smaller decrease in high-frequency fluctuations around framing onset and a stronger decrease at about 0.5 seconds post-framing.

Overall, the reported dynamics of attention strength point towards the conclusion that not only the 'where' (Fig. 5c) of attention and perception during decision-making is important but also the 'when' (compare also Fig. 4b). This challenges prevalent models integrating attention as time-independent variable into the sequential decision-making process (see for example Krajbich, 2019, *Curr Opin Psychol*, Link). We further argue that even though low frequency fluctuations descriptively decrease during the later stages of the trial (Fig. 3g) the findings presented in Figure

4b speak for a task relevant role of attention strength fluctuations until the stimulus offset. This might indicate that during error trials participants are 'loosing focus' and are less attentive than during correct trials (compare Fig. 3a & 4a), which predominantly translates to fluctuations between 8-16Hz (Fig. 4b).

We revised the discussion section of the manuscript to put these aspects forward (see pp. 26-27).

R1.1.18

Results in the section "Attention strength fluctuations correlate with behavioral performance" and Figure 4:

Same issues as in Figure 3E. There is no evidence or values reported for the statement in Page-10, "However, this was a nonsignificant trend".

We thank the reviewer and revised the manuscript accordingly.

p. 10: "[B]roadband attention strength (compare Fig. 3a) was slightly but not significantly increased late in the decision phase when choosing correctly (Fig. 4a; all $|t_{19}| < 1.5$, $p > 0.05$)."

R1.1.19

Additionally, authors report a peak in attention strength fluctuation in a frequency range from 8-16 Hz, then state that "fluctuations around 11 Hz might be predictive..." Once again, the focus on 11 Hz is neither supported nor explained robustly.

We agree with the reviewer and revised accordingly.

p. 10: "We identified significantly increased attention strength fluctuations related to choosing correctly from 1.5 seconds post framing until the end of the trial, peaking in a frequency range from 8 – 16 Hz ($p_{FDR} < 0.05$). This indicates that band-limited attention strength fluctuations are predictive of choosing correctly (Fig. 4b)."

R1.1.20

Switch and Stay attentional events v/s High and Low contrast framing: Did the authors analyze the switch and stay latencies for low and high framed trials separately? In high contrast framing trials, the salient stimulus is also the task-relevant stimulus, thus it is expected that the individual's attention will "stay" following Framing cue, and will last longer than switch. On the other hand, in low contrast framing trials, the salient stimulus (high contrast) is different from the framed, task-relevant stimulus (low contrast). This could lead to a faster switch from the bottom-up target (high contrast stimulus) to the top-down target (low contrast stimulus) after framing cue onset. While the study reports latencies between two events, could this framing effect (i.e., latency difference w.r.t. framing onset) contribute to the phase difference between switch and stay event frequencies?

We thank the reviewer for this insight. We addressed this question directly and compared inter-event-interval histograms and latencies separately for both framing conditions. However, we did not find any effect of the framing condition on the attentional event latencies.

Fig. R5 | Inter-event-interval histograms separately for high and low framing trials.

Uncorrected inter-event-interval histograms separately for **a** stay and **b** switch attentional events in High- (colored solid line) and Low-Framing conditions (colored dashed line; compare Fig. 5). The black dashed line indicates event intervals under random conditions.

As we have also pointed out above the IEM approach in combination with a bottom-up driven sensory localizer task can also be sensitive to top-down evoked signals. This finding here further strengthens our conclusion. But we agree with the reviewer that this is an important distinction and we added the above analysis (p. 15) and figure (Supplementary Fig. S8) to the revised manuscript.

R1.1.21

Figure 7: “N denotes the number of participants for which a sine-fit was possible”. The rejection criteria and sine-fitting is not explained in the Methods. What are the parameters of this sine fit? Were the fit parameters different for each subject, and in turn, was the rejection subjective? What is the justification for rejecting a participant?

We thank the reviewer for the attentive reading and pointing this out. This was indeed incorrect and the legend corresponded to an older version of the figure. We did not derive the frequencies through a sine fit but compatible to Figure 5 by computing the latencies between local maxima from the detrended and smoothed time courses (Fig. 7b). Thus, it was only possible to derive this frequency if a subject displayed two or more peaks. We revised the figure legend accordingly and updated the Methods section (pp. 39-40).

R1.1.22

Furthermore, it is difficult to track which analyses are for paired data and which are not (Figures 5 and 7). What is the justification for the choice of paired or unpaired tests across analyses?

All statistical comparisons between conditions were conducted with paired tests. We revised the manuscript to clarify this aspect.

R1.1.23

Results in the section “Brief attentional saccades impact the ongoing attention strength fluctuation” (Page 16) “We would expect that events, ..., are falling out from the ongoing 11 Hz oscillation.” It might be recommended to avoid using relatively colloquial terms like “falling out” while describing observations. This could perhaps be replaced with terms like “deviate”?

The “naive model” requires a clearer background and explanation. What is the naive model? What are the expectations and how the diverging findings can be reconciled with the naive model can be explained in more detail.

We thank the reviewer for pointing us to this potential misunderstanding. The section itself is outlining a central finding that is the gateway to understanding the ‘reset’ in the following section. The ‘naïve model’ is the first intuitive interpretation of the results, based on the idea that attentional saccades follow the ongoing attentional fluctuation (the “follow” interpretation). But what we show through this section and the next section (attention fluctuation reset) is that this interpretation fails to capture the data. Instead, we show that attentional saccades, i.e. particularly switches, interfere and interact with the ongoing attention strength oscillation.

Under the ‘follow interpretation’ switch and stay events are consecutively organized within the attention fluctuation. Switches are expected to follow switches at full cycle distance and stays

at half a cycle distance. As we outline in Figure 6a this assumption of consecutiveness would necessitate that the IEI histograms would display two peaks per oscillation because under the 'follow interpretation' the identity of the previous event matters. As reported in Figure 3 and 5, and also when we take the event history into account (Figure 6b,c), we do not find any evidence that the event history has an effect on IEIs.

We thoroughly rephrased the text on pp. 16-17 to improve the readability and to better display the significance of that section.

R1.1.24

We see in Figure 5G that there is a consistent phase offset between Switch and Stay events. However, it is unclear which results were used by authors to infer or derive the conclusion of Stay events clustering during the oscillation peaks and Switch events clustering during the oscillation troughs.

We thank the reviewer for pointing to this potential misunderstanding. This conclusion is based on three observations. First, the first peak latency defines the most probable time interval between the current and the previous event. Second, by definition, each event exceeds the attention strength threshold and therefore is a peak in itself. In an event sequence this is true for the current as well as the previous event. Third, events and in particular switches, appear to reset the underlying attentional fluctuation frequency (Fig. 7). Thus, between events, the oscillation is best described as a cosine with time zero set at the previous event. Peaks of this cosine oscillation would reoccur at full cycle distance and a trough would occur at half a cycle. Consequently, finding first peak latencies at around 0.5 cycle length for switches is in line with these events occurring predominantly at the trough of the 11 Hz attention oscillation. Peak latencies at around a full cycle after the previous event for stays corresponds to the peak of the attention oscillation.

Fig. R6 | Schematic relationship between the attention strength fluctuation ~ 11 Hz (top) and the event probability distribution for stay (middle) and switch (bottom) relative to the previous event. The schematic is building on the results of Fig. 3, 5-7 and the schematics in Fig. 5h & 7f.

The rhythmicity of the event distributions is thus a consequence of rhythmic attention: Switch events predominantly cluster around the trough of the 11 Hz attention oscillation. Stay events cluster around the attention oscillation peaks. We revised the manuscript to clarify the aspects of rhythmicity in the results (p. 15-16), Figure 5i and discussion sections (p. 26).

R1.1.25

Result in Page 20: "Given the presence of a higher frequency attentional oscillation prior to switching, we next asked whether faster or slower attention strength fluctuations predict the occurrence of switch or stay events. We thus correlated the attention strength fluctuation amplitude in each frequency range (compare with Figure 3E) with the event probability over the course of the trial (Figure S5). A stronger 11 Hz attention modulation

predicted more stay events (Figure 7E, orange line, $pFDR < 0.08$) and a stronger modulation between 19 - 40 Hz more switch events (Figure 7E, green line, $pFDR < 0.05$).” There is once again a disproportionately strong conclusion drawn for 11 Hz versus a very broad band of 19-40 Hz. Results should appropriately characterize these claims. Again, the significance of 11 Hz versus any adjacent frequency should be explained in the discussion, given the claims made.

We thank the reviewer and as pointed out above agree. We comprehensively revised the section (p. 20) and explicitly point to the significantly tested frequencies.

R1.1.26

“Hence, faster attention strength dynamics prior to switching the locus of attention generally corresponded to the appearance of high-frequency modulations of attention strength throughout the trial.”

That is, (faster) attention strength dynamics...corresponded to...modulations of attention strength. This statement appears circular and could be rephrased better.

We thank the reviewer and rephrased the sentence in the revised manuscript (p. 20):

“Hence, faster attention strength dynamics corresponded to an increased probability to re-allocate attention (switching).”

R1.1.27

Result in the section “Dynamics of covert attention can be dissociated from (micro)saccadic eye movements” (Pages 23-24)

A) “...we found that attention switches are positively correlated with the MS rate ($r = 0.06$; $p = 0.02$).” Is a correlation coefficient of $r = 0.06$ a robust correlation? In fact, the result

here indicates there is no significant correlation between MS rate over trial and attentional event probabilities.

B) Similarly, in “Conversely, attention switches were negatively correlated with the change in pupil size ($r_{\text{mean}} = -0.07$, $p = 0.01$; Figure S9C)...”, the correlation coefficient of $r = -0.07$ indicates that there is no meaningful negative correlation between attention switches and changes in pupil size. The inferences should be re-evaluated and updated.

We agree with the reviewer that these are critical points that need clarification. The reported correlation A) of the trial-averaged MS rate (Fig. 8A) with the event-probability (Supplementary Fig. S6) was calculated within each participant. The reported correlation ($r = 0.06$) is the average over all participants and the p-value is based on the t-test ($df = 19$) against zero. The result displays thus a small but significant correlation on the group level ($t_{19} = 3.23$, $p = 0.005$). We added the statistical testing procedure to the revised manuscript.

We applied the same approach to the relationship B) between the pupil size and event distribution (S11C) and identified small but significant correlations between pupil diameter and stay probability ($r_{\text{mean}} = 0.10$, $t_{19} = 3.26$, $p = 0.004$) as well as between pupil size change and switch probability ($r_{\text{mean}} = -0.07$, $t_{19} = -2.87$, $p = 0.01$).

R1.1.27

“...we found that (micro)saccades preceded switching events (mean = 35%, range = 25 – 50%) more often than staying events (mean = 26%, range = 17 – 38%; $t_{19} = -13.2$, $p < 10^{-9}$). Taken together, oculomotor activity appears to play a role in switching attention between decision alternatives but not when staying focused on the same alternative.”

Switching involves planning and preparation for eye movements. Is the finding in the above statement an unexpected or novel result? The direction of the microsaccades could be informative -- for example, if the microsaccades are highly directed toward the next

alternative even prior to switching, the finding above is not a surprising correlation, but predictive of the chosen alternative. If non-directional, the microsaccades might indicate a general "dislodge" from the attended location before the actual voluntary "reorienting" and subsequent "anchoring" to the next alternative.

We thank the reviewer for the insightful comment and generally agree with the assessment that further analyzing the (micro)saccade direction might increase our understanding of the interactions between oculomotor activity and covert attention (see for example also Liu, Nobre, & van Ede, 2022, NatComm, Link). However, we feel that this analysis is beyond the scope of the presented manuscript.

Our motivation to include (micro)saccades and pupil measures was to control for their effects on the identified dynamics of covert attention. Covertly switching attention and oculomotor activity display a relationship (Fig. 8c-f), which is expected from a growing body of literature. However, the rhythmicity appeared to be unaltered by the occurrence of (micro)saccades, which is a novel finding.

We revised the manuscript to better motivate the analyses presented in Figure 8 (p. 21).

R1.1.28

"Recent literature highlighted pupil size as a peripheral measure associated with switching attention." Authors could elaborate on the direction of modulation observed in the cited recent literature here (or in the introduction), i.e., were there reports of an increase or decrease in size with switching? Does the result in this analysis support or contradict the existing literature? The motivation for this analysis and significance could be explained in further details.

We thank the reviewer for the attentive reading. The literature highlights the change in pupil size as a marker for increases in overt re-allocation of attention through (micro)saccades (Wang & Munoz, 2021, EJM, Link). The motivation for the analyses presented in this paragraph and the

related Supplementary Figure is to shed further light on the relationship between our measure of covert attention and existing markers of overt attention. However, the pupillometric results rather highlight an increase in pupil diameter during periods of strong attentional focus (stay events), which we interpret as increased arousal (compare Nassar et al., 2012, NatNeuro, Link). We revised the section to motivate and clarify the outcome of this control analysis. Please note that we moved the analyses on pupillometric measures to the Supplement in the revised manuscript.

R1.1.29

Discussion: (Page 27): “The reset might serve two purposes. First, it could correspond to an internal go cue[54], indicating readiness of the system to change[20] and briefly enhancing processing power[27,55].” Could this be elaborated? Why would readiness to change be expected after a switch (instead of before the switch)?

We thank the reviewer for pointing towards this misunderstanding. We do not argue that the reset is occurring after the switch but around the same time. We revised the manuscript accordingly.

R1.1.30

Similarly for (Page 28): “These sampling dynamics challenge extant models of choice, which assume that value information is optimally read-out by downstream areas implementing comparison between alternatives.” This statement could be expanded in further details. What do the extant models suggest about temporal dynamics of sampling and comparison downstream? How are the current results challenging these extant models?

We agree with the reviewer. Our findings point towards the conclusion that not only the ‘where’ (Fig. 5c) of attention and perception during decision-making is important but also the ‘when’

(compare also Fig. 4b). This challenges prevalent models integrating attention as time-independent variable into the sequential decision-making process (compare for example Krajbich, 2019, *Curr Opin Psychol*, Link). We explicated the shortcomings of existing models of decision-making incorporating attention in our conclusion as well (p. 29):

“These dynamics challenge extant sequential sampling models of choice, which assume stable and parallel throughput of value information to downstream accumulator representations.”

R1.1.31

Methods (Page 30): “Participants had 2 seconds to indicate their decision with a left thumb, right thumb or foot button press for the left, right or top alternative, respectively.”

Can a foot make dexterous motions like pressing a button, matching an individual’s thumbs? If not, how did this design affect the response time or accuracy for the top alternative? Why was foot used instead of another (e.g., index) finger? Furthermore, I could not find the participants’ handedness reports. Were the left and right thumb associations with the left and right alternatives counterbalanced? How are these possible motor biases accounted for in the behavior?

We thank the reviewer for pointing to the incomplete description. All participants were right-handed. We used the fixed response mapping – left thumb, right thumb, left/right foot – to maximize the decodability of the response (results not shown and in preparation). The foot pedal was mounted on Styrofoam. We adjusted the height and location of the pedal for each participant such that the position was comfortable and a response could be triggered with a small flexion of the porcellus fori. With the individual adjustments to the foot pedal placement, we tried to make the motions between effectors as compatible as possible. Median response times (RT) were slightly but not significantly slower for foot responses ($RT_{\text{foot}} = 0.46 \text{ sec} \pm 0.18 \text{ sec}$; $RT_{\text{left}} = 0.40$

sec \pm 0.18 sec; RT_{right} = 0.39 sec \pm 0.16 sec, each mean \pm standard deviation; paired sign test p_{foot,left} = 0.06; p_{foot,right} = 0.08). We revised the Methods section accordingly (pp. 31-32).

Minor concerns:

R1.2.1

Page 2: The use of words like “universal” (in “universal cognitive mechanism”) should be justified or appropriately rephrased to match findings.

We rephrased the abstract to “an adaptive cognitive mechanism”.

R1.2.2

Page 3: “Spontaneous” seems to contradict the notion of serial execution. Should be removed or replaced appropriately.

We revised the entire introduction. However, the term in the previous version referred to the dynamics and the temporal evolution, which even in serial execution can be spontaneous at a given time point.

R1.2.3

Page 3: Potentially incorrect reference citation numbers in “Moreover, the deployed non-invasive markers of attention, namely (micro)saccades and lateralization of posterior alpha activity, might display only an incomplete picture of covert attention and its dynamics[38,69-71].” The preceding citations are numbered at [9-11] (first line in the same paragraph).

We thank the reviewer for the attentive reading and corrected the reference numbering in the revised manuscript.

R1.2.4

Page 4: There is a word (“and”) omitted. The statement should say, “Our approach showed that during complex decisions, the strength of covert attention and decision formation oscillates at 11 Hz.”

Thank you for pointing this out, we revised the manuscript accordingly.

R1.2.5

Page 10: Please add the missing figure reference for Figure 4B.

We added the Figure reference to the text.

Reviewer #2 (Remarks to the Author):

This MEG study investigates whether the attentional focus oscillates. First, in a localiser run, human participants see each of three single gabor patches with different contrasts and locations, used to train an inverted encoding model. This model is then tested on the main task where participants see all three gabor patches and are then cued to respond to which has the highest or lowest contrast. The success at decoding the correct stimulus is thresholded to give a modelled measure of when participants are fully attending to any particular stimuli and of overall attentional strength. This decoded attentional measure is then used to generate the following list of results: Attention strength increased most in the range 8-22 Hz and decreased between 25-45 Hz. Attention strength correlated with neuronal single amplitude envelope in the alpha and gamma bands especially over occipital and dorsolateral frontal sensors. A separate analysis of these attentional events shows that participants tended to stay at the same location or switch to another location with alternating phases matching the 11 Hz frequency reported previously, but with stay and switch events half a cycle apart. Some dissociations between switch and stay events are observed: Attentional strength was weaker around switching than stay events. Attentional strength varied in phase after either event but was not aligned before events. Attentional strength fluctuated at 22 Hz before switch events only. 11 Hz signals predicted stay, whereas higher frequencies predicted switch. Eye movements did not show such 11 Hz effects, rather were present at a lower frequency. There were more eye movements after attentional events and before switch events. Attentional events were less likely in the 0.2 s after microsaccades. A pupil analysis does not show that pupils were bigger on switch trials but if anything the opposite.

There is an interesting literature providing evidence in primate and human that attention operates rhythmically. The novel contribution here is that the analysis of oscillatory behaviour is not applied to behaviour or neural activity directly but rather to the parameters of attentional strength and locus which are generated through decoding. This comes across as a noteworthy and original development that breaks new ground while itself being well founded. It is a fertile approach in that it immediately enables the large swathe of analyses included here. Across many analyses a general common pattern does emerge, highlighting the presence of 11 Hz oscillations in attentional shifts. In general this is a striking and interesting set of results. The major concerns I have regard how the results are interpreted.

We thank the reviewer for their in-depth assessment of our manuscript and the constructive comments given to improve the clarity and reproducibility of the reported findings. We will address all of the reviewer's comments – point by point – below.

Major concerns:

R2.1.1

To prevent the results being purely driven by bottom-up attention, participants do not always simply respond to the highest contrast stimulus but rather are cued afterwards whether to respond to the highest or lowest contrast stimulus. However both these two trial types - high and low contrast targets - are collapsed together. Is there evidence for rhythmic attention when looking at the low contrast targets only? If not then the results may purely be revealing the operation of bottom-up orienting mechanisms on the high contrast trials which survive averaging together with noise on the low contrast target trials.

We thank the reviewer for this important comment. We agree that in principle more salient high-contrast framing trials might display a stronger rhythmicity than low-framing trials. We thus

separated the data for the analysis of rhythmicity (compare Fig. 5e-g) into high- and low-framing trials:

Fig. R5 | Inter-event-interval histograms separately for high and low framing trials.

Uncorrected inter-event-interval histograms separately for **a** stay and **b** switch attentional events in High- (colored solid line) and Low-Framing conditions (colored dashed line; compare Fig. 5). The black dashed line indicates event intervals under random conditions.

We do not find evidence that bottom-up salience during high contrast trials drives the reported dynamics. In both framing conditions we can identify rhythmicity and do also not find any evidence for dissociable inter-event-intervals.

Sparked by the comment from another reviewer we would like to further point out that the decodability of the highest (decision) valued (high contrast in high framing, low contrast in low framing) and the lowest (decision) valued options (low contrast in high framing, high contrast in low framing) throughout the trial do not differ between the framing conditions (see Fig. R1). We thus conclude that low-level features of the stimuli cannot explain the reported results on covert attention dynamics.

Fig. R3 | Decodability (estimated response) differences between High- and Low-Framing trials separately for **a** high, **b** medium, and **c** low value option within each respective trial. The circles schematically depict the compared contrast levels. The figure has been added to the revised manuscript.

We appended both Figures (R1 & R3) into the Supplement and thoroughly revised the manuscript to better explicate the influence of low-level features on the reported results.

R2.1.2

I do not follow the argument for the key “resetting” claim here e.g. that “attentional switching resets rhythmic attention strength fluctuations.” The current argument for that includes at the moment statements like on pg 20: “Overall, our findings suggest that in particular switching events ‘reset’ the ongoing attention oscillation to the attentional up-state (the focused mode): post-event activity is synced between event types restarting the dominant 11 Hz (Figure 7A and 7B) and attention strength peaks at around 0.09 s and 0.18 s post-event, i.e. approximately after one and two full cycles.” Why is this suggesting resetting after switch events? I would imagine that evidence for switch-specific resetting would be showing that the phase of the oscillation is always the same after switch events but not stay events - please clarify where this is present in the results.

We thank the reviewer for pointing us to the lack of clarity. We revised the manuscript to clarify two points.

First, the reset following switching is predominantly a frequency reset and not necessarily a phase-reset: the frequency of the attention strength fluctuation changes after each attention switch (Fig. 7a-d). A reset is tuning the system back to the default state, which in our case is the 11Hz oscillation in attention strength (Fig. 7d). The findings presented in Fig. 7 combined with the findings reported in Fig. 5 and 6 all point to the frequency reset that we propose, specifically: (i) the offset between switch and stay events (Fig. 5e & h), (ii) the lack of event history effects (Fig. 6b,c), (iii) the change in attention strength frequency from before to after switches (Fig. 7d), and (iv) the similarity of post-event activity (Fig. 7a,b).

Second, we have realized that our statement that post-event activity is ‘synced’ between switch and stay events was misleading. The empirical finding is that the default state of the oscillation, starting at switch events, is an 11 Hz cosine with time zero at the event (Fig. 7a,b, green line). This starting point of the post-switch oscillation coincides with the starting point around the average stay event. We cannot and do not claim that this observation reflects an active synchronization. Correspondingly, we have dropped the above statement from the manuscript.

R2.1.3

Before switch events two things happened compared to stay events. Attentional strength fluctuated at 22 Hz, and there were more microsaccades. How are these related, could one be causing the other?

We thank the reviewer for the insightful comment but we argue that the results displayed in Figure 8d speak against this hypothesis. Figure 8d displays the attention strength before and after a (micro)saccade. If the high frequency attention strength modulation would generally be related to oculomotor activity, we would also expect a 22 Hz modulation around (micro)saccade onset. But,

particularly at around 0.15 – 0.2 seconds after a (micro)saccade, when the appearance of switches is predominantly triggered (Fig. 8c), we only observe a single large increase in attention strength and no evident fluctuation. Moreover, through the entire depicted period we do not find consistent fluctuations.

R2.1.4

The “attention strength” is an extremely global measure. What more precisely is meant by this, how does it relate to other attentional terms in the literature? Why hypothesise that “that attention strength (i.e., vector length) would vary throughout the course of the trial as a result of the distinct processing demands in the sensory- and decision-phases of the task.” (Pg 7). If attention is defined in a very global way, potentially encompassing many processes, why is it considered to vary during the task? Is the idea that this “attention” is equivalent to arousal and should vary during the task? Or that this “attention” reflects specific processes within the “sensory- and decision-phases of the task”. Why would we expect “attention” to peak three times close to stimulus onset, then again early and then later in the decision phase?

We thank the reviewer for highlighting this open definition. Generally, attention strength indicates how well the decision alternatives are represented, i.e. decodable (compare p. 6). In principle, high attention strength might thus signal increased sensory processing, task engagement or arousal. We updated the manuscript and the definition of attention strength (p. 6) accordingly.

R2.1.5

It is confusing in the manuscript whether stay and switch are both attentional events with their own (orthogonal) phase - the impression received from the results section - or

whether are they part of one ongoing oscillation - as they are interpreted in the discussion?

Are there two cycles for staying and switching or one?

We apologize that our previous use of the terms for both event types was unclear and inconsistent. In short, our data indicate that both event types are part (at different phases) of the ongoing oscillation of attention strength. This is illustrated schematically below:

Fig. R6 | Schematic relationship between the attention strength fluctuation ~11 Hz (top) and the event probability distribution for stay (middle) and switch (bottom) relative to the previous event. The schematic is building on the results of Fig. 3, 5-7 and the schematics in Fig. 5i & 7f.

Our findings suggest that the time-varying probabilities of switch and stay events (Fig. 5) – simplified as sinusoids for illustration purposes– result from the attention strength fluctuation (Fig. 3). The peaks of these event probabilities (mid and bottom panels in Fig. R6) occur at distinct phases of the attention strength oscillations, namely at the peak and trough for stay and switch (Fig. 6,7), respectively.

The event-related attention fluctuations are particularly strong at 11 Hz and this frequency appears to define the temporal frame for event occurrences and their temporal coordination.

However, we do not argue that there is only one specific attention oscillation frequency plus noise. For example, we find that the attention strength signal displays strong power for a broad range of frequencies from 8 – 32 Hz (see Fig. 3d). Consequently, different attention oscillation frequencies might be relevant for distinct functions of attention much like distinct cognitions display relations to specific neuronal frequency channels (see Siegel, Donner & Engel, 2012, NatRevNeuro Link). The finding that pre-event attention strength fluctuations display diverging frequency content (Figure 7a,b,d) and that switch occurrences are also affected by attention fluctuations above 11 Hz (Fig. 7e) are novel findings. Their interpretation is speculative at this point but hints towards a multi-spectral interpretation of attentional fluctuations (see also Michel, Dugue & Busch, 2021, EJM, Link; Fiebelkorn & Kastner, 2019, TiCS, Link). We argue that stay and switch events are indicative of two attention functions (or modes), namely focused processing ('in-depth evaluation', stay) and the re-allocation of the focus ('comparison', switch). Each of the two modes is related to distinct neuronal substrates and functional networks (see also Supplementary Fig. 10). Thus, while 'in-depth evaluation' (stay) and 'comparing alternatives' (switch) can be described as distinct (attentional) functions/modes with potentially unique spectral features, their interaction appears to be coordinated at around 11 Hz. This is an important distinction that we clarified in the revised manuscript.

R2.1.6

Please reconcile the descriptions of these in results and discussion.

We have revised the manuscript in order to clarify the points raised above. We explicated how both attentional modes might be confined within the attentional oscillation in the Intro (p. 2), Results (pp. 14-15) and Discussion (p. 26). The Intro for example now reads:

p. 2: “[W]hen attention is at an oscillatory peak, processing of a selected stimulus is enhanced [source]. By contrast, when at an oscillatory trough, attention is relatively disengaged and might be shifted from one stimulus to another [source]. Distinct attentional

modes supporting different functions can be thus confined in one cycle of this oscillation [source].”

Further, we added intuitions and interpretations of the multi-spectral nature of the attention signal in the Results (pp. 20-21) and Discussion (p. 27). For example:

p. 27: “Specifically, the frequency of the attentional oscillation is higher prior to switching (Fig. 7), indicating unique spectral features for both attentional modes. Hereby, ‘in-depth evaluation’ (stay, ~11 Hz) and the ‘comparison of alternatives’ (switch, ~22 Hz) might for example be coordinated through cross-frequency interactions of distinct attentional oscillations [source]. With a switch event, the frequency of the attentional oscillation is reset to the dominant ~11 HZ, signaling a shift from a disengaged to a focused mode of processing.”

R2.1.7

Why is it interpreted that “oculomotor activity itself did not display rhythmicity” (pg 27) - the impression I have from the oculomotor data here is that there was clear rhythmicity but was just dissociable from that of the attentional strength. What was the ocular rhythmicity - reflecting three serial saccades to the stimuli?

This is indeed a very important dissociation. We find a prominent increase in (micro)saccade probability around 0.25 sec post-saccade (Fig. 8b). As the reviewer pointed out serial saccades each ~0.25 sec apart might be triggered and this could appear like rhythmicity. We however argue that this deterministic relationship does not suffice as an intrinsic oscillation. If in the serial saccade scenario, the next saccade after another 0.25 sec does not occur, the entire temporal relationship breaks down. In a persistently ongoing oscillation, we would expect that even when a saccade isn’t triggered in a cycle there should still be an increased chance for a saccade at the same phase in the next cycle. This relationship beyond direct cycle-to-cycle serials would be

captured as multiple peaks in the ISI histogram. The temporal structure identified for (micro)saccades is thus qualitatively different to the attention events. We revised the manuscript to clarify this dissociation (see p. 22).

R2.1.8

Why were effects not apparent in the theta range, which is usually highlighted in other studies of attentional rhythmicity such as by Kastner, or Van Rullen.

Several studies in the past have reported attentional rhythmicity at around 4-8 cycles per second and related neural activity in the theta range. However, findings for rhythmic sampling at higher frequencies at around 7-12 Hz are common as well (compare Kienitz, Schmid & Dugue, 2021, EJN, Link; Van Rullen, 2016, TiCS, Link). There might be several reasons for a higher sampling rate in our study potentially resulting from the task setup. Our protocol consisted of a three-alternative protracted decision-making task with peripheral stimuli. First, one speculative interpretation highlighted in the literature is that peripheral stimuli elicit a higher sampling frequency than centrally presented stimuli (van Rullen, 2016, TiCS). Second, attention re-allocation in a protracted decision-making task might rely on different goal-oriented cognitive strategies than target-detection or simple visual search. Studies with target detection and visual search protocols often display rhythmicity around 4-8Hz (Kienitz, Schmid, & Dugue, 2021, EJN; Fiebelkorn & Kastner, 2019, TiCS, Link; Senoussi, et al., 2019, J Vis, Link), whereas protocols relying on more extensive processing of attended stimuli (compare for example visual recognition: Caplette, Jerbi, & Gosselin, 2023, JNeurosci, Link) as well as graded response reports (compare Michel, Dugue, & Busch, 2020, EJN, Link) can display sampling at higher rates. It appears that further increasing the stimulus and task complexity can impact the sampling frequency (compare for example Merholz et al., 2022, SciRep, Link; Gaillard et al., 2020, NatComm, Link).

We agree with all the reviewers that the details of the rhythmic attention literature and its implications for our task and findings need to be discussed in more detail. We thus broadly revised the Introduction and Discussion sections to address this field of research in more depth.

R2.1.9

I do not understand the “caveat” on pg 16: Why is that if stay events are more likely, why would this have an effect on the first peak offset between stay and switch events? Most of the first page of introduction is embedding this within a decision making context, this seems unnecessary and the space might instead be used to address the above.

We agree with the reviewer that the section on “Brief attentional saccades impact the ongoing attention strength fluctuation” is very dense and could lead to misunderstandings. We thoroughly revised the entire section to clarify the problem/caveat addressed here and in Figure 6 (see pp. 16-17).

Minor concerns:

R2.2.2

As I understand it the effect shown if Fig 4 A was not statistically significant, this panel should then either be removed or made clear in the legend that its not significant.

We explicated the statistical significance, i.e. the lack thereof, further in the paragraph on p. 10.

R2.2.3

Do the authors think that the same results could be measured with EEG, albeit without the spatial resolution

Speculatively, we would argue that the results should be reproducible with EEG as well. We have found in previous studies that EEG and MEG can grant compatible insights into large-scale neuronal dynamics (for example functional coupling: Siems et al., 2016, NImg, Link).

R2.2.4

Pg 15: I don't get the interpretation of the stay data: why would "more sustained "stay rate" early-on in the decision can be mapped to the independent evaluation of the different alternatives" - why is staying for longer related to processing different alternatives independently?

The highlighted statement is an interpretation of the identified attention allocation dynamics in the framework of neuroeconomic decision computations. In this framework a decision is a serial process in which each option is first valued before these values are compared (see for example Platt & Plassmann, 2014, Neuroeconomics, Chapter 13, Link). Under the assumption that stay events represent valuation and switches comparisons the results in Figure S6 follow the rationale in this framework. However, we agree that this is a top-level interpretation and moved the argument from the Results to the Discussion section.

R2.2.5

Why more microsaccades after events, or before switch events? Please tone down the causal relationship implied here: “Indeed, we showed that fixational eye movements and microsaccades had an effect on switching attention between Alternatives” pg 27

We agree and revised the manuscript accordingly.

R2.2.6

More details necessary pg 24 for this analysis: “We identified post-saccadic rhythmicity (first-to-second peak latency) in 15 out of 20 participants for both event types with no detectable change in oscillation frequency”

The analysis follows the same criteria as the original first-to-second peak latency analysis reported in Figure 5 (p. 14 and p. 37). We can only define rhythmicity if we identify at least two peaks in a participant’s corrected IEI histogram (compare Fig. 5e). The comparison in oscillation frequency was then computed between IEI derived rhythmicity between no-MS (Fig. 8g-h solid line) and post-MS (dashed line) conditions for switch and stay separately. We have added this information in the revised manuscript (p. 22).

R2.2.7

Pg 30: Why are there 10 possible combinations if three stimuli are drawn without replacement from a set of 5 linearly spaced contrast levels

The sets of possible stimulus combinations are here reported independent of their ordering to the three locations. If we include the exact order to the locations this amounts to 60 combinations. Below is the overview of all contrast combination sets without separating by location:

Fig. R7 | Full set of possible contrast combinations, one per column. Darker colors indicate stronger contrast.

R2.2.8

Pg 30: what was the purpose of the experimental manipulation whereby the three gabor patch orientations either all had the same or different orientations? Were there other analyses here not reported in the manuscript?

We thank the reviewer for the attentive comment. Indeed, the orientation manipulation was not leveraged in the presented manuscript. The rationale behind the manipulation was 1) to include another feature besides location to better decode each option and 2) be able to address decoy effects, where objects with similar appearance are compared more frequently within a trial (see for example Noguchi & Stewart, 2018, Psy Rev, Link). However, we did not find any decision related effects of orientation (results not shown) and were well able to decode the stimuli independent of their orientation. We thus pooled all trials together for the reported analyses. We revised the Methods section accordingly (p. 31).

R2.2.9

I don't understand what is going on in this sentence; "To visually guide the participants we tilted the entire stimulus array 60 clockwise or counter-clockwise when participants were using their right or left foot, respectively." When was this tilting happening?

The tilting did not change on a trial-by-trial basis but it was fixed throughout the experiment for a given participant. Each participant always saw one of the two possible tilted stimulus sets, depending on which foot they were instructed to use in the experiment (at 6°, 126°, 246° for right foot, -6°, 114°, 234° for left foot). Foot side was randomized between participants and did not change throughout the experiment for each participant (compare p. 31). We revised the description in the Methods section for clarity.

R2.2.10

Pg 39: give the most important parameters for the blink and saccades definition beyond just saying "default settings" of the software.

We agree with the reviewer and included a more detailed description of (micro)saccade definition in the revised manuscript (p. 42). Blinks were defined through manual inspection of the raw signals further described on p. 33.

R2.2.11

Why are saccades here defined as including "microsaccades" (e.g. pg 40): what was the sensitivity of the eyetracker, was it even capable of measuring microsaccades?

We thank the reviewer for the attentive reading. Generally, the definition of oculomotor activity being a saccade or a microsaccades includes both velocity and a threshold on the overall gaze

displacement (below 1° for microsaccades). We here derived oculomotor events solely based on velocity. Our approach is agnostic to this overall displacement and we thus chose the wording (micro)saccade and revised the Methods section accordingly. The spatial resolution of the EyeLink 1000 is <math><0.01^\circ</math> and the average position accuracy was between 0.15-0.3°. We appended further details to the eyetracker as well as the saccade detection algorithm to the Methods section (p. 42).

R2.2.12-15

- Pg 7 “channels” > “channel’s”

- Pg 32 comma after eye-blinks, formatting of wavelet bandwidth, “participants” > “partcipant’s”

- Reference 51: check if this is the right citation, is divisive normalisation the correct reference for rhythmic attentional gain?

- Reference “79” is missing on page 34

Based on the points raised above we revised accordingly.

R2.2.16

“Micromechanisms” needs defining/qualifying on pg 25

We dropped the prefix “micro” in the revised manuscript.

Reviewer #3 (Remarks to the Author):

The study utilized MEG recordings to track attention during an alternative decision task. Presumably, both the strength and locus of covert attention were extracted using data-driven analyses tools to examine how attention fluctuates during decision-making with multiple options. The approach is based on the idea that different phases of brain waves, such as peaks and troughs, are linked to distinct mental processes or states, such as maintaining focus or shifting attention. The findings reveal that the strength of attention oscillates rhythmically at around 11 Hz, influencing where attention is allocated. Attention is more likely to switch location at the trough of the 11Hz oscillation and at the peak of the oscillation, attention is more likely to stay at the previously sampled location. Further, findings show that prior to switching location, attention strength starts to oscillate at a higher frequency, thereby predicting the switch. Notably, switching attention resets the 11Hz rhythmic oscillation. The authors finally show that oculomotor behavior in terms of micro-saccades did not show rhythmic behavior and was not exclusively associated with covert information sampling. The study puts forward very interesting and novel insights into covert information sampling during decision-making. However, we have some major concerns related to the methodology and interpretation of the results that we believe must be addressed and clarified prior to any publication.

We thank the reviewer for the insightful and very detailed assessment of our manuscript. We will address all of the raised concerns and potential ambiguities point-by-point in the following.

Major concerns:

R3.1.1

The authors claim that different aspects of covert top-down attention are tracked in time. In other words, the inverted encoding model must then have identified MEG activity patterns related to covert attention orientation. Since this model has been trained on data from the sensory localizer task in which Gabor patches were displayed while subjects maintained their attention on the fixation cross (i.e. did not orient attention), it does not seem possible. It is plausible that the sensory localizer task involves exogenous/bottom-up attentional shifts and perceptual processing of the Gabor patches, and hence this is more likely what the authors are decoding. Indeed, previous reports have shown rhythmic modulation of perception (vanRullen 2016, 2018) which might be in line with what the current study is showing. We believe the results remain novel and relevant, however, the authors need to clarify the context of the sensory localizer task and the impact it has on the interpretation of the decoding. Also, the decoding of covert attention needs to be justified.

We thank the reviewer for this important comment. The nature of the sensory localizer task indeed comprises an exogenously driven bottom-up capture of peripherally presented stimuli. However, we argue that the (sensory localizer) task design in and of itself does not isolate bottom-up processes but is further sensitive to higher-order processing of the Gabor stimuli during the main task. We base this argument on a set of new analyses and insights from the literature. Reviewer #1 commented very similarly. Our reply to both is identical and we thus paste the text for convenience here as well:

The inverted encoding model (IEM) is a method for reconstructing population-level stimulus representations from aggregate neural activity data. We thus agree that the approach is generally different from mapping out an attention-based topography where the contribution of visual stimulation could be partialled out almost completely (see for example Silver, Ress, & Heeger, 2005, *J Neurophysiol.*, Link). Further, we followed the assumption that the stimulus representation would be stronger for attended than for unattended stimuli (compare for example Buschman & Kastner, 2015, *Neuron*, Link; Posner, 2016, *Quart. J. Exp. Psy.*, Link). However, we do not argue that the estimated responses are pure ‘attention signals’ but could as well be a compound signal of attentionally modulated decision input.

Furthermore, our choice of a sensory localizer was motivated by the goal to eliminate the effects of eye movements as much as possible. Recent literature has shown that cued covert attention protocols display strong contaminations with (micro-)saccadic eye movements (see for example Hafed, Lovejoy, & Krauzlis, *J Neurosci*, Link; Liu, Nobre, & van Ede, *NatComm*, Link), which can potentially affect stimulus processing and their representations (see for example Hafed, Chen & Tian, 2015, *FiSN*, Link). This relationship has largely been overlooked. Based on these findings a localizer including spatially cued targets would thus be contaminated with eye-movements or movement preparation artifacts. Consequently, decoded attention allocation and its dynamics would be intertwined with oculomotor dynamics.

For the revision we further quantified whether low-level features, i.e. contrast, lead to differences in stimulus decodability, i.e. estimated responses. We ask whether the estimated response is different within each decision value class (high, medium, low) between the two framing conditions. For example, in the “high value” class, we compare the decodability of the highest contrast (high framing) with the lowest contrast (low framing). Thus, the low-level (bottom-up feature differs but the decision value is the same. If contrast was the driving force behind the estimated response we would observe a significant difference between framing conditions within each decision value:

Fig. R3 | Decodability (estimated response) differences between High- and Low-Framing trials separately for **a** high, **b** medium, and **c** low value option within each respective trial. The circles schematically depict the compared contrast levels. The figure has been added to the revised manuscript.

We find a significant difference ($p < 0.01$) right after the stimulus comes onto screen (-1 seconds) but not in the decision-relevant stages of the trial (i.e., after the framing cue is presented). Thus, low-level features of the stimulus appear to affect only the onset response. We thus conclude that the IEM in combination with a bottom-up driven sensory localizer task is sensitive to top-down evoked signals. We added the above analysis to the revised manuscript and included the Figure in the Supplement (S4).

However, we do not argue that the estimated responses of the IEM decoder are a pure 'attention signal' but could as well be a compound signal of attention, arousal, task-engagement, sensory processing and decision-making components. We thus revised the manuscript to clarify the potential pitfalls and caveats in the interpretation of the decoded attention signal (compare for example p. 6).

R3.1.2

Much of the novelty of the study relies on the link between attention and decision-making processes. However, what makes this study exceptional in this regard? Is it the use of a multiple-choice task (instead of e.g. a detection task used in many previous studies)? To make a credible statement on the relationship between attention and decision-making processes, the tracking of attention should be correlated with decision behavior during the decision task. The authors show that attention strength is correlated to performance, how about the correlation to the decision (independent of performance)? How is attention strength and locus correlated to the decision?

Generally, we agree with the reviewer that a detection of a target describes a decision. In principle, every paradigm that requires an overt response might do. It is thus important to clarify what is special about the task we employed. Our claim is that our task emulates more closely real-life multialternative decisions (e.g., consumer choices) as it entails high-level valuation of options (as a function of the variable task framing) and comparison between options. The need to compare between options in our task departs from the processing demands in a classic target detection or visual search paradigm. Further, using three alternatives is another important departure from simpler perceptual choice tasks, such as a 2AFC random dot discrimination task, where decision-relevant information is presented centrally, and spatial attention is not as pivotal. By contrast, in our task spatial attention is necessary to assign value and to compare options, which helps ground these crucial processes onto the dynamics of information sampling (equivalently, of attentional allocation). We extensively revised the Introduction and Discussion sections to better elucidate the full scope and rationale of the presented manuscript.

We thank the reviewer for the point regarding the functional role of attentional fluctuations. We agree that it is extremely important to establish that these fluctuations can be related to the decisions participants made. Before elaborating on this point, we would like to note that we defer

describing in detail the higher-level information sampling strategies employed in this paradigm (and their impact on decision behavior) for a different manuscript. Instead, due to space constraints, here we exclusively focus on describing the dynamics of attentional allocation in detail. With regards to the locus of attention, we can identify critical periods throughout the decision phase that display attention allocation towards alternatives of different identity (for example from 0.25 to 0.75 seconds post framing, the high-value alternative is more frequently sampled):

Fig. R8 | Windowed time resolved attention allocation, i.e. switch event, **a** targets and **b** sources. The decision values of the represented options are depicted through green lines from dark to light for high to low values. Thick black bars indicate significant dissociations for any of the three value comparisons (ANOVA, $p_{FDR} < 0.05$).

These patterns of attentional allocation effectively allude to the algorithm through which humans make multialternative decisions. As mentioned above, interpreting these patterns of sampling at the algorithmic level is beyond the scope of the present manuscript. Nevertheless, we agree with the reviewer that it is important to relate attention and decisions in the present manuscript and present new analyses where we established a direct link between attentional events and choices. We used logistic regression to predict choices from event counts (Fig. R9). That way we identified for example that the higher the overall number of both stay and switch attentional saccades to the

highest-value option (i.e. the correct option) the more likely a correct choice becomes (both event types, $p < 0.05$):

Fig. R9 | Logistic regression predicting choice (H, M, L individually) from event counts (stay at, and switch to H, M, L). Bars denote the mean over participants, vertical lines the SEM.

This analysis establishes that attentional deflections in this task are functionally/behaviorally relevant. We added the analysis to the main text (p. 13) and Figure 5 as panel c.

R3.1.3

Overall, the manuscript lacks proper acknowledgement to previous literature on attentional rhythms. The introduction should contain a broader description of relevant studies on attentional rhythms and the results should be discussed and put in context to previous findings. There is quite a lot of previous work that has detected attentional sampling at lower frequencies, 8 Hz (e.g. Gaillard et al 2020 which you have cited). Also, previous work suggests that information sampling across hemispheres occurs at a Theta rhythm (Fiebelkorn 2013; vanRullen 2016). Since the Gabor patches were displayed across

hemispheres in the current study, how would you interpret your results and the lack of theta rhythm in your data? A more complete picture will strengthen the article and highlight the novelties.

As we have also pointed out in the previous comment, we agree that the focus of the Introduction can benefit from a redirection.

Several studies in the past have been reporting attentional rhythmicity at around 4-8 cycles per second and related neural activity in the theta range. However, findings for rhythmic sampling at higher frequencies at around 7-12 Hz are common as well (compare Kienitz, Schmid & Dugue, 2021, EJM, Link; Van Rullen, 2016, TiCS, Link). There might be several reasons for a higher sampling rate in our study potentially resulting from the task setup. Our protocol consisted of a three-alternative protracted decision-making task with peripheral stimuli. First, one speculative interpretation highlighted in the literature is that peripheral stimuli elicit a higher sampling frequency than centrally presented stimuli (van Rullen, 2016, TiCS). Second, attention re-allocation in a protracted decision-making task might rely on different goal-oriented cognitive strategies than target-detection or simple visual search. Studies with target detection and visual search protocols often display rhythmicity around 4-8Hz (Kienitz, Schmid, & Dugue, 2021, EJM; Fiebelkorn & Kastner, 2019, TiCS, Link; Senoussi, et al., 2019, J Vis, Link), whereas protocols relying on more extensive processing of attended stimuli (compare for example visual recognition: Caplette, Jerbi, & Gosselin, 2023, JNeurosci, Link) as well as graded response reports (compare Michel, Dugue, & Busch, 2020, EJM, Link) can display sampling at higher rates. It appears that further increasing the stimulus and task complexity can impact the sampling frequency (compare for example Merholz et al., 2022, SciRep, Link; Gaillard et al., 2020, NatComm, Link).

We agree with all the reviewers that the details of the rhythmic attention literature and its implications for our task and findings need to be discussed in more detail. We thus broadly revised the Introduction and Discussion sections to address this field of research in more depth.

Minor concerns:

R3.2.1

In figure 1B. It is not clear what the different option values are (or decision values ?)? In particular, in the low framing condition, why are the participants choosing the high option values (high contrast Gabor patch)??

We thank the reviewer for the feedback and clarified this aspect in the main text (p. 5), figure legend (p. 6) and method section (p. 31). During high framing trials the highest contrast represents the highest decision value while during low framing trials the highest contrast Gabor has the lowest decision value. The framing manipulation decoupled the low-level features (i.e., contrast) from the task-relevance (decision value) of the choice alternatives. Choosing the alternative with the highest option value means choosing correctly given the framing within each trial. The categories H, M, L correspond to the highest, middle, and lowest decision value within each trial.

R3.2.2

Figure 2. The Design matrix is missing color coding, which hampers interpretation.

Thank you for the attentive reading. We fixed the panel in the revised manuscript.

R3.2.3

Page 7: The comparison between “Evoked pupil responses” and trial-averaged attentional vector dynamics (Figure 3B) needs clarification. Please specify which figures are being compared.

We thank the reviewer for pointing to this potential misunderstanding. The discussed correlation on the top of page 7 ($r_{\text{pupil,veclen}}$) refers to the comparison between Figure 3a and 3b. The ‘evoked pupil response’ hereby refers to the trial-averaged normalized pupil diameter. We revised the manuscript accordingly.

R3.2.4

Page 9: The paragraph about decoding the three alternatives lacks clarity regarding the mapping of Low, Mid, and High options to target positions. The relationship between Figures 2B and S3 is unclear, especially since Figure 2 does not mention High, Medium, and Low options, while Figure S3 does. Same in Figure 1B, it is not clear the context of H, M and L.

Thank you for pointing this out. Indeed, the estimated responses (Fig. 2b) are not mapped to their respective decision value (see also previous comments). We revised the manuscript to include the remapping of each response to decision value classes.

pp. 9-10: “[W]e assessed the dynamics of the estimated responses (Fig. 2b), the precursor to the attentional vector, and remapped each response to its option value class (High, Medium, Low).”

R3.2.5

Figure 3D is not clear. What does the color bar correspond to?

The colors depict the window onset of the spectrum from early (blue) to late (pink) windows. We added the explanation to the figure legend (pp. 8-9).

R3.2.6

Figure 4C; it is not clear which time period is used. The figure legend describes “stimulus on” but it doesn’t seem to fit. Do the authors mean “framing cue on”? Using a hyphen in “stimulus/framing cue-ON period” could enhance clarity by specifying that this period refers to when the stimulus is actively "ON."

We thank the reviewer for the attentive reading. The data in figure 3c is averaged over the entire stimulus presentation (Gabor patches) period (-1 s to 2.8 s). We revised the legend (p. 11) and text accordingly.

R3.2.7

At the beginning of page 16 there seems to be a mistake in the latency written in the text. They do not correspond to the figures. Should it be 0.085s? I.e. all latencies should be divided by 10 in this section? Or are you referring to latencies relative to cycle length? In that case, it should be clarified, and seconds should be omitted.

We thank the reviewer for this observation. The values were typos and we divided by 10 in the revised manuscript.

R3.2.8

Page 22: Rephrase for clarity: “ISI histograms showed a single peak around 0.25 seconds (Figure 8B), indicating that the latency for the first peak of covert attention events was significantly longer.”

Thank you for pointing to this potential misunderstanding. The oculomotor finding in Figure 8b (an ISI peak around 0.25 seconds) is compared to the attention events reported in Figure 5

(particularly 5e-f, earliest IEL peaks <0.1 seconds). We thus revised the manuscript to clarify this finding (p. 22).

R3.2.9

Page 23: Clarify that oculomotor activity may play a role in maintaining focus but is less pronounced compared to its role in switching attention. The statement “oculomotor activity appears to play a role in switching attention between decision alternatives but not when staying focused on the same alternative” could be misinterpreted.

We agree with the reviewer and rephrased this conclusion in the revised manuscript (p. 23).

R3.2.10

Page 38: Correct the reference to Figure S11 in the Methods section to Figure S10, as Figure S11 is not present.

Thank you, we revised the manuscript accordingly.

R3.2.11

There is no mention of figure 7C in the text.

Figure 7c was introduced at the end of the paragraph on p. 18. But we extracted the reference of the panel to independent parentheses for better visibility.

p. 18: “[...] Moreover, attention strength was generally weaker around switching- compared to stay events (Fig. 7c) both pre- (mean ratio_{Switch/Stay} = 0.90, range_{25%-75%} = 0.85 – 0.94; $t_{19} = -8.09$, $p < 10^{-6}$) and post-event (mean ratio_{Switch/Stay} = 0.92, range_{25%-75%} = 0.88

– 0.95; $t_{19} = -7.40$, $p < 10^{-6}$) but increased post-event by almost 2% (mean $\Delta\text{ratio}_{\text{Switch/Stay}} = 0.019$, $\text{range}_{25\%-75\%} = 0.011 - 0.029$; $t_{19} = 7.21$, $p < 10^{-6}$).”

R3.2.12

In methods, under spectral analysis, there seems to be some text missing “... 0.5 octaves (f/sigma.....sigma).”

Thank you, we added the missing values to the revised manuscript.

R3.2.13

While the language is generally clear, consider simplifying it for a broader audience to enhance readability

We thank the reviewer for the general advice and tried to keep the balance between concise, accurate and simple language in the manuscript.

References

VanRullen, R. (2018). Attention cycles. *Neuron*, 99, 632–634.

<https://doi.org/10.1016/j.neuron.2018.08.006>

Fiebelkorn, I. C., Saalman, Y. B., & Kastner, S. (2013). Rhythmic sampling within and between objects despite sustained attention at a cued location. *Current Biology*, 23, 2553–2558.

VanRullen, R. (2016). Perceptual cycles. *Trends in Cognitive Sciences*, 20, 723–735. <https://doi.org/10.1016/j.tics.2016.07.006>

Reviewer #4 (Remarks to the Author):

I co-reviewed this manuscript with one of the reviewers who provided the listed reports.

This is part of the Nature Communications initiative to facilitate training in peer review and to provide appropriate recognition for Early Career Researchers who co-review manuscripts.

We thank the reviewer for the effort and input on our manuscript.

We would like to thank the reviewers and the editor for their constructive feedback and careful evaluation of our manuscript. The constructive comments have helped us further improve the manuscript. Inspired by the comments, we performed several new analyses that further substantiate our main conclusions. We believe the reviewers' detailed assessment has strengthened the manuscript and enhanced the accessibility of our findings to a broader audience. Before providing point-by-point replies to the reviewers' comments, we first summarize the main changes and new results:

- 1) We clarified the nature of our localizer task and how we can use it to capture the dynamics of attention. In short, it is a retinotopic stimulus localizer and captures the strength of a stimulus representation in cortical visual maps in occipital as well parietal cortex. It is well established from previous work that attention manifests as a spatially-selective modulation of neural activity in these retinotopic maps. Thus, our approach is well suited for tracking the spontaneous allocation of attention in these visual field maps within the context of our 3-alternative forced choice task. Our localizer approach is *not* suitable for tracking the control of attention shifts. We apologize for our previously misleading wording. We have now clarified this in the manuscript, with a focus on the distinction between the expression and control of attention.
- 2) We directly quantified that attentional switches and stays occur predominantly during attention strength troughs and peaks, respectively. We included these quantifications as new Supplementary Figure S13.
- 3) In a new set of analyses, we tested both i) attention strength (Fig. S2 & R3) and ii) rhythmicity dynamics (Fig. R2) over the entire course of the trial. In agreement with previous results (Fig. 3-4) we conclude that our main findings are not constrained to certain trial epochs and that the chosen trial duration allowed us to meaningfully study attention dynamics.

Reviewer #1 (Remarks to the Author):

We thank the authors for addressing and clarifying several important issues raised. However, we still have a few remaining queries. We would appreciate it if the authors could address them to substantiate the claims made in the paper.

We would like to thank the reviewer for their continued work on our manuscript. The reviewer's insights and detailed questions helped us making the presented work more accessible. We will address all the raised points in the following.

Comments:

Re-R1.1.2 We continue to remain unclear about the significance of the medium decision value: if there are only two framing conditions, when does the medium value occur? What was the purpose of using this condition?

We thank the reviewer for this comment. This gave us the opportunity to add a more detailed description of the various experimental conditions in the Methods section of the manuscript. We had omitted these details in the previous version due to space constraints and because they are secondary to the main results we present.

The experimental task entailed judging which of the 3 simultaneously presented stimuli had the highest (high framing) or the lowest (low framing) contrast. This is a classical multi-alternative choice setting, analogous to, say, choosing which of three snack items one prefers (high framing; see also Gluth et al., Nat Hum Beh, 2020, link) or wants to avoid (low framing). Overall, we created 10 types of trials (Figure. R1 and now panel b in Figure 1) by systematically varying the contrast levels (relative to the background) of the 3 items, i.e. Gabor patches. Contrast levels mapped onto decision values differently, depending on the framing condition. Practically,

the alternative with the second highest contrast value is always the alternative with the medium decision-value regardless of the framing condition. This does not apply to the alternatives with the highest and lowest contrast, which can have the highest or lowest decision-value depending on the framing condition.

Figure R1 | Mapping from stimulus contrast levels to decision values depending on the framing condition. **a** Schematic of the ten unique Gabor grating contrast conditions. For each trial the three eccentric stimuli (red squares per row) were randomly drawn without replacement from a set of five distinct contrast levels (grey columns, from low to high contrast, i.e. darker to lighter shade, $n_{\text{trials,session}} = 1080$, 54 trials per condition-framing pairing within session). **b** Two example trials from the same stimulus condition (Condition 6) displaying distinct contrast-to-decision-value mappings depending on the framing condition. During “High” framing trials (top row) the contrast levels are equivalent to the decision value. During “Low” framing trials (bottom row) the mapping reverses and lowest contrast is the highest valued ($c1 = d5$), and the highest contrast the lowest valued ($c5 = d1$).

Figure R1 demonstrates how the mapping from contrast to decision values changes across framing conditions. To illustrate, we denote the minimum level of contrast with $c1$ and the maximum level with $c5$. In the “high” framing, contrast and decision values are equivalent. Consider an example trial including three options with contrast values: Left location = $c5$, Right location = $c3$, Middle location = $c1$. In the “high” framing condition the contrast values would map

onto the following decision values: Left = d5, Right = d3, Middle = d1. In this example, the correct option in the “high” framing condition is the Left one. In the “low” framing condition, the three options would map onto the following decision values Left = d1, Right = d3, Middle = d5, and the correct option is the Middle one. The framing manipulation allowed us to dissociate perceptual intensity from decision value.

Thus, within a framing condition, each of the 10 trials types featured a different combination of decision values. We were interested to examine how decision accuracy varies across these 10 trial types, which in turn can be informative when it comes to modelling the decision process. From a normative viewpoint, decision accuracy should vary as a function of the difference of the two better alternatives (highest vs. medium). For example, when 3 distinct options (X, Y, and Z) are presented, a trial with X = d5 (best option), Y = d2 (second best), Z = d1 (worst) should lead to higher accuracy relative to trials featuring X = d5 (best), Y = d4 (second best), Z = d1 (worst). Interestingly, previous research has shown that the value of the worst alternative (often referred to as the “distractor”) influences *relative* accuracy (see also Gluth et al., Nat Hum Beh, 2020, [link] and references therein), a phenomenon that normative and most descriptive decision-making models cannot capture. For example, the relative accuracy (i.e., $P(X)/(P(X)+P(Y))$, with P() denoting the probability of choosing the option in parentheses. Choosing X = d5 (best) over Y = d4 (second best) is typically higher when the worst alternative has a lower decision value (Z = d1) relative to when it has a higher decision value (Z = d3). As this so-called distractor effect is at odds with the predictions of normative frameworks, it can provide constraints about the underlying decision-making principles (for example Noguchi & Stewart, Psych Rev, 2018, [link]; Krajbich, Curr Op Psych, 2019, [doi]; Gluth et al., Nat Hum Beh, 2020, [link]).

A broader scope of this project was to shed light onto decision-making mechanisms by associating the distractor effect to patterns of information sampling. However, including this aspect in the present paper would be overwhelming and dilute the presentation of the central

rhythmic sampling findings, which we believe merit detailed presentation. We hope that the reviewer finds the updated description of the experimental design and trial structure sufficiently detailed. As a side note, we look forward to connecting (rhythmic) patterns of information sampling to the distractor effect (and assorted behavioural phenomena) in future work.

We summarize the outlined details of the 10 trial types in the revised Figure 1 and the design rationale in the revised Methods section (p. 23).

Re-R1.1.3 The authors have clarified the issue on fatigue and IOR due to long presentation duration. Unfortunately, it is still not clear to us whether the authors are claiming that there was fatigue (or not) during that period? Furthermore:

(a) Although the late interval was a reliable predictor of decision accuracy, the pupillometry results show that task engagement varied within the trial.

(b) As the authors mention in the next comment (R1.1.4), there was no significant difference in overall accuracy between shorter and longer time durations: does addition of extra time during the trial might still offer any benefits in terms of supporting participant behavior or performance? Moreover, as observed in figure 4 (in main text) and figure R2, the attention strength reaches its peak at 1s. Therefore, could the authors kindly provide further clarification regarding the necessity of long stimulus presentations for studying attention strength?

We thank the reviewer for following up on this point. Although we do observe a non-monotonic pattern of task engagement (obtained both through pupillometry results and decoded attention strength) – with engagement peaking mid-way through the trial – we are hesitant to attribute this effect to trial duration causing fatigue. The non-monotonic pattern could reflect fatigue; but alternatively, it could reflect a general decision-making strategy invariant to trial length (of course,

breaking down for extremely short durations). This strategy would involve participants arriving at an early decision as soon as possible and then reducing engagement in the remaining time as they are trying to confirm their decision (see also Resulaj et al., 2020, Nature, [link] for extensive discussion on “changes of mind”). Properly disentangling these interpretations would require systematically varying trial duration and concurrently examining the resulting patterns in decoded attention strength. Unfortunately, our MEG dataset features only a single trial duration. As mentioned in the previous reply, we did not attempt to optimize trial durations for behavioral performance in the MEG study but opted for a longer trial duration in order to facilitate the manifestation of information sampling patterns.

Although we can only speculate about the exact nature of these potentially non-monotonic patterns of decision formation, it is important to establish that our central findings regarding attention allocation were not time-dependent. The evidence we have provided throughout the manuscript and the replies show that including the later epoch (beyond 1 sec post-framing) is not detrimental to the analysis of dynamic attention processing (Fig. 3-4; Fig. S9). We thus come to the conclusion that our central findings regarding attention strength fluctuations (Fig. 3d-e) and properties of rhythmic sampling are time-invariant (see below).

To explicitly characterize the stability (or variability) of rhythmic sampling we windowed our analysis of attention allocation dynamics (see Fig. 5e) in seven 1 second windows (0.5 sec overlap) from the start of the sensory phase (-1 to 0 s post-framing) up until the end of the decision phase of the trial (2 to 3 s post-framing) and can identify rhythmicity in all windows with overlapping peaks (Fig. R2).

Figure R2 | Inter-event-interval histograms within different epochs of the trial. IEI histograms for **a** stay and **b** switch events color-coded from early (earliest -1s to 0s, relative to framing cue; blue) to late windows (latest 2s to 3s; green). The windows are half-overlapping and last 1 second each.

Overall, we do not find evidence that the reduced engagement (which, as described above, can be strategic or fatigue-driven) had any effect on the presence or detection of attention strength fluctuations (Fig. 3) or sampling dynamics (Fig. 5, 7, & R2). We will further discuss the effects of trial length with comments **Re-R3.1.8-9** (see below) and have added respective information in the manuscript (see for example p. 4 and p. 6 as well as the new Supplementary Fig. S2).

Re-R1.1.6 *We understand that the authors wanted to train an IEM without the confound of oculomotor dynamics. However, we wonder if it would not be convenient to design a localiser task closer in design to the main task. We would also like to know if the authors controlled for eye-movements in the main task. It would help if the authors could also elaborate on how the IEM is able to capture top-down attentional processes in the later periods of the trial.*

We thank the reviewer for pointing this out and we addressed misleading formulations from previous versions in the revised manuscript. We agree with all reviewers that have raised the issue that this is a crucial definition.

The localizer task is a retinotopic stimulus localizer, which captures the strength of the representation of any stimulus at each of the three task-relevant positions in the retinotopic maps of the cortical system (including visual cortex as well as higher-tier areas of the parietal cortex). We hypothesize that sequential attentional selection of different stimulus locations is at play in our main task where multiple stimuli compete for processing resources (for example McMains & Kastner, *JNeurosci*, 2011, [doi]; Womelsdorf & Fries, *Curr Op Neurobio*, 2007, [doi]). A large body of literature shows that top-down attention modulates visual stimulus representations in a spatially-selective manner in retinotopic maps of the cortical visual system (Silver, Rees & Heeger, *J Neurophys*, 2007, [doi]; Kastner et al., *Neuron*, 1999, [link]; Womelsdorf & Fries, *Curr Op Neurobio*, 2007, [doi]; Spyropoulos, Bosman & Fries, *PNAS*, 2018, [doi]; Siegel et al., *Neuron*, 2008, [doi]). Our localizer enables us to build an encoding model for the different stimulus locations in those retinotopic visual field maps. We attribute the fluctuating neuronal representations in early visual areas during the main task (that we quantify with our IEM approach) to the **effects of attention**; it may not enable us to track neural processes involved in the **control of attention** by networks of higher-tier fronto-parietal cortical areas (Fiebelkorn & Kastner, *Ann Rev Psy*, 2019, [doi]).

Thus, although we did not deem necessary to align the design of the localizer with the main task (of note, motivated by the reviewer's comment, we think that comparing results using different localizers would be an interesting future experimental direction), we agree with the reviewer that this is an important dissociation that needs to be explicated in the manuscript. We thus revised the discussion section (p. 24) and particularly elaborated this rationale with the introduction of the IEM approach in the revised manuscript (pp. 4 - 5):

“We developed an approach to continuously track the locus and strength of covert spatial attention during decision processing. Specifically, we used the retinotopic stimulus localizer to train an inverted encoding model[source] on the angular positions of the three stimulus positions occupied by the choice alternatives in the main task (Fig. 2a and Supplementary Fig. S1; see Methods). Presenting a single stimulus on one of these positions on each trial of the localizer task, enabled us to “tag” the strength of the representation of any stimulus at each position within the retinotopic maps of the visual cortical system[sources]. Top-down attention modulates visual stimulus representations in these visual field maps in a spatially-selective manner[sources]. Thus, our inverted encoding model enabled us to track to spontaneous allocation of attention to these different stimulus positions in the main decision-making task, where all three locations were simultaneously occupied by stimuli that competed for processing[source]: The expression of the allocation of attention during that task should manifest as a (stimulus-independent) increase or decrease, respectively, of the strength of the stimulus representations at different positions in the visual field maps. Specifically, the inverted encoding model yielded a time-resolved estimate of the response to each of the three angular positions during the decision-making task (Fig. 2b).”

Further, we are not claiming that we are decoding the control of the attention shifts. Indeed, the control of attention differs between top-down attention and bottom-up attention yet shares overlapping neuronal substrates (for example Fiebelkorn & Kastner, *Ann Rev Psy*, 2019, [doi]; Katsuki & Constantinidis, *Neuroscientist*, 2014, [doi]). Hence, our localizer would not be suited to track this difference.

Empirically, we observed that occipital and parietal activity is representing the three on-screen stimuli for the extended period of 3.8 seconds and displays competition between them (see Fig. 3-4, S1-5, S10). As we have pointed out this suggests that attentional selection is

expressed throughout the trial and not only right after visual stimulation. We anticipate that a large part of this attentional competition is top-down. This conclusion is based on findings prompted by reviewer comments during the previous revision round (Fig. S5 & S9). First, the task framing manipulation enabled us to dissociate the effect of low-level (contrast) from high-level features (decision value). We did not find any behavioral differences between the two framings. Second, low-level features did not affect the sampling frequency (Fig. S9) or attention allocation during the decision period (post-framing, Fig. S5). Taken together, we have evidence that attentional allocation was largely top-down in this task, however how top-down or bottom-up exactly interact during multi-alternative decision making needs to be addressed in the future.

We explicated the overlap between top-down/bottom-up attention control in the revised manuscript on p. 19:

“We can thus conclude that our findings, and particularly the dominant attention oscillation frequency around 11 Hz, were not solely driven by bottom-up sensory cortical responses (Supplementary Fig. S3, S4, S9). However, our analyses cannot fully disentangle bottom-up and top-down attentional control for each attention allocation[source].”

Lastly, eye movements have been tracked throughout the main task for all participants and have been reported in Fig. 8. We further computed (micro-)saccade rates for the main task separately for high- and low-framing conditions for all contrast pairs and did not find a significant difference for any time point during the task (all $p_{FDR} > 0.05$).

For completeness we note that we also controlled for eye-movements during the retinotopic stimulus localizer task (RS) and excluded RS trials that contained (micro-)saccades within the first 200 msec after stimulus onset from model training (model trained on 140-170 msec post stimulus onset). On average we excluded 3% of all sensory localizer trials per participant (mean = 3.03%, median = 1.82% range = 0.50% - 10.68%, 3 participants over 5%). We missed to report this detail in previous versions but revised the manuscript accordingly (p. 28).

Re-R1.1.8 & Re-R1.1.9 *We found it somewhat unconventional to normalize the attentional vector strengths with the stimulus presentation period, particularly given that there is active processing of the visual stimuli during this time. Would it not be convenient to use the pre-stimulus phase for normalisation where the random fluctuations of the attention vector would nullify each other and act as a more reliable baseline?*

We agree with the reviewer that the motivation for the normalization approach could be described clearer. The normalization is meant to build a reference frame to interpret the results and is thus part of its interpretation. In the presented case (Fig. 3a) normalization is meant to annotate trial dynamics of active processing. The applied z-score normalization indicates variations of attention strength throughout the stimulus presentation: when do participants display relatively stronger attention while the stimulus is on screen.

We believe that here the utility of the pre-stimulus period as a baseline is different to other experimental protocols involving visual stimulation: Our analysis quantifies the strength of visual representations (decodability) and not the strength of neuronal responses. Moreover, attention strength is represented by the length of the vector and thus does not nullify, even when neuronal activity would trend towards zero. Rather, it is small when no stimulus location is internally represented. Further, absolute values of attention strength strongly vary between participants. It is unclear what participants represented internally without external visual stimulation. Are participants for example thinking about the last stimulus or lunch? We argue the pre-stimulus period does not serve the purpose of a classic baseline period because it is thus under-controlled and changes relative to that, i.e. activity offsets, cannot be easily interpreted.

Z-scoring decodability, i.e. attention strength, while the stimulus is present can be interpreted as relative changes over time. A positive value indicates a time point when more

attention, than on average, is paid to the task/stimuli. Vice versa negative values indicate a relative decrease in attention. Thus, the normalization can identify periods within the trial of explicitly strong attention (“active processing”). For example, the pre-stimulus period - on average - demands less attention. We quantified the mean z-score differences between the pre-stimulus (-1.5 to -1 s from framing) and different windows during the stimulus presentation period and will discuss the findings with the next comment.

We further elaborated on the motivation for this specific normalization method in the revised manuscript (p. 6):

“We z-scored attention strength over the full stimulus-presentation window to compare visual representation dynamics between participants and identify the periods of strongest attention. Overall, during the full stimulus presentation attention strength is stronger than during the pre-stimulus period (Supplementary Fig. S2; $t_{19, vs.pre-stimulus} = 5.79 - 8.83, p < 1.5 \cdot 10^{-5}$).”

Re-R1.1.8 & Re-R1.1.9 Add *Additionally, in figure 3a, we observe that the attentional strength decreases below average after ~1s. This observation raises questions about the necessity of a longer task duration and the reliability of the results during the later stages of the trial.*

We thank the reviewer for the comment and see the interpretational pitfall that could emerge. Indeed, attention strength is below the stimulus presentation window average from ~1.5 s post-framing onwards. That, however, cannot indicate whether or not attention or visual sampling drops to uninterpretable levels per se. The z-scoring puts attention strength into a relative frame (to trial average) and can't be interpreted in absolute terms.

We do for example show that attention during later periods, although below average, is indeed very relevant for solving the task (Fig. 4 & 5c) and displays rhythmicity (Figure R2). Further, attention strength during the late stages of the trial is still higher than during the pre-stimulus period (Fig. R3 below). On average the z-scored attention strength in the latest trial stages (2.3-2.8s to framing cue onset) is less than 1 standard deviation below average ($\text{mean}_{z,\text{late}} = -0.83$, $\text{std} = 0.45$) while during the pre-stimulus period it is about 3 standard deviations below the stimulus presentation average ($\text{mean}_{z,\text{pre}} = -2.93$, $\text{std} = 1.51$). The latest trial stages thus show attention strength about 2 standard deviations higher than periods without stimulus presentation. This difference is statistically significant ($\text{mean}_{z,\text{late-pre}} = 2.10$, $\text{std} = 1.64$, $t = 5.75$, $p = 1.5 \cdot 10^{-5}$).

Figure R3 | Attention strength pre-stimulus and during late stimulus presentation. Z-scored attention strength compared between the pre-stimulus period (-1.5s to -1s, relative to framing cue) and late presentation stage (2.3s to 2.8s).

However, we agree that the interpretation might not come intuitively without quantifying all the information. We thus included the comparison of pre-stimulus attention strength versus the full stimulus presentation period in the main text (p. 6) and as a new Supplementary Figure S2.

Conclusively, we argue that the new analyses presented here (comments Re-R1.1.3, Re-R1.1.8, Re-R1.1.9) provide further evidence that relevant attentional sampling can occur all throughout the applied trial duration. All main findings presented in the manuscript pertaining to the rhythmicity of attention strength and allocation are unchanged over the entire trial period. Thus, our main findings are not constrained to certain trial epochs but generalize to the entire stimulus presentation period (3.8 s). Given this, and although trial duration was not something we tried to optimize in the task, we are confident that the chosen trial duration allowed us to meaningfully study attention dynamics.

***Re-R1.1.23** We appreciate the authors' explanation of the naive model and the "follow" interpretation. However, we think it would improve the readability further if the authors could elaborate the interpretation in more detail with its underlying assumptions and hypotheses.*

We thank the reviewer for urging us to improve the readability of the "Brief attentional saccades impact the ongoing attentional rhythm" section. This is an important section as it provides the foundation for fully understanding the reset interpretation in the following section.

We agree that there was room for improvement. Our central hypothesis is that attentional events impact the ongoing attentional rhythm, that is the 'reset' quantified in the following section and Figure 7. Apart from the literature addressing attentional reset phenomena to stimulus onset (for example Landau & Fries, CurrBio, 2012, [link]) it was previously unclear whether attentional events are similarly triggering resetting phenomena. Here, we cannot only directly test for the reset (i.e. interaction hypothesis) but also discard the alternative hypothesis that events only follow (and have no active effect on) the rhythm (Fig. 6).

In the revised manuscript we thus explicated both hypotheses more clearly in the beginning of the section on p. 12:

“The offset between switch and stay events can help us address another central hypothesis: Are attentional saccades impacting, for example by resetting[sources], the attentional rhythm? The related alternative hypothesis would be that attentional events passively follow the attentional oscillation (“follow” hypothesis), with stay at the peak and switch events occurring at the trough.”

We further described the implications of the follow hypothesis, namely the history effect of a time-invariant attention oscillation (p. 12):

“Under the follow hypothesis, event probabilities should depend on the previous event identity (Fig. 6a). For example, switch events should occur i) half a cycle after a previous stay event but ii) at full cycle latency following switches. This would induce a history effect on the IEI histograms: Depending on the preceding event, IEI modes should be temporally offset. Further, when ignoring the history this should result in IEI mode-latencies at double the frequency and no offset between switch- and stay events (Fig. 6a, lower panel).”

The implications of the follow hypothesis can be directly tested (and rejected) by quantifying the history effects through second-order event sequences (Fig. 6b,c). We can thus empirically and quantitatively show that events do not just follow the attention strength oscillation and reject the follow hypothesis, i.e. the alternative hypothesis to the interaction hypothesis. We broadly revised the section on p. 12 to clarify the presented analyses and better motivate the hypotheses tested in this and the following section.

Re-R1.1.24 *While we appreciate the arguments, they are qualitative in nature. It would be beneficial to provide additional quantitative evidence regarding the occurrences of “stay”*

and "switch" events to support the conclusions. Additionally, it would be helpful to include a more detailed quantitative explanation for the cosine fitting of the event oscillations.

We thank the reviewer for pushing for further clarity. Explanations that have grown for months and years with us authors might not always come across as obvious. We have thus included a new explanatory figure (Fig. R4) that summarizes the quantitative evidence for the stay-peak, switch-trough interpretation. We further conducted a new set of analyses that directly relates the attention strength fluctuation with event occurrences and is based on the results presented in previous versions (New Supplementary Fig. S13). For clarity the term "peak" will be reserved for the attention strength fluctuation, while local event-probability maxima are described as "mode". This terminology was adapted in the reply as well as in the revised manuscript.

Figure R4 | Temporal relationship between post-event attention strength and attentional events. **a** Inter-event-interval histograms (Fig. 5e) and boxplots indicating the distribution for participant specific 1st and 2nd event-probability modes for stay (orange) and switch events (green) separately. **b** Post-event attention strength detrended and smoothed for each participant (grey lines) and the group average (black line). The signal was detrended within the displayed 0.3 s window and smoothed for each participant. The boxplots indicate the participant specific 1st and 2nd peak/trough latencies. Peaks are local maxima ($n_{1st} = 16$, $n_{2nd} = 18$), troughs are local minima occurring before the peaks ($n_{1st} = 10$, $n_{2nd} = 16$). **c** Relative latency between the first post-event attention strength peak (see **b**) and the first- and second event mode (see **a**, color coded as above) in attention strength cycle length (see Fig. 7d). Values of zero indicate that an event occurred right at the attention peak. Negative and positive values indicate events occurring before or after the peak, respectively. All box plots display the median (red), 25%-75% range (colored box) and outliers outside 1.5 times the interquartile range (red cross).

We argue that stay and switch events occur predominantly at the peak and trough of the underlying attention strength fluctuation of around 9 - 12 Hz. This interpretation is based on a series of quantitative analyses presented in Figure 5 and 7 of the manuscript. We summarize these analyses below:

First, after an attentional event the next switch will occur preferentially around 0.05 s and 0.14 s (first and second mode); the next stay slightly later at 0.09 s and 0.17 s (see Fig. 5e-h and R4a, p. 14). Given the 9-12 HZ rhythm, these timings agree with the peak-trough interpretation.

Second, also following an event, the attention strength fluctuates at around 9 - 12 Hz (Fig. 7d, p. 19). These fluctuations are highly similar both after switch and stay events (mean $r_{post} = 0.60$, $r_{post,25\%-75\%} = 0.54 - 0.76$; $t_{19} = 2.59$, $p = 0.009$; p. 13). In the following we will thus average post-event attention strength over both event types to improve statistical power (Fig. R4b). Post-event attention strength displayed peaks at 0.09 s and 0.20 s ($std_{1st} = 0.014$ s; $std_{2nd} = 0.028$ s) as well as troughs at 0.06 s and 0.15 s ($std_{1st} = 0.015$ s; $std_{2nd} = 0.013$ s). Qualitatively, the attention strength peaks overlap well with the modes of stay event probabilities. Across the same vein, attention strength troughs overlap with switch modes (Fig. R4a vs. R4b box plots).

Now, we applied a new analysis to directly quantify the relationship between attention strength and attention event modes. We computed the latency between the first attention strength peak and the event modes and used the attention strength cycle length as unit of time (see Fig. 7d). A positive or negative value indicates that attentional events predominantly occur after or before the first attention peak, respectively. Further, due to normalizing by cycle length, an integer (full-cycle) difference indicates events occurring at a peak of the attention strength oscillation. Similarly, differences around half a cycle indicate events occurring during attention troughs. In Figure R4c we show that stay events occur right around the first and second attention peak (median latency_{1st stay} = -0.08 cycles, range_{25-75%,1st stay} = -0.13 – -0.01 cycles; latency_{2nd stay} = 0.97 cycles, range_{25-75%,2nd stay} = 0.79 – 1.04; see Fig. R4c and p. 20 of the revised manuscript), while switch events predominantly occurred half a cycle earlier during attention strength troughs (median latency_{1st switch} = -0.51 cycles, range_{25-75%,1st switch} = -0.54 – -0.47 cycles; median latency_{2nd switch} = 0.48 cycles, range_{25-75%,2nd switch} = 0.40 – 0.57).

All together, we find cohesive evidence for the stay-peak, switch-trough interpretation of our results. The additional analyses conducted here (see Fig. R4) directly quantify and explicate this interpretation. We thus added the new analyses as Supplementary Fig. S13 to the revised manuscript and described the results in the main text (p. 14):

“We further directly compared the temporal contingencies between post-event attention strength and the switch/stay IEI probabilities (Supplementary Fig. S13). We find that stay events cluster around post-event attention strength peaks (time from first attention peak: median_{1st stay} = -0.08 cycles, range_{25-75%,1st stay} = -0.13 – -0.01 cycles; median_{2nd stay} = 0.97 cycles, range_{25-75%,2nd stay} = 0.79 – 1.04), while switch events predominantly occurred half a cycle earlier during attention strength troughs (time from first attention peak: median_{1st switch} = -0.51 cycles, range_{25-75%,1st switch} = -0.54 – -0.47 cycles; median_{2nd switch} = 0.48 cycles, range_{25-75%,2nd switch} = 0.40 – 0.57). For each participant the cycle length was defined from the post-event fluctuation frequency (Fig. 7d).”

Reviewer #2 (Remarks to the Author):

In this resubmission, the authors have addressed my comments adequately as I briefly outline below for completeness and I have no further comments.

- The authors have substantially overhauled the abstract and introduction to focus more on the attentional sampling literature.

- The authors have well accommodated a point also raised by the other reviewers about the need to address whether bottom-up processing may be driving these effects. They have principally addressed this in a new analysis (Fig. 3) that shows that the key effects are present later on in the presentation interval suggesting that it is not driven solely by bottom-up effects which would be affected to be only early and transient. Another new analysis shows that the decoder still works when the task was to decode which stimulus was the lowest contrast: earlier in the trial the decoder is affected so the low-level features affect only the early response.

- The authors have clarified that the process described as “reset” relates principally to frequency and not phase, and they now better explain their definition of “Attention strength” as how well the decision alternatives are represented.

- A new figure illustrates how the two event types (switch and stay) are considered part of (and arising from) an ongoing oscillation of attention strength and the authors argue they are also indicative of two modes with distinct substrates.

(Note to authors: In the pdf I received of the rebuttal letter the Links didn't work, some references were listed as [source] and the logistic regression in Fig 5 was described as “logarithmic” but these problems did not extend to the actual manuscript.)

We would like to thank the reviewer very much for their time, expertise and effort put into this

manuscript. The revision process has strengthened the clarity of the manuscript and this is due to the reviewers' work.

Although the reviewer felt their comments adequately addressed, similar exchanges with other reviewers have been ongoing in the current revision. It thus might be of interest to the reviewer that we further elaborated points from the previous round in the revised manuscript. This includes: First, we explicated the relation of our findings to bottom-up/top-down processing (pp. 5 & 19). Second, we clarified the nature of the (retinotopic stimulus) localizer task and its inverted encoding during the main task to capture the manifestation of attention rather than the control of attention (p. 5). Third, we included a direct quantification of our conclusion that stay/switch events occur at the peak/trough of the attention strength oscillation in the main text and as a new supplementary figure (p. 14, Supplementary Fig. S13).

Reviewer #3 (Remarks to the Author):

We thank the authors for their extensive revision of the manuscript. Overall, the manuscript has become much clearer with a better focus. However, we still have some concerns that we would like the authors to address.

We would like to thank the reviewer for their continued work on our manuscript. Their pending questions are important issues and we gladly address them in the following.

Comments:

Re-R3.1.1 *The procedure of IEM is clear but the rationale behind using the sensory localizer (SL) task to reconstruct representations of covert spatial attention still needs some clarification in the main text. The use of a task that does not involve covert shifts of attention to prevent micro-saccadic contamination of the data is well motivated. However, using the SL task, in which participants were given both instructions to ignore the peripheral stimuli and a specific central flickering detection task to prevent covert attentional shifts, seems counter-intuitive when the aim is to extract cortical representations of covert attentional shifts. As mentioned in my first review, if there were shifts in attention during the SL task, they are most likely bottom-up driven as the stimuli were behaviorally irrelevant for the task. However, from figure S1, there is a sustained representation in the data related to the Gabor patch location (i.e. indeed probably an attentional shift). The sensor contribution during the optimal decoding time shows contribution from the occipital cortex which suggests that the SL task initially evoked visual responses to the Gabor patch locations (which is expected). However, for the sustained*

attentional information, we would expect the sensor contributions to cover more parieto-frontal areas. We would therefore like to see the sensor contribution after the initial transient increase in decoding accuracy (e.g. 0.2-0-35s) as this would clarify the information that is encoded in the IEM.

We agree with the authors that the use of the IEM on data from the three-alternative decision-making task seems to be able to reconstruct the strength and locus of covert attention (and plausibly other intertwined cognitive processes), due to the tight correlation with behavior that you show during the task. However, we would like the authors to elaborate on alternative interpretations related to the origin of the IEM (i.e. built from the SL task). For example, the weight matrix G represents how each MEG sensor weighs different positions of the Gabor patches, which were displayed without behavioral relevance (i.e. MEG activity should dominantly reflect visual processing and bottom-up driven attention). When applying this G matrix on MEG data from the three-alternative decision-making task (i.e. multiplying the inverse G with MEGMDM), it should weigh the MEGMDM data based on the activity evoked by the Gabor patches in the SL task. If this activity dominantly represents visual information processing and bottom-up attention, from a functional perspective, what does this mean? Is there a functional overlap between visual, bottom-up and top-down attention that you are capturing? We believe it is crucial to clarify and discuss this in the main text.

We thank the reviewer for pointing this out and we addressed misleading formulations from previous versions in the revised manuscript. We agree with all reviewers that have raised the issue that this is a crucial definition.

The localizer task is a retinotopic stimulus localizer, which captures the strength of the representation of any stimulus at each of the three task-relevant positions in the retinotopic maps of the cortical system (including visual cortex as well as higher-tier areas of the parietal cortex).

We hypothesize that sequential attentional selection of different stimulus locations is at play in our main task where multiple stimuli compete for processing resources (for example McMains & Kastner, JNeurosci, 2011, [doi]; Womelsdorf & Fries, Curr Op Neurobio, 2007, [doi]). A large body of literature shows that top-down attention modulates visual stimulus representations in a spatially-selective manner in retinotopic maps of the cortical visual system (Silver, Rees & Heeger, J Neurophys, 2007, [doi]; Kastner et al., Neuron, 1999, [link]; Womelsdorf & Fries, Curr Op Neurobio, 2007, [doi]; Spyropoulos, Bosman & Fries, PNAS, 2018, [doi]; Siegel et al., Neuron, 2008, [doi]). Our localizer enables us to build an encoding model for the different stimulus locations in those retinotopic visual field maps. We attribute the fluctuating neuronal representations in early visual areas during the main task (that we quantify with our IEM approach) to the **effects of attention**; it may not enable us to track neural processes involved in the **control of attention** by networks of higher-tier fronto-parietal cortical areas (Fiebelkorn & Kastner, Ann Rev Psy, 2019, [doi]).

Thus, although we did not deem necessary to align the design of the localizer with the main task (of note, motivated by the reviewer's comment, we think that comparing results using different localizers would be an interesting future experimental direction), we agree with the reviewer that this is an important dissociation that needs to be explicated in the manuscript. We thus revised the discussion section (p. 18) and particularly elaborated this rationale with the introduction of the IEM approach in the revised manuscript (pp. 4 - 5):

“We developed an approach to continuously track the locus and strength of covert spatial attention during decision processing. Specifically, we used the retinotopic stimulus localizer to train an inverted encoding model[source] on the angular positions of the three stimulus positions occupied by the choice alternatives in the main task (Fig. 2a and Supplementary Fig. S1; see Methods). Presenting a single stimulus on one of these positions on each trial of the localizer task, enabled us to “tag” the strength of the representation of any stimulus at each position within the retinotopic maps of the visual

cortical system[sources]. Top-down attention modulates visual stimulus representations in these visual field maps in a spatially-selective manner[sources]. Thus, our inverted encoding model enabled us to track to spontaneous allocation of attention to these different stimulus positions in the main decision-making task, where all three locations were simultaneously occupied by stimuli that competed for processing[source]: The expression of the allocation of attention during that task should manifest as a (stimulus-independent) increase or decrease, respectively, of the strength of the stimulus representations at different positions in the visual field maps. Specifically, the inverted encoding model yielded a time-resolved estimate of the response to each of the three angular positions during the decision-making task (Fig. 2b).”

Further, we are not claiming that we are decoding the control of the attention shifts. Indeed, the control of attention differs between top-down attention and bottom-up attention yet shares overlapping neuronal substrates (for example Fiebelkorn & Kastner, *Ann Rev Psy*, 2019, [doi]; Katsuki & Constantinidis, *Neuroscientist*, 2014, [doi]). Hence, our localizer would not be suited to track this difference.

Empirically, we observed that occipital and parietal area activity is representing the three on-screen stimuli for the extended period of 3.8 seconds and displays competition between them (see Fig. 3-4, S1-5, S10). As we have pointed out this suggests that attentional selection is expressed throughout the trial and not only right after visual stimulation. We anticipate that a large part of this attentional competition is top-down. This conclusion is based on findings prompted by reviewer comments during the previous revision round (Fig. S5 & S9). First, the task framing manipulation enabled us to dissociate the effect of low-level (contrast) from high-level features (decision value). We did not find any behavioral differences between the two framings. Second, low-level features did not affect the sampling frequency (Fig. S9) or attention allocation during the decision period (post-framing, Fig. S5). Taken together, we have evidence that attentional

allocation was largely top-down in this task, however how top-down or bottom-up exactly interact during multi-alternative decision making needs to be addressed in the future.

We explicated the overlap between top-down/bottom-up attention control in the revised manuscript on p. 19:

“We can thus conclude that our findings, and particularly the dominant attention oscillation frequency around 11 Hz, were not solely driven by bottom-up sensory cortical responses (Supplementary Fig. S3, S4, S9). However, our analyses cannot fully disentangle bottom-up and top-down attentional control for each attention allocation[source].”

Overall, the distinction between the **control** of visuo-spatial attention (likely fronto-parietal) and its manifestation (**expression**) on visual areas is pivotal to interpret the decoding results based on the applied IEM decoder. We have revised the manuscript to explicate this distinction and better explain the rationale for the nature of the retinotopic stimulus localizer task as outlined above both during the introduction of the decoder (pp. 4 - 5) and the discussion (pp. 18 - 19).

Re-R3.1.3 The revised introduction does a better job incorporating literature on attentional rhythms, and the discussion of oscillations across theta, alpha, and gamma is a welcome addition. However, we noticed that some key points outlined in your response to our comment, particularly the discussion of how task complexity and peripheral stimulus presentation might influence the observed attentional frequency—are not explicitly reflected in the revised text.

In the discussion you added a brief section stating that the 11Hz rhythm might further depend on the cognitive domain and task and stimulus complexity. However, it would be appropriate to elaborate a bit more (like in your response) on how your observed 11 Hz frequency compares to previous findings, which have reported attentional sampling at both

lower (4-8 Hz) and higher (7-12 Hz) frequencies. Additionally, prior work suggests that information sampling across hemispheres is often associated with theta rhythms (e.g., Fiebelkorn, 2013; vanRullen, 2016). Given that your task involves sampling across hemispheres (Gabor patches were displayed across hemispheres), could you clarify or attempt to explain why a theta rhythm was not observed in your data? Including these points would provide a clearer connection between your findings and the existing literature.

We agree with the reviewer that interpreting the exact sampling frequency with respect to previous research will better bridge our findings with the existing literature. We thus appended the discussion of the results in the revised manuscript (p. 19):

“Rhythmic attention has been associated with the 4 – 8 Hz range[sources] but also found with higher sampling rates (7 – 12 Hz)[sources]. The exact frequency channel, here 11 Hz, might depend on the cognitive domain and also the task and stimulus complexity[sources]. For example, tasks with extensive processing demands of the attended stimuli[sources], peripherally presented stimuli[sources] and as recently shown sampling extracted from stimulus decoding[source] can display compatible rhythms. However, further research is needed to explicate the influence of external (e.g. binary vs. ternary decisions) and internal factors (e.g. neuromodulation) on the sampling speed.”

Re-R3.2.4 *Thank you for the clarification. However, we still find this point somewhat unclear in the text. Do you mean that Figure 2B illustrates the estimated responses and attentional vector, while Figure S3 demonstrates that—regardless of the option values (High, Medium, or Low)—these responses consistently show a broadband decrease*

between 8–45 Hz? If so, it might help to state this more explicitly in the manuscript to ensure clarity.

We thank the reviewer for the comment and their interpretation of the previous version is correct.

To clarify the interpretation, we explicated the control analysis in the revised manuscript (p. 7):

“Similarly, we assessed the dynamics of the estimated responses (Fig. 2b), the precursor to the attentional vector, and remapped each response to its decision value class (High, Medium, Low). The estimated responses for all alternatives displayed a broadband decrease between 8 – 45 Hz (Supplementary Fig. S4) and could not explain the increase of attention strength between 7 – 13 Hz.”

Reviewer #3 (Remarks on code availability):

It seems that the link above is a view-only link. I cannot find any files or data.

We thank the reviewer for bringing this issue to our attention. We updated the online repository and made the access public. Please find the files and scripts under the following link:

<https://osf.io/tf6bw/>

Reviewer #4 (Remarks to the Author):

I co-reviewed this manuscript with one of the reviewers who provided the listed reports.

This is part of the Nature Communications initiative to facilitate training in peer review and to provide appropriate recognition for Early Career Researchers who co-review manuscripts.

We thank the reviewer for the effort and input on our revised manuscript.

We would like to thank the reviewers and the editor for their constructive feedback on our revised manuscript. We have addressed all the outstanding comments. Specifically, we have addressed the minor comments (pertaining to typos) raised by Reviewer 2. As pointed out by Reviewers 3 we have now made the code repository publicly accessible.

In the text below we provide point-by-point replies to Reviewer's 1 pending concerns. We quantified if participants' choices could be predicted from time-resolved neural representations of decision alternatives, i.e. decoded MEG activity. Particularly, late in the trial, increased sampling of the highest-valued alternative was associated with correct choices. These empirical findings provide converging evidence that neuronal processing of the alternatives as well as increased attention strength fluctuations (see Fig. 4) contribute to a participant's overt behavior until the trial end. This new analysis allowed us to draw stronger conclusions from our findings, to refute the concern that participants may disengage from the task in later trial intervals, and to temporally disentangle rhythmic attentional sampling from decision weighting.

The major updates to the manuscript are:

- 1) We computed a time-resolved analysis predicting single-trial choices from the decoded neural representations of decision alternatives. We show this analysis in a new Supplementary Figure S7. The results of the analysis enable us to refute the concern that participants may disengage from the task in later trial intervals. As another result, we further discuss time-varying effects and relations between information sampling and decision-making processes in the revised manuscript (Discussion).
- 2) We included the windowed analysis of inter-event-intervals (first brought up in the previous revision round) as a new Supplementary Figure.

Reviewer #1 (Remarks to the Author):

Thank you for your efforts with addressing our comments. We have a few pending concerns.

We appreciate the reviewer's pending concerns and addressed these in our replies below.

Comments:

R1.1.2 - We appreciate the additional explanation and background reference provided regarding the task design. However, examining the referenced study raises new questions. In that study, participants valued the stimuli before the task, whereas in the present work, responses are determined by the experimenter based on contrast differences, which may introduce a substantial bottom-up component of attention. This again raises the question of whether participants maintained attention for an extended duration. Additionally, we understand that the use of three stimuli was meant to increase task difficulty by introducing more distractors, but the authors seem to suggest that the justification for this design is beyond the scope of the current paper.

R1.1.3 - While we agree with the authors' analysis, the specific points regarding pupillometry and behavioral results remain insufficiently addressed. The authors demonstrate that attention allocation is consistent throughout the trial duration. However, our concern persists regarding whether participants are actively sampling the stimuli during this period to make a decision, or whether decisions are made almost immediately after stimulus onset given the strong bottom-up component of the task.

We greatly appreciate both comments as they help us explicate the functional relevance of attentional fluctuations in the main task. We want to address R1.1.2 and R1.1.3 together as they overlap and represent common concerns: a) the participants' ability to maintain attention throughout the trial, b) the presence and role of bottom-up attentional biases in our "perceptual" task, relative to the "valued-based" one that we referenced in our previous reply, c) the reasons why we opted for a 3-alternative choice task.

Regarding a), the Reviewer questions whether participants could maintain attention for the extended duration of the trial. This is a critical point, given the long duration of each trial, and the reviewer is right in pushing for further analyses and clarifications of that aspect. In principle, attentional allocation may or may not be directly related to choices (e.g., people could be sampling information without utilizing it for their subsequent choice). For instance, information sampled during the sensory (pre-framing) and decision (post-framing) phases of the trial, is bound to be differently utilized towards making the final choice. Our approach allows us to directly ask: how does decodability at different time-points influence the overt choices participants make?

In Figure 5c we established the overall link between whole-trial sampling aspects (the number of switch and stay events) and choices. But this analysis lacks the temporal resolution necessary to address the reviewer's concern. Thus, to establish the link between decoding strength and choice in a time-resolved manner, we used logistic regression to predict choices from the attention allocation to the different decision alternatives (see Supplementary Figure S7, also shown in Figure R1). These analyses rely on the decodability for the high minus the decodability for medium alternative ($IEM_{high} - IEM_{mid}$) as regressor¹ to predict whether the "high" or the "medium" alternative was chosen (red line). The results show choices were influenced by attentional sampling early around 0.5 seconds and later from 1.8-2.8 seconds post-framing. Thus,

¹ We used the decodability difference between "High" and "Mid" because in 3-alt choices, the nominal decision variable is the value of the best minus the value of the second-best alternative (McMillen, & Holmes, 2006, *J Math Psych*, [link]).

attentional allocation during both early and late intervals (with a gap at around 1-1.5 secs) predicted behavioral performance. We do not find any evidence that choice-predictive sampling monotonically dissipated with time from framing (as would be predicted by purely bottom-up driven effects) but rather increased again towards stimulus offset and the concurrent go-cue. We note that the bottom-up component is in principle strong but, as our results suggest, showed negligible effects on decision-related stimulus decodability. The biphasic pattern after frame-cue onset contradicts the view that the late decision period reflects disengagement and instead indicates active engagement, linking neural activity to choice behavior.

Figure R1 | Neural representation of decision alternatives predicts choices. Time resolved logistic regression predicting choice from the relative decodability of decision alternatives ($IEM_{high} - IEM_{mid}$), i.e. increased stimulus representation of the highest decision alternative, at each time point. The blue line and shaded area indicate the mean and SEM over participants. Thick lines at the bottom depict statistically significant differences from zero ($|t|_{19} > 0$, $p < 0.05$, cluster-mass permutation corrected). Dashed vertical lines denote the framing cue onset, offset and stimulus offset (from left to right).

More broadly, this finding concurs with the observation that during protracted decisions, incoming evidence is weighted differentially across time (for example Tsetsos, Usher & McClelland, FiNS, 2011, [link]). Interestingly, a robust conclusion in our manuscript is that the parameters of rhythmic

sampling and attention are time-invariant (see Fig. 3e, S2 and Fig. R2, replicated from the previous revision that we added to the Supplement (Fig. S15) in the revised manuscript).

Figure R2 | Inter-event-interval histograms within different epochs of the trial. IEI histograms for **a** stay and **b** switch events color-coded from early (earliest -1s to 0s, relative to framing cue; blue) to late windows (latest 2s to 3s; green). The windows are half-overlapping and last 1 second each. The Figure has been replicated from the previous revision round and was added as Supplementary Fig. S15 to the revised manuscript.

This points to the existence of two dissociable, yet interlinked, processes. The one is a constant, time-invariant sampling mechanism that supplies information to downstream time-dependent decision processes. These downstream decision processes dynamically modulate the gain (i.e., the effective impact) of incoming information on choice. Previous work has considered this dynamic weighting as emerging from evidence accumulation mechanisms, such as accumulation leak or winner-take-all competition among alternatives (Usher & McClelland, Psych Rev, 2001, [link]), or from explicit top-down weighting (Cheadle et al, Neuron, 2014, [link]; Lam et al., J neurosci, 2022 [link]). In other words, whether attended information is integrated into the final decision depends not only on whether an alternative is sampled but, due to downstream decision mechanisms, also on when it is sampled (Fig. S7). Our manuscript characterizes the former and paves the way for better understanding the latter. To explicitly elucidate this temporal distinction

between sampling and decision weighting we added Fig. R1 as Supplementary Fig. S7 (see p. 9) and appended the matter in the Discussion (p. 20):

“Yet, while attentional rhythmicity appeared to be stable over time (Fig. 3e and Supplementary Fig. S15) its effect on overt behavior might vary with time (Fig. 4b and Supplementary Fig. S7). In other words, the effect attended information has on the upcoming decision does not only depend on whether[sources] a decision alternative is sampled but also on downstream decision mechanisms, such as e.g. accumulation leak[source] or adaptive gain[source], that modulate incoming information. This points to temporally dissociable, but interlinked, processes of attention and decision-making. Future research is needed to explicitly elucidate the temporal relationship between the two processes.”

Regarding b), we agree that information processing in our perceptual decision-making task could be influenced by bottom-up attention² because our stimuli varied in contrast². However, we found no empirical evidence for such modulation, as detailed below. The framing manipulation allows us to assess the extent to which bottom-up influences are in play, by comparing behavior and neural decoding in the “high” vs. “low” framing trials. First, there was no significant difference in choice accuracy between the ‘high’ and ‘low’ framing trials (Figure 1c, paired t-test choice probability (per alternative) “low” – “high” framing trials; $t_{19,p(H)} = 1.46$, $p = 0.16$, $t_{19,p(M)} = -1.68$, $p = 0.11$, $t_{19,p(L)} = -0.53$, $p = 0.60$). This suggests that participants’ choices were not driven by bottom-up brightness (e.g., higher accuracy in “high” trials, where the correct option had also the highest brightness). Second, neural decodability did not differ between “high” and “low” trials during the interval following the framing cue. For convenience, we append below Supplementary Figure S5

² Behavior in value-based tasks, like the one in Gluth and colleagues (Nat Hum Beh, 2020, [link]), can also be impacted by low-level features, such as the brightness of the packaging of each snack item. In passing we note that we opted for a perceptual task where the choice criterion is objective (unlike the value-based task, there is a correct answer on every trial) and where the decision-value of each option is under full experimental control.

as Fig. R3 (also included in the point-by-point replies accompanying the previous revision) showing this result. The only significant decodability differences are seen early in the pre-framing interval (circa -1 sec) for the extreme-value options (high value, S5a; low value S5c), whose contrasts reversed across the “high” and “low” trials. This pattern indicates that decodability for the high-contrast option is stronger only in the pre-framing interval and only for a brief period after stimulus onset. Together these analyses suggest that bottom-up attention did not influence the subsequent decision processes.

Supplementary Fig. R3 | Differences in estimated responses between framing conditions. a-c Trial-averaged decodability (estimated response) differences for High- minus Low-Framing trials separately for **a** high, **b** medium, and **c** low decision value option within each respective trial. The circles schematically depict the compared contrast levels. Thick lines and shaded areas display the mean difference and the standard error, respectively. Here, contrast affected decodability immediately after stimulus onset (-1 sec) (c.f., S5a,c) and this effect dissipated quickly.

Finally, regarding c), the use of three stimuli was a deliberate choice to investigate how people sample information during *multialternative* decisions (i.e., choices among more than two options). As outlined in the Introduction, such decisions remain poorly understood at the mechanistic level, with most prior research (both experimental and theoretical) focusing on binary choices. In our original submission, we had not described the 10 distinct stimulus conditions in detail; this was clarified in our previous point-by-point replies (see Figure 1b). The design of these 10 conditions was informed by the “distractor” effect, a behavioural phenomenon previously documented in a value-based paradigm (for example Cao & Tsetsos, eLife, 2022, [link]; Chau et

al., eLife, 2020, [link]; Louie, Khaw, & Glimcher, 2013, PNAS, [link]). However, analyzing this effect was beyond the scope of the manuscript. For our purposes here, analyses aggregated across these 10 conditions, as we wanted to examine the (generic) patterns of rhythmic sampling from three alternatives (irrespective of condition-specific differences).

R1.1.6 - We appreciate the acknowledgment that the localizer used is not optimal for disentangling top-down and bottom-up processes between the localizer and the main task. The explanation is helpful; however, the approach misses support from existing literature to establish the protocol's validity. The argument that task framing can dissociate low-level from high-level features remains unconvincing, as participants were always asked to choose one of the extreme stimuli. The eye-tracking controls are clarified sufficiently.

We thank the reviewer for this comment. We have acknowledged that our localizer is agnostic about *processes*, bottom-up or top-down. However, we respectfully disagree with parts of this assessment and maintain that the framing manipulation can dissociate the *influence* of a specific low-level feature (contrast) from that of a specific high-level feature (decision value) on decodability. This is achieved by comparing decodability for the extreme options (high and low decision value) across “high” vs. “low” framing trials (Fig. 5c and Fig. R2; see also reply above) provide strong evidence that the sensory localizer task offers a valid method of reading out the attentionally modulated sensory signals that can drive decisions.

R1.1.8 & R1.1.9 - The analysis to show increased attention in the late-trial period (when stimulus is on the screen) compared with pre-stimulus periods seems to confirm an expected result. Our main concern persists that the reported late-trial attentional dynamics may not be necessary for participants' decision-making.

Thank you for urging us to explore the relevance of late-trial attentional dynamics. With assessing the relationship (impact) of time-resolved IEM signals on overt behavior (see Fig. R1) we addressed this concern with our reply to R1.1.2 and R.1.1.3.

R1.24 - The details regarding methodology used for cosine fitting is not fully addressed and the references to appropriate literature to support the method are lacking.

We would like to note that we did not apply cosine fitting within the presented manuscript and hope to resolve the pending issue. For describing temporal dynamics emerging from event-related activity (Fig. 5, 7, 8, S12, and S14) we consistently used the inverse of peak-to-peak (mode-to-mode) latencies with an algorithm further detailed in the Methods section (pp. 31-32 and 33-34).

During previous replies to the reviewer's comments, we semantically compared the rhythmic patterns identified in Fig. 5-7 to sinusoidal wave-forms (for example: *"between events, the oscillation is best described as a cosine with time zero set at the previous event"*, reply to revision 1, R1.1.24). Nevertheless, these were semantics and we did not apply cosine-fitting. We apologize if our explanation caused the misunderstanding.